# Gene refashioning through innovative shifting of reading frames in mosses

Yanlong Guan[1], Li Liu [2], Qia Wang[1], Jinjie Zhao[1], Ping Li[2,3], Jinyong Hu[1], Zefeng Yang[4], Mark P. Running[5], Hang Sun[1] & Jinling Huang[1,6,7]

Early-diverging land plants such as mosses are known for their outstanding abilities to grow in various terrestrial habitats, incorporating tremendous structural and physiological innovations, as well as many lineage-specific genes. How these genes and functional innovations evolved remains unclear. In this study, we show that a dual-coding gene *YAN/AltYAN* in the moss *Physcomitrella patens* evolved from a pre-existing hemerythrin gene. Experimental evidence indicates that *YAN/AltYAN* is involved in fatty acid and lipid metabolism, as well as oil body and wax formation. Strikingly, both the recently evolved dual-coding *YAN/AltYAN* and the pre-existing hemerythrin gene might have similar physiological effects on oil body biogenesis and dehydration resistance. These findings bear important implications in understanding the mechanisms of gene origination and the strategies of plants to fine-tune their adaptation to various habitats.

[1] Key Laboratory for Plant Diversity and Biogeography of East Asia, Kunming Institute of Botany, Chinese Academy of Sciences, 650201 Kunming, China. [2] Key Laboratory of Economic Plants and Biotechnology, Yunnan Key Laboratory for Wild Plant Resources, Kunming Institute of Botany, Chinese Academy of Sciences, 650201 Kunming China. [3] University of Chinese Academy of Sciences, 100049 Beijing, China. [4] Jiangsu Key Laboratory of Crop Genetics and Physiology, Co-Innovation Center for Modern Production Technology of Grain Crops, Key Laboratory of Plant Functional Genomics of the Ministry of Education, Yangzhou University, 225009 Yangzhou, China. [5] Department of Biology, University of Louisville, Louisville, KY 40292, USA. [6] Institute of Plant Stress Biology, State Key Laboratory of Cotton Biology, Henan University, 475001 Kaifeng, China. [7] Department of Biology, East Carolina University, Greenville, NC 27858, USA. These authors contributed equally: Yanlong Guan, Li Liu. Correspondence and requests for materials should be addressed to J.H. (email: huangj@ecu.edu)

Early-diverging lineages of land plants (embryophytes) provide some unique opportunities to understand the mechanisms of plant adaptation to terrestrial environments. During their early adventure of land colonization, plants faced tremendous challenges, such as drought, intense UV radiation, and scant soil nutrients. On the other hand, an open terrestrial environment free of competition by many other organisms might have provided ecological opportunities, as evidenced by the dramatic structural and morphological changes in a diverse group of early land plant lineages and their charophycean ancestors[1,2]. Although significant progress has been made recently[3–5], how early land plants adapted to hostile terrestrial environments remains elusive. It is clear that many of the structural and physiological innovations in early land plants are linked to lineage-specific genes and the functions they introduced[6–8]. The origin of such genes might be attributable to multiple mechanisms such as gene duplication, de novo gene origination, and horizontal gene transfer (HGT)[9,10]. Nevertheless, the dynamics and interplay of these mechanisms in land plant evolution have rarely been illustrated.

Mosses, liverworts, and hornworts are extant representatives of the earliest land plant lineages. These plants have simpler structures, and many of them lack cuticle, stomata, true vascular tissues, and other features that are critical for efficient water conduction and conservation in more complex land plants (e.g., vascular plants)[1]. The moss *Physcomitrella patens* (Funariaceae) is frequently used as a model to study the biology of early land plants[6,11]. In this study, we provide evidence that a gene encoding hemerythrin (Hr), a versatile oxygen-binding protein[12,13], was likely transferred from fungi to early land plants. We show that this *Hr* gene subsequently evolved into a dual-coding gene with two overlapping reading frames in the *Physcomitrium–Physcomitrella* species complex. Even more intriguingly, both the later evolved dual-coding gene and the pre-existing *Hr* gene might have similar physiological effects on oil body biogenesis and dehydration tolerance. These findings provide major insights into the origin of new genes and demonstrate how complex mechanisms of gene evolution facilitated the adaptation of plants to terrestrial environments.

## Results

**Distribution and origin of the plant *Hr* gene**. The plant *Hr* gene encodes a single HHE cation-binding domain (pfam 01814) and has been annotated in the genomes of nonvascular plants, such as the liverwort *Marchantia polymorpha*, the moss *Sphagnum fallax*, and the seedless vascular plant *Selaginella moellendorffii*. With land plant Hr sequences as query, a BLAST search of the NCBI non-redundant (*nr*) protein sequence database identified hits (*E* value cutoff = $1e-8$ with the substitution scoring matrix BLOSUM45) predominantly in fungi and bacteria, a green alga *Monoraphidium neglectum*, and an angiosperm *Quercus suber* (Supplementary Fig. 1a and Supplementary Note 1). With the *M. neglectum* sequence (NCBI accession number: XP_013901669) as query, we were able to identify hits from other green algae, including *Gonium pectorale*, *Volvox carteri*, *Tetrabaena socialis*, and *Chlamydomonas reinhardtii* (Supplementary Fig. 1b). All these identified green algal hits belong to Chlorophyceae, a class of chlorophyte green algae. In contrast, although many other green algae and red algae have been sequenced and their data are deposited in NCBI, no hits were found from these taxa, including charophytes, a paraphyletic group from which land plants evolved. Search of additional databases, including the JGI Plant Genomics Resource Phytozome (http://phytozome.jgi.doe.gov) and *Cyanophora* Genome Project (http://cyanophora.rutgers.edu/cyanophora) identified homologs in another chlorophycean alga

*Chromochloris zofingiensis*, as well as the glaucophyte *C. paradoxa*. With the only exception of tardigrades that reportedly acquired *Hr* from fungi[14], we were unable to detect this gene in any other animal. Furthermore, algal Hrs appear to be less similar to land plant homologs. For instance, the conserved HHE domain of land plants shares 31–49% protein sequence identities with fungal homologs, but 28–32% with chlorophycean and glaucophyte sequences (Supplementary Fig. 2).

To further assess the distribution of the *Hr* gene in plants, we performed some additional comprehensive searches of transcriptomic data such as the 1000 Plants (1KP) database (https://db.cngb.org/blast/onekp), transcriptomes of the charophytes *Nitella hyalina* and *Closterium peracerosum–strigosum–littorale* complex in NCBI, as well as the over 650 transcriptomes in the Marine Microbial Eukaryote Transcriptome Sequencing Project (MMETSP) (http://marinemicroeukaryotes.org/resources). BLAST search of 1KP database using *Marchantia* and *Sphagnum* Hr protein sequences (JGI IDs: Mapoly0112s0056.1 and Sphfalx0048s0034.1) as query identified homologs in all major lineages of nonvascular land plants (liverworts, hornworts, and mosses) and in many seedless vascular plants and angiosperms (*E* value threshold = 0.01) (Supplementary Table 1). Although over 200 species from various algal groups were covered in 1KP, most identified algal homologs belonged to Chlorophyceae, and only sporadic hits from other lineages were detected. When these other algal sequences were used to search the *nr* database, they usually shared the highest sequence coverage and identity with either fungal or bacterial homologs (Supplementary Note 1; also see related discussions below). Such a higher similarity with fungal and bacterial sequences is noteworthy, particularly considering the availability of complete genome sequence data from many algal species in NCBI. BLASTN search of the transcriptomic data of charophytes *N. hyalina* and *C. peracerosum–strigosum–littorale* complex using coding sequence (CDS) of *Marchantia* Hr (JGI ID: Mapoly0042s0093) as query identified matches to two to three fragments of 19–32 nucleotides; these matches, however, were part of miscellaneous genes other than *Hr* (Supplementary Note 2). Consistent results were also generated in the search of MMETSP. Overall, these data again suggest that algal and land plant Hr sequences may not be particularly related.

Because no *Hr* has been annotated in any published angiosperm genomes, the presence of many hits from angiosperms in 1KP was surprising. We therefore searched the *nr* database using the Hr sequences derived from these angiosperms and other land plants, particularly hornworts and seedless vascular plants. With the only exception of two lycophyte genera (*Selaginella* and *Isoetes*), all other seedless vascular plant Hr sequences shared the highest percent identities with fungal homologs. The same scenario was also observed for Hr sequences of angiosperms and two hornworts, *Phaeoceros carolinianus* and *Paraphymatoceros hallii*. We further performed PCR amplification and sequencing of *Hr* in the hornwort *P. carolinianus* and the fern *Pteris vittata* (GenBank accession numbers: MG254885 and MG254886). The resulting sequences were nearly identical with those in 1KP. Because no scaffold information is currently available for these sequences, we cannot confidently exclude the possibility that these sequences might be derived from symbiotic fungi (e.g., mycorrhizae).

Of the two lycophyte genera containing *Hr* homologs, an N-terminal oleosin domain is fused to *Hr* in *Isoetes* (*I. tegetiformans* and *I. sp.*; 1KP IDs: PKOX-2096679 and PYHZ-2006808, respectively). We were also able to confirm this gene fusion event in another species of *Isoetes* (*I. yunguiensis*) through reverse transcription PCR (RT-PCR) reactions and

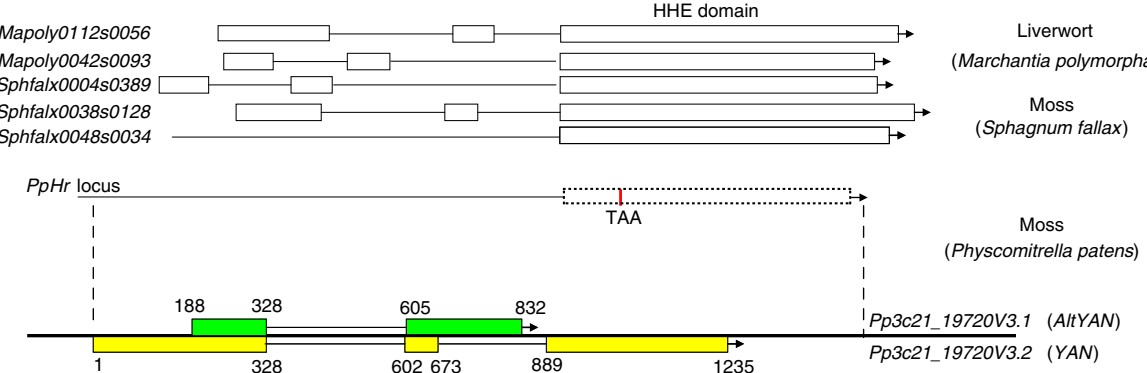

**Fig. 1** Gene structure of the dual-coding gene *YAN/AltYAN*. Two copies of the *Hr* gene exist in the liverwort *M. polymorpha* and three copies in the flattop bog moss *S. fallax*. The identifiers of *Hr* sequences from Phytozome are indicated. The *Hr* locus in *P. patens* contains no start codon ATG but a premature stop codon TAA. The dual-coding gene *YAN/AltYAN* evolved from the pre-existing *Hr* locus. The two transcripts, *YAN* and *AltYAN*, are shown in yellow and green, respectively, and their exon positions are indicated

sequencing (Supplementary Fig. 3). No such gene fusion was detected in eight *Selaginella* species covered in 1KP.

Phylogenetic analyses of Hrs indicated that sequences from liverworts, mosses, as well as the lycophytes *Selaginella* and *Isoetes* formed a clade with tardigrade and fungal homologs, which in turn grouped with other fungal sequences with strong support (Supplementary Fig. 4a, b). All other vascular plant Hr homologs identified in 1KP were affiliated with various fungal sequences (Supplementary Fig. 4a), suggesting that they might indeed be derived from sequencing contamination. As expected, land plant Hr sequence clade was not closely related to chlorophycean homologs (Supplementary Fig. 4b).

**A dual-coding gene evolved from the pre-existing *Hr* locus.** The length of *Hr* CDSs ranges from 636 (JGI ID: Sphfalx0048s0034) to 957 (JGI ID: Sphfalx0004s0389) base pairs (bp) in land plants. In *M. polymorpha* and *S. fallax*, two to three intact *Hr* copies were identified in our analyses. Except *Sphfalx0048s0034*, which only contains the HHE cation-binding domain, each of the other copies contains three exons, with exon 3 being the largest and containing the HHE domain (Fig. 1). Two *Hr* homologs were also identified in *S. moellendorffii*, but both appear to only possess the HHE domain without introns, although these annotations remain to be verified experimentally. In comparison, only a partial *Hr* nucleotide sequence was found in *P. patens*. This partial *Hr* sequence is located on chromosome 21 of *P. patens* and corresponds to exon 3 (i.e., the HHE domain) of *M. polymorpha* and *S. fallax* (Fig. 1). Compared to *M. polymorpha* and *S. fallax*, no start codon ATG was found in the corresponding *Hr* locus of *P. patens*; instead, a premature stop codon TAA was identified in the HHE domain region (Fig. 1). Our further PCR reactions and sequencing confirmed that these annotations in *P. patens* were not caused by genome sequencing errors. These data suggest that the *Hr* gene is either lost or has evolved into new functions in *P. patens*.

During our additional investigations on *Hr*, we found two overlapping transcripts on the pre-existing *Hr* locus in *P. patens*. These two transcripts use different open reading frames, and they were annotated as *Pp3c21_19720V3.1* and *Pp3c21_19720V3.2*, respectively, in Phytozome. *Pp3c21_19720V3.2* is 1235 bp in length with two introns, whereas *Pp3c21_19720V3.1* is only 645 bp long and completely internalized within *Pp3c21_19720V3.2*. The latter also contains a single intron that largely matches (only 3-bp shorter) the first intron in *Pp3c21_19720V3.2* (Fig. 1 and Supplementary Fig. 5). Because the two transcripts use different reading frames, they constitute a single dual-coding gene.

We here refer to the longer transcript *Pp3c21_19720V3.2* as *YAN* (a popular name of newborns in Chinese, meaning "stunning and fascinating") and the alternative transcript *Pp3c21_19720V3.1* as *AltYAN*. To verify the identity of *YAN/AltYAN* as well as the deactivation of the pre-existing *Hr* gene, we performed RT-PCRs to amplify their CDSs. For both transcripts, their CDSs were amplified and their sizes were consistent with the genome annotation (Supplementary Fig. 6). Additional sequencing of the RT-PCR products confirmed the splice sites of *YAN/AltYAN*, as annotated by Phytozome. To verify that both messenger RNAs were intact rather than functional relics of *Hr*, we designed five forward primers (F1–F5) upstream of this locus, and, in cooperation with a reverse primer R2, conducted various RT-PCR reactions (Supplementary Fig. 6). These experiments failed to amplify any appropriate product, suggesting that *Hr* has evolved into *YAN/AltYAN* through frameshift in *P. patens*.

To assess the taxonomic distribution of *YAN/AltYAN*, we further searched the *nr* and 1KP databases, as well as other published transcriptomic data of nonvascular plants such as *Ceratodon purpureus* and *Funaria hygrometrica*, using the encoded protein sequences of the two transcripts. Particularly, the current 1KP database contains transcriptomic data of over 90 nonvascular plants, including an unspecified species of *Physcomitrium*, a closely related genus and a paraphyletic group in which *Physcomitrella* is embedded[15]. No homolog was identified in *nr*, whereas only fragmented sequence matches, often with premature stop codons, were found in 1KP. In particular, the search of 1KP for *AltYAN* provided no hit outside *Physcomitrium*; only a single hit (1KP ID: YEPO-2062682), which corresponded to the 5′UTR and the first exon of *AltYAN*, was identified in *Physcomitrium*. Our additional RT-PCR reactions and sequencing, however, recovered transcripts of both *Hr* and *YAN/AltYAN* in another unspecified species of *Physcomitrium* (Supplementary Figs. 7 and 8). These findings suggest that this dual-coding gene *YAN/AltYAN* exists in the *Physcomitrium–Physcomitrella* species complex.

**YAN and AltYAN are localized in oil bodies and chloroplasts.** We investigated whether *YAN/AltYAN* encoded proteins, and, if so, their cellular localization. The online targetP server (http://www.cbs.dtu.dk/services/TargetP) predicted that AltYAN was likely localized in chloroplasts, but failed to provide a clear hint about the localization of YAN. We then tagged the two CDSs with a *GFP* gene and visualized the cellular localization of target proteins via protoplast transient expression in *P. patens*. The

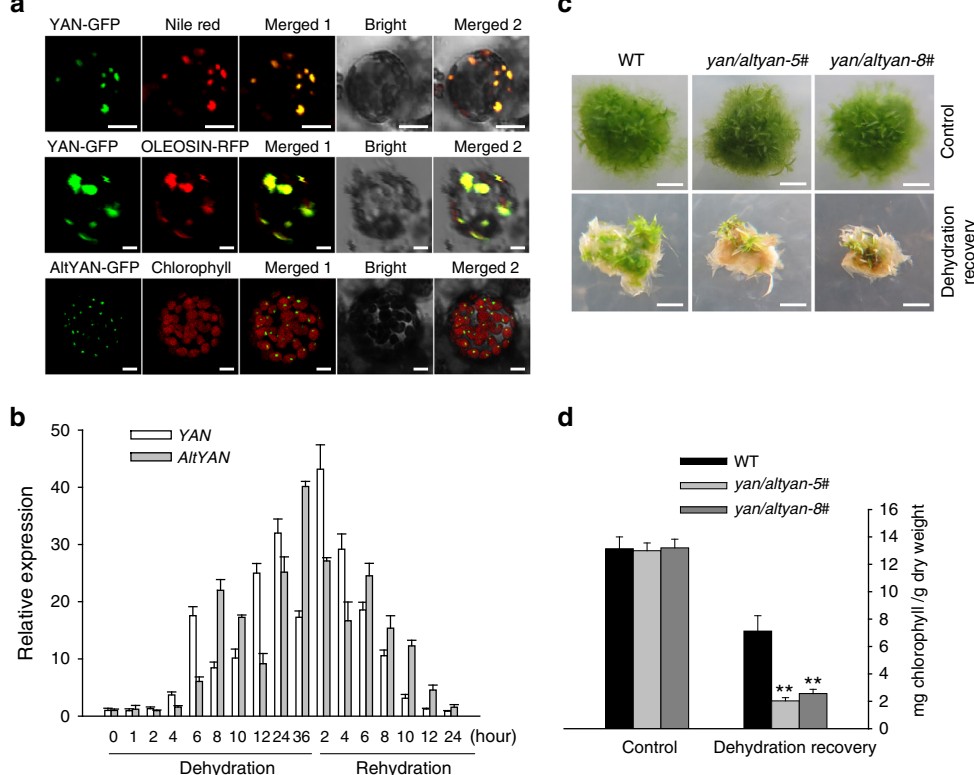

**Fig. 2** Protein subcellular localization and recovery of *P. patens* clones after dehydration treatments. **a** YAN is localized in oil bodies as indicated by the green fluorescence of YAN-GFP merged with Nile red and OLEOSIN-RFP (red), respectively. AltYAN-GFP is localized in chloroplasts as shown by the merged green fluorescence of AltYAN-GFP with chlorophyll auto-fluorescence (red). Scale bar = 4 μm. **b** Expression profile of *YAN/AltYAN* under dehydration and rehydration. Results represent means and standard deviations of three biological replicates for the wild-type plants. **c** Phenotypes of wild-type plants and *yan/altyan* mutants under control and dehydration treatments. Wild-type and *yan/altyan* plants were dehydrated for 20 h and transferred into sterile water for 1 h, and then transferred onto standard medium for recovery. Photos were taken after plants were recovered for 2 weeks. Scale bar = 2 mm. **d** Chlorophyll contents of wild-type and *yan/altyan* plants after 2 weeks of recovery. Results represent means and standard deviations of three biological replicates for the wild-type and two independent mutants. Double asterisks indicate a statistically significant difference compared with wild-type plants based on a two-tailed Student's *t* test ($p < 0.01$)

results indicated that both transcripts encoded proteins. According to the approximate locations of these proteins, we used mitochondria marker plasmid *p35S:PpCOX11-RFP*, peroxisome marker plasmid *p35S:PpPEX11-RFP*, Golgi marker plasmid *p35S:PpCTL1-RFP*, and oil body marker plasmid *p35S:PpOLEOSIN-RFP* to co-transfect the protoplast with *p35S:YAN-GFP* and *p35S:AltYAN-GFP*, respectively. As shown in Fig. 2a, the green fluorescence of YAN-GFP overlapped the red fluorescence of oil body dye Nile red and oil body marker *p35S:PpOLEOSIN-RFP*, indicating that YAN is localized in oil bodies. The green fluorescence of AltYAN-GFP, on the other hand, overlapped the red auto-fluorescence of chloroplasts, indicating that AltYAN is localized in chloroplasts (Fig. 2a).

**The novel dual-coding gene mediates dehydration tolerance**. To understand the physiological processes in which *YAN* and *AltYAN* are involved, we searched Phytozome for the expression profile of the two transcripts. Both transcripts were found to be strongly induced by dehydration and rehydration (Supplementary Fig. 9). We further confirmed this expression pattern by quantitative real-time polymerase chain reaction (qRT-PCR). As shown in Fig. 2b, the transcription levels of *YAN* and *AltYAN* increased in a fluctuating manner during dehydration and decreased gradually after rehydration. These data are in line with the localization of YAN and AltYAN in oil bodies and chloroplasts (Fig. 2a); while chloroplasts are involved in many cellular

activities including stress responses[16,17], oil bodies are often linked to plant dehydration tolerance[18].

We then investigated the role of *YAN* and *AltYAN* in dehydration stress response. The two transcripts were knocked out altogether in *P. patens* through homologous recombination using the polyethylene glycol-mediated method[19] (Supplementary Fig. 10a). The knockout mutants, namely *yan/altyan*, were identified by genotyping the transformants using PCR and semi-quantitative RT-PCR (Supplementary Fig. 10b, c). A total of five *yan/altyan* mutants were generated, and they were all haploids based on flow cytometric analyses (Supplementary Fig. 11). Previous studies suggest that *P. patens* wild-type plants are able to tolerate severe dehydration and return to normal growth[20,21]. To assess the capacity of dehydration tolerance of *yan/altyan*, both wild-type and *yan/altyan* plants were dehydrated for 20 h, which led to about 86% water loss of fresh weight, and then rehydrated for 1 h before they were transferred to standard medium for recovery. After 2 weeks, *yan/altyan* remained severely damaged, whereas wild-type plants displayed marked recovery (Fig. 2c). Compared to wild-type plants, the chlorophyll content (mg g$^{-1}$ dry weight) of *yan/altyan* decreased by 64–72% (Fig. 2d). More superoxide anions and toxic nitric oxide (NO) were observed in *yan/altyan* (Supplementary Fig. 12).

**The dual-coding gene is involved in oil body biogenesis**. Because *YAN* and *AltYAN* contain no known protein domains,

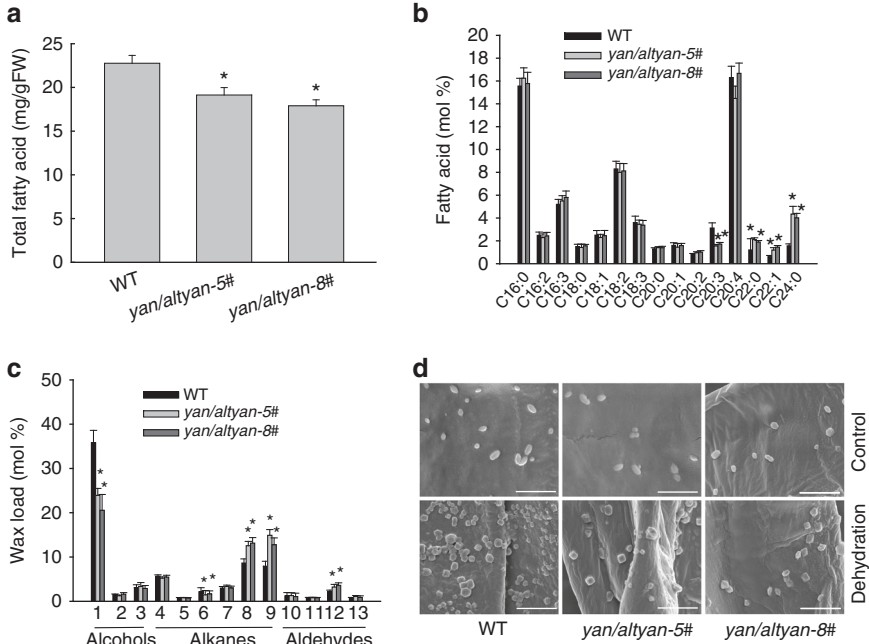

**Fig. 3** Quantification of fatty acids and wax loads in wild-type plants and yan/altyan mutants. Ten-week old plants were used to detect the composition and total amount of fatty acids and waxes. Results represent means and standard deviations of three biological replicates for the wild-type and two independent mutants. Asterisks indicate a statistically significant difference compared with wild-type plants based on a two-tailed Student's t test (p < 0.05). **a** Total fatty acid content in the wild-type and yan/altyan. **b** Fatty acids composition in wild-type plants and yan/altyan mutants. **c** Wax composition in wild-type plants and yan/altyan mutants. 1. 16-Hydroxy ent-kaurane; 2. 1-Tetracosanol; 3. 1-Hexacosanol; 4. Octadecane-1-bromo; 5. Kaur-15-ene; 6. Ent-kaurene; 7. 1,19-Eicosadiene; 8. 1,21-Docosadiene; 9. 1,21-Tetracosene; 10. Octadecanal; 11. Eicosenel; 12. Docosanal; 13. Tetracosanal. **d** Wax crystals observed using scanning electron microscopy. Scale bar = 2 μm

we searched the gene co-expression data of Phytozome in order to infer their functional roles. Both YAN and AltYAN were found to be co-expressed with long-chain acyl-CoA synthetase 8 (LACS8) (Supplementary Fig. 9), a member of the LACS gene family that is critical in cellular fatty acid and lipid metabolism[22,23]. This finding is in agreement with the fact that YAN is localized in oil bodies, a reservoir of neutral lipids, and AltYAN in chloroplasts, the primary site for plant fatty acid biosynthesis[24,25]. To further investigate the role of YAN and AltYAN in fatty acid and lipid metabolism, we first analyzed the changes in fatty acid contents in yan/altyan mutants. Compared to wild-type plants, yan/altyan had about a 13% reduction in total fatty acids (Fig. 3a). In addition, yan/altyan demonstrated a 47.5% decrease in C20:3, but 66–157% increase in C22:0, C22:1, and C24:0 (Fig. 3b). We then generated three and four over-expression mutants for YAN and AltYAN (i.e., YAN-OE and AltYAN-OE), respectively, using the pPOG1 vector (PpEF1-α promoter-driven) (Supplementary Fig. 13). As expected, YAN-OE and AltYAN-OE exhibited an about 66 and 14.5% increase in total fatty acids, respectively (Supplementary Fig. 14a). Significant variation in the amount of individual fatty acids was also detected in the two over-expression mutants (Supplementary Fig. 14b). Taken together, the above evidence suggests that both YAN and AltYAN are indeed involved in the metabolism of fatty acids and lipids.

Because dehydration leads to an increase in both the number and size of oil bodies[18], we also investigated the effects of dehydration on oil body biogenesis in the wild-type and yan/altyan plants. Ten-day old protonemata and 4-week old gametophores were treated by fast dehydration on filter paper for 30 min. Both treated plants and untreated controls were stained with Nile red dye (25 μg ml⁻¹ methanol), and their oil bodies were then observed. As shown in Fig. 4a, c, oil bodies were hardly noticeable in wild-type and yan/altyan plants under normal growth conditions, but were plentiful in YAN-OE and

AltYAN-OE. Compared to wild-type plants, YAN-OE and AltYAN-OE also became yellowish and exhibited delayed growth (Fig. 4c). This finding indicates that over-expression of YAN and AltYAN led to excessive oil body accumulation, which in turn might have impaired the growth of P. patens. After dehydration, numerous oil bodies could be readily observed in both wild-type and yan/altyan plants, but its total number decreased by 44% in yan/altyan (Fig. 4a, b).

Given the role of lipid metabolism in sexual reproduction of plants[26], we also investigated the development of sporophytes of P. patens. Plants were grown for 8 weeks at 25 °C before they were transferred to a short-day regime at 15 °C. In both wild-type and mutant (yan/altyan and over-expression) plants, sporophytes became visible after 4 weeks of induction (Supplementary Fig. 15). The mature spores were spread on BCDAT medium to assess their viability. After 4 days of incubation, spores began to germinate and grew into chloronemata (Supplementary Fig. 16). These results indicate that YAN/AltYAN may have a very limited effect on the sexual reproduction of P. patens.

**The dual-coding gene affects wax content and crystals.** Wax crystals are a major component of cuticle, which protects plants from water loss. Because long-chain fatty acids and very long-chain fatty acids are precursors for wax production[27], we investigated whether the changes in fatty acid content in yan/altyan and over-expression mutants were associated with variation in wax production, by quantifying the amount and composition of waxes. Compared to wild-type plants, the total amount of several wax molecules, including 1,21-tetracosene, 1,21-docosadiene, and docosanal, increased significantly in yan/altyan. Particularly, the amount of 1,21-tetracosene in yan/altyan increased by 74.8% (Fig. 3c). As expected, the amount of these wax molecules decreased in YAN-OE and AltYAN-OE, (Supplementary Fig. 17).

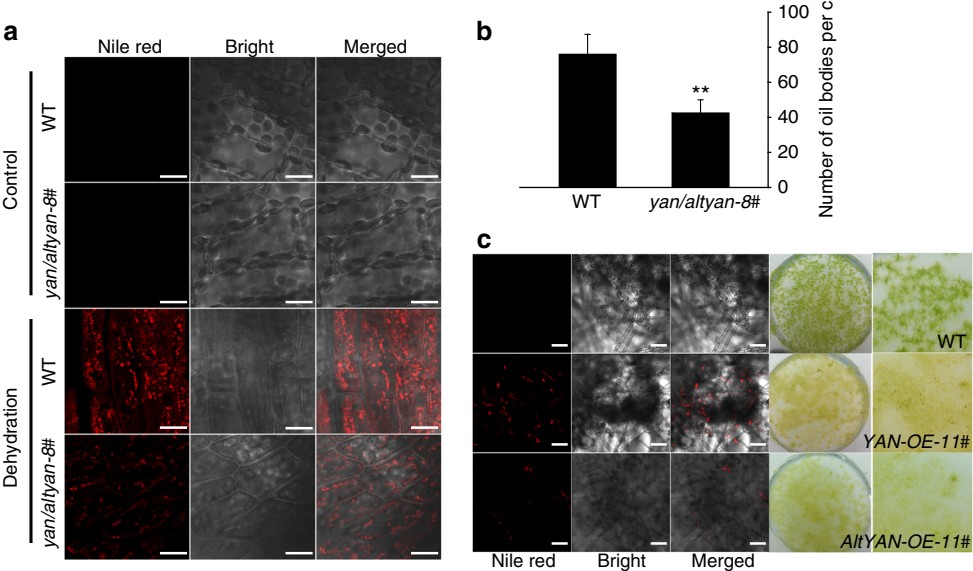

**Fig. 4** In situ oil bodies and their quantification. **a** Accumulation of oil bodies in wild-type plants and *yan/altyan* mutants. Four-week old plants were dehydrated for 30 min, and plants under normal growth conditions were used as control. Plants were stained with Nile red dye (25 µg ml$^{-1}$ methanol) for 12 h before observation. Scale bar = 20 µm. **b** Quantification of oil bodies in wild-type plants and *yan/altyan* mutants under dehydration treatments. Results represent means and standard deviations of three biological replicates for the wild-type and *yan/altyan-8#*. Double asterisks indicate a statistically significant difference compared with wild-type plants based on a two-tailed Student's *t* test ($p < 0.01$). **c** *YAN-OE* and *AltYAN-OE* plants accumulated excessive oil bodies under normal conditions and showed delayed growth. Two-week old wild-type as well as *YAN-OE* and *AltYAN-OE* plants were used to observe oil bodies. Scale bar = 0.2 mm

On the other hand, the amount of two other wax molecules, including ent-kaurene and 16-hydroxy ent-kaurane, decreased by 27.8% and 46.2%, respectively in *yan/altyan* (Fig. 3c). While *YAN-OE* and *AltYAN-OE* had about a 75.2–84.1% increase in ent-kaurene, an 85.3–88.1% reduction in 16-hydroxy ent-kaurane was observed in the two over-expression lines (Supplementary Fig. 17).

Under both normal growth conditions and dehydration treatments, fewer wax crystals were observed on cell wall surfaces of *yan/altyan* knockout mutants. This difference in the number of wax crystals was particularly pronounced when plants were treated by dehydration (Fig. 3d).

## Discussion

Organisms adapt to shifting environments frequently through changes in their gene repertoires. In prokaryotes, this process is often reflected in their fluid genomes that are "sampled" from a global gene pool[28]. In eukaryotes, lineage-specific genes of distinctive functions have been under some extensive studies[9,29], and it has been shown that all non-coding genomic regions may potentially evolve into functional genes[30]. However, other than gene duplication and subsequent functional differentiation, rapid and dramatic refashioning of pre-existing genes into new functions has rarely been illustrated.

The origin of a dual-coding gene *YAN/AltYAN* from a preexisting *Hr* locus provides new insights into the mechanisms of gene origination. Several previous studies showed that the evolutionary history of *Hr* in eukaryotes, including tardigrades, has been complicated by HGT[12,14,31]. In our current study, *Hr* homologs could be clearly identified in glaucophytes, Chlorophyceae, and land plants. It is likely that land plant *Hr* was ultimately derived from other organisms, possibly fungi, but other scenarios, such as vertical inheritance combined with lineagespecific gene loss, cannot be ruled out. Additional investigations are needed to understand the origin of *Hr* in land plants. HGT

has traditionally been considered pervasive in prokaryotes and frequent in unicellular eukaryotes[28,32], but also reported in complex multicellular eukaryotes such as animals and plants[33,34], including mosses[33]. In particular, HGT from fungi to earlydiverging land plants appears to be a common scenario[5,33]. Although we cannot determine whether the *Hr* gene in hornworts and ferns was replaced by an independently acquired copy from fungi, it appears that this gene has been secondarily lost in seed plants. Such gene loss is consistent with the observation that horizontally acquired genes are often related to metabolism and functionally peripheral[35]. As such, they are frequently involved in the adaptation of recipient organisms to shifting environments and short-lived[36].

An interesting finding from our study is the refashioning of *Hr* into *YAN/AltYAN* in *P. patens* through nonsense mutations and translational frameshift. The existence of both *Hr* and *YAN/AltYAN* in *Physcomitrium* (Supplementary Figs. 7 and 8), suggest that this refashioning event occurred at least in the *Physcomitrium–Physcomitrella* species complex. Future thorough investigations on the distribution of *YAN/AltYAN* are needed to understand when this dual-coding gene evolved. Even more astonishing is the fact that the recently evolved *YAN/AltYAN* gene contains two overlapping transcripts (i.e., *YAN* and *AltYAN*), indicating that functional innovation can be rapidly achieved through gene recruitment and refashioning. Conceivably, the refashioning of *Hr* into *YAN/AltYAN* might reuse the pre-existing regulatory sequence, thus allowing smooth transition of the two reading frames into a fully expressed dualcoding gene[37]. Dual-coding through overlapping reading frames (overprinting) is a mechanism of de novo CDS origination[38]. Such a mechanism is most common in viruses and represents a genomic novelty to encode more proteins in a compact genome[39]. In animals, several experimentally confirmed dual-coding genes have been reported[40–42], in which the same gene encodes two distinct proteins using alternative transcripts. Unlike most dualcoding genes reported in the literature, both transcripts, *YAN* and

*AltYAN*, in *P. patens* contain introns. In certain respects, *YAN/AltYAN* represents the combination of classic examples of human *INK4a*[40] and *ATXN1*[42]; it is similar to *INK4a* in containing introns and to *ATXN1* in the alternative transcript (*AltYAN*) being completely internalized in the regular transcript (*YAN*). Because mutations in overlapping coding regions usually cause amino acid changes in two proteins, it has been suggested that protein products encoded by regular and alternative transcripts often perform related functions or interact with other[43,44]. Indeed, both transcripts, *YAN* and *AltYAN*, are involved in the fatty acid and lipid metabolism (Fig. 3). To our knowledge, *YAN/AltYAN* represents the first experimentally demonstrated dual-coding gene in plants.

Origin of new genes (functions) could potentially expand the adaptive abilities of organisms[9,45]. Our experimental evidence from *yan/altyan* mutants, as well as *YAN-OE* and *AltYAN-OE* lines, clearly indicates that the involvement of *YAN/AltYAN* in lipid metabolism is linked to oil body biogenesis (Fig. 4). This evidence is further bolstered by the observation that YAN is localized in oil bodies (Fig. 2a). In mosses and many other land plants (e.g., seed plants), oil bodies play an important role in desiccation tolerance of plant tissues[18,46]. In fact, oil bodies have long been suggested as a strategy of plants to tolerate water loss, and some early land plants might kill living tissues in exchange for oil body production under prolonged periods of dehydration[47]. In line with the role of *YAN* and *AltYAN* in oil body biogenesis and lipid metabolism, fewer wax crystals were observed in *yan/altyan* knockout mutants under dehydration treatments (Fig. 3d). Wax crystals on plant surface are the main barrier to water loss, UV radiation, insect herbivory and pathogen infection[48]. Such a critical role of oil bodies and wax crystals in various stress-related processes also explains the observation that *yan/altyan* became hypersensitive to dehydration (Fig. 2c).

Compared to wild-type *P. patens*, *yan/altyan* exhibited noticeable changes in wax composition, including a significant reduction in the amount of ent-kaurene and 16-hydroxy ent-kaurane (Fig. 3c), both of which are generated in *P. patens* in the biosynthetic pathway of gibberellins (GAs)[49,50]. In particular, ent-kaurene is a common precursor for all GAs in plants. Although *P. patens* and other nonvascular plants lack bioactive GAs[51], it has been demonstrated that ent-kaurene regulates protonema differentiation and blue-light avoidance response in *P. patens*[50,52]. 16-Hydroxy ent-kaurane, on the other hand, has a restricted distribution in mosses and can be released into the air[53]. This volatile nature of 16-hydroxy ent-kaurane at least partly explains the fact that 16-hydroxy ent-kaurane decreased in *YAN-OE* plants, which also exhibited delayed growth (Figs. 3c and 4c). Thus far, the biological role of 16-hydroxy ent-kaurane remains largely elusive, but this compound has been suggested as a protective mechanism against microbes and photo-oxidation[53,54], again pointing to a role of *YAN/AltYAN* in stress tolerance. It is also notable that the de novo biosynthesis of both fatty acids and GAs occurs in plastids[25,55], where AltYAN is localized (Fig. 2a), indicative of a possibly inherent link between AltYAN and these two processes.

The HHE cation-binding domain may function either alone as single-domain Hrs or as part of multi-domain proteins, where it plays a key role in iron sensing and homeostasis[56]. The single-domain Hrs are involved in miscellaneous physiological processes related to oxygen sensing, storage, and transport in bacteria and animals[13,57], but their functions in fungi and plants have not been investigated intensively. Nevertheless, circumstantial evidence suggests that Hr is also involved in oil body biogenesis in land plants. In particular, a mosaic gene derived from fusion of oleosin and *Hr* is present in the lycophyte *Isoetes* (Supplementary Fig. 3). It has long been accepted that genes linked by fusion events are usually functionally associated, and their protein products either interact with each other directly or participate in related activities[58,59]. Oleosins are the major component of oil body surface, and they are essential for maintaining the structural stability of oil bodies[18,60]. As such, the functional linkage between *Hr* and oleosin strongly suggests that the pre-existing *Hr* in *P. patens* most likely participated in oil body biogenesis before evolving into the dual-coding *YAN/AltYAN*. This suggestion is consistent with the observation that *Hr* is induced by dehydration and rehydration in two resurrection plants, the twisted moss *Syntrichia ruralis* and the spikemoss *Selaginella lepidophylla* (Supplementary Fig. 18), and that the two processes are mediated by the same sets of genes[61].

If the pre-existing *Hr* gene in *P. patens* was indeed functionally related to oleosins, it would have similar physiological effects as the dual-coding *YAN/AltYAN*; both genes would be involved in oil body biogenesis and dehydration tolerance (Fig. 2), despite their apparent lack of common functional domains. This phenomenon suggests that similar physiological effects and adaptive strategies can be achieved convergently through gene refashioning or replacement. Thus far, we are not aware of any other similar examples, and experimental evidence for *Hr* in land plants, particularly *P. patens* and its close relatives (e.g., *Physcomitrium*), is still lacking. On the other hand, it is possible that the functional innovations associated with *YAN/AltYAN* could provide additional benefits to *P. patens* to fine-tune its adaptation to specific habitats. This is evidenced by the potential role of 16-hydroxy ent-kaurane in biotic and abiotic stress tolerance[53,54]. Such an expansion of adaptive capabilities is conceivably critical for nonvascular plants, which lack sophisticated protective structures and often rely on numerous secondary metabolites or other molecules to tolerate biotic and abiotic stresses[62].

As a model organism of early-diverging land plants, *P. patens* provides some unique opportunities to understand the origin and role of lineage-specific genes in organismal adaptation to shifting environments. The origin of dual-coding *YAN/AltYAN* from a pre-existing *Hr*, which in turn was likely acquired from fungi, highlights a rapid process of gene origination during the evolution of early-diverging land plants. Innovative shifting of reading frames results in dual-coding genes, with primary and alternative transcripts encoding distinct, but often functionally related proteins. Presumably, if nonsense mutations occur in the primary transcript, only the alternative transcript will function, which essentially transforms the pre-existing gene into a new gene. How *YAN/AltYAN* might have achieved physiological effects on oil body biogenesis and dehydration tolerance similar to the pre-existing *Hr* is perplexing. We speculate that *YAN* might have initially evolved as an alternative transcript of *Hr*, and their overlapping in coding regions led to functional linkage. Currently, dual-coding genes have not been seriously studied in plants. However, a manual inspection of *P. patens* chromosome 21, where *YAN/AltYAN* is located, indicated that at least three additional dual-coding genes (JGI IDs: *Pp3c21_16010*, *Pp3c21_7180*, and *Pp3c21_10530*) exist on the chromosome. It merits further detailed genome analyses and functional investigations to understand the role of this mechanism in the evolution of different plant lineages.

## Methods

**Phylogenetic analyses**. Multiple protein sequence alignments were performed using MUSCLE and refined manually. Gaps, ambiguously aligned sites, and sequences whose real identity could not be confirmed were removed from alignments. Phylogenetic analyses were performed with a maximum likelihood method using PhyML 3.1 and a distance method using neighbor of PHYLIP-3.695. The optimal model of amino acid substitution and rate heterogeneity was chosen based on the result of ModelGenerator.

**Plant materials and growth conditions**. *P. patens* "Gransden 2004" was used as wild-type plants in this study. Cultivation of *P. patens* plants followed Nishiyama et al.[63] Protonemata were grown on BCDAT medium containing 1 mM MgSO$_4$·7H$_2$O, 1.84 mM KH$_2$PO$_4$ (pH 6.5 adjusted with KOH), 10 mM KNO$_3$, 45 μM FeSO$_4$·7H$_2$O, 0.22 μM CuSO$_4$·5H$_2$O, 5 mM ammonium tartrate, 1 mM CaCl$_2$, and trace element solution (10 μM H$_3$BO$_3$, 0.23 μM CoCl$_2$·6H$_2$O, 0.1 μM Na$_2$MoO$_4$·2H$_2$O, 0.19 μM ZnSO$_4$·7H$_2$O, 2 μM MnCl$_2$·4H$_2$O, 0.17 μM KI, agar 0.8% (w/v)). Gametophores were grown on BCD medium (minus 5 mM ammonium tartrate). Plants were cultured under 25 °C with a 16-h light photoperiod per 8-h dark photoperiod, light intensity 80 μmol photons m$^{-2}$ s$^{-1}$. Sporophytes were induced by decreasing the day lengths and growth temperatures. Briefly, protonemata were grown under full light for 8 weeks at 25 °C, and then transferred to a short-day regime (8-h light photoperiod per 16-h dark photoperiod) at 15 °C. Sporophytes began to develop after about 4-week induction of low light and low temperatures. Mature sporangia were collected into a 1.5 ml centrifuge tube and sterilized using 10% sodium hypochlorite solution for 10 min. The sporangia were washed five times using sterilized water and then crushed in 1 ml sterilized water using the tip of a pipette, and mixed gently. The spore suspension was poured onto BCDAT medium and incubated at 25 °C under 16-h light photoperiod per 8-h dark photoperiod. The germinated spores were photographed after 4 days of incubation.

**PCR verification**. To verify the existence of the two transcripts *YAN* and *AltYAN* (and deactivation of the pre-existing *Hr*) in *P. patens*, the following primer pairs specific to the two transcripts (and the *Hr* locus) were used for PCR verification: (*YAN*-F: 5′-ATGTATAATATGTGCAGCCT-3′, *YAN*-R: 5′-TCAAACAGGTC GTCGTCCTC-3′), (*AltYAN*-F: 5′-ATGGCCGCAGTCGCATACAT-3′, *AltYAN*-R: 5′-TTAGCTGTGCCCAGCTGGAA-3′), and (*Hr*-F: 5′-AAGAGTTTGGAGGT GGGC-3′, *Hr*-R: 5′-CTAGTTGTCCTTGGGATACGG-3′).

**Dehydration assay**. Two-week old plants of *P. patens* were used to perform dehydration assays. The cellophane-layered plants were transferred to a sterile empty petri dish and dehydrated for 20 h in fume hood. The dehydrated plants were rehydrated for 1 h with sterilized water and then transferred onto standard medium. Plants were photographed and the chlorophyll content was measured after 2 weeks of recovery. Water loss percentage was calculated based on the dehydrated plant weight compared with the initial weight of the plants.

**Knockout and over-expression plasmid construction**. The vector pTN182 (DDBJ/EMBL accession number: AB267706, PHYSCObase) was used to create the targeting *YAN/AltYAN* construct. The upstream (817 bp) and the downstream (540 bp) of *YAN/AltYAN* were PCR-amplified using the following primer pairs: (upstream forward 5′-CCCGGTACCTACCAGTCATCGTCCT CCCCAAT-3′ (KpnI), upstream reverse 5′-CCCAAGCTTCCAAAACAAAACC ACCACCTCTT-3′ (HindIII)) and (downstream forward 5′-CCTCTAGAGCCG TATCCCAAGGACAACT-3′ (XbaI) and downstream reverse 5′-CCGGATCCA TGCCTCTTCTCAAACTGACCTAC-3′ (BamHI)). The two PCR fragments were cloned into pTN182 vector, respectively.

The vector pPOG1 (provided by Professor Mitsuyasu Hasebe) was used to create the over-expression construct. CDS of *AltYAN* was amplified using forward primer 5′-GCGGCCGCATGGCCGCAGTCGCAT-3′ (NotI) and reverse primer 5′-GTCGACTTAGCTGTGCCCAGCT-3′ (SalI). CDS of *YAN* was amplified using forward primer 5′-GCGGCCGCATGTATAATATGTGCA-3′ (NotI) and reverse primer 5′-GTCGACTCAAACAGGTCGTCGT-3′ (SalI). CDSs of *YAN* and *AltYAN* were, respectively, cloned into the pPOG1 vector.

**Protein subcellular localization**. The vector pM999 was used to create the *p35S: YAN-GFP* and *p35S:AltYAN-GFP* construct. CDSs of the two transcripts *YAN* and *AltYAN* were amplified respectively using *P. patens* complementary DNA (cDNA). Forward primer 5′-ggGGTACCATGTATAATATGTGCAGCCTCGG-3′ (KpnI) and reverse primer 5′-gcTCTAGAAACAGGTCGTCGTCCTCGAT-3′ (XbaI) were used to amplify *YAN* CDS. Forward primer 5′-ggGGTACCATGGCCGCAGTC GCATACAT-3′ (KpnI) and reverse primer 5′-gcTCTAGAGCTGTGCCCAGC TGGAAGAA-3′ (XbaI) were used to amplify *AltYAN* CDS. The amplified sequences were cloned, respectively, into pM999 vector to generate *p35S:CDS-GFP* fusion. Protoplasts of *P. patens* were isolated from 1-week old protonemata. The constructs were co-transformed respectively into protoplasts with mitochondria marker plasmid *p35S:PpCOX11-RFP*, peroxisome marker plasmid *p35S:PpPEX11-RFP*, Golgi marker plasmid *p35S:PpCTL1-RFP*, and oil body marker plasmid *p35S: PpOLEOSIN-RFP*. The transformed protoplasts were incubated in darkness at 25 °C for 16 h and then observed using an Olympus FV1000 confocal microscope. In addition, protonemata transfected by *p35S:YAN-GFP* were stained with Nile red dye (25 μg ml$^{-1}$ methanol) for 12 h, and their oil bodies were then observed. The emission filters were 500–530 nm for GFP, 580–620 nm for RFP, and 493–636 nm for Nile red.

**Protoplast transformation**. Preparation of protoplasts and genetic transformation was performed using the polyethylene glycol-mediated method[19]. Briefly, 1.6 × 10$^6$ protoplasts per ml were incubated with 15–30 μg linearized DNA. After polyethylene glycol treatment, the stable transformants were screened by three

successive cycles (1 week) of incubation on nonselective media and selective media (containing 20 μg ml$^{-1}$ G418 or 20 μg ml$^{-1}$ hygromycin). The stable transformants were confirmed through molecular characterization (see below).

**PCR characterization of knockout and over-expression mutants**. To confirm knockout mutants, we performed PCR genotyping using following primers: 1F (5′-CGTTGTCACTGAAGCGGGAAGG-3′) and 1R (5′-GAGCGGCGA-TACCGTAAAGCAC-3′) annealing to the selective cassette (p35S-nptII); 2F (5′-TGGATTCGGCCTTCAGGTGTC-3′) and 2R (5′-TGTCTGTTGTGCCCA GTCATAG-3′) annealing to upstream of *YAN* and the selective cassette, respectively; 3F (5′-GCTGACCGCTTCCTCGTGTCTT-3′) and 3R (5′-TTCCC GTACTATTCCTGCTTC-3′) annealing to the selective cassette and downstream of *YAN*, respectively (Supplementary Fig. 10).

To verify that *yan/altyan* mutants lacked functional *YAN* and *AltYAN*, we isolated the total RNA from both wild-type and mutant plants after 6 h of dehydration. Semi-quantitative RT-PCR analyses were performed as described below. Briefly, cDNA was synthesized using M-MLV reverse transcriptase (Promega M1701). Primers specific to the transcript *YAN* were designed, including a forward primer (5′-TCCCCGTCACTTTCTCCA-3′) and a reverse primer (5′-TGAACCATTATCCGGCCT-3′). Primers specific to *AltYAN* included a forward primer (5′-CATTGTTGCGTTCAAGCG-3′) and a reverse primer (5′-TC CACCACGTCACGATTGC-3′). *ACTIN* (XM_001775899) was used as control gene, with a forward primer (5′-CAGATCATGTTCGAGACGTTCAACG-3′) and a reverse primer (5′-CGAGCTGCTCCGTGCCGTGTCCAA-3′). Products of 30 PCR cycles were used for electrophoresis detection (Supplementary Fig. 10).

To confirm the *YAN* over-expression lines, we performed genotyping using CDS forward primer (5′-ATGTATAATATGTGCAGCCTCGG-3′) and reverse primer (5′-AACAGGTCGTCGTCCTCGAT-3′). To confirm the *AltYAN* over-expression lines, genomic DNA PCR was conducted using CDS forward primer (5′-ATGGCCGCAGTCGCATACAT-3′) and reverse primer (5′-GCTGTGCCCA GCTGGAAGAA-3′). Quantitative PCR was performed to confirm the gene over-expression fold in *YAN-OE* and *AltYAN-OE* lines. Forward primer (5′-AAGTTC CACAAGCAGGGCAA-3′) and reverse primer (5′-GTACAACACCGCTC CTCGG-3′) were used to detect the expression of *YAN*. Forward primer (5′-CC ATGTCGATCGCAAGAGGT-3′) and reverse primer (5′-TCATCCACCACGTCA CGATT-3′) were used to detect the expression of *AltYAN*.

**Real-time qRT-PCR**. Total RNA was extracted from *P. patens* clones using RNaEXTM total RNA isolation solution (Generay). cDNA was synthesized using M-MLV reverse transcriptase (Promega M1701). cDNA was diluted fivefolds and then used for qRT-PCR. All reactions were carried out on ABI7500 (Life Technologies) using TransStart Top Green qRT-PCR kit (Transgen Biotech) with three independent biological replicates. Housekeeping gene *ACTIN* (XM_001775899) was used as control to calculate the relative gene expression.

**Measurement of chlorophyll content**. Measurement of chlorophyll content followed Frank et al.[64] Briefly, 0.4 g 2-week recovered plants were homogenized using 1.5 ml of 80% (v/v) acetone. Samples were centrifuged at 12,000×$g$ for 5 min. Supernatants were used to measure absorbance at wavelengths of 645 and 663 nm for chlorophyll quantification. The supernatant was dried up in the initial tube at room temperature in order to measure the dry weight of each sample. The content of chlorophyll was calculated using the following formula: mg chlorophyll per g dry weight = [($A_{663}$) (0.00802) + ($A_{645}$) (0.0202)] × 1.5/dry weight.

**NO fluorescence**. Plant samples were dehydrated for 20 h, rehydrated 1 h, and then recovered 2 days. Recovered plants were incubated in 0.1 M phosphate buffer (pH 7.4) containing 20 μM of 4,5-diaminofluorescein diacetate (DAF-2DA) (Calbiochem) for 15 min in the dark. Incubated plants were then observed using an Olympus FV1000 confocal microscope with the 488 nm line of Argon laser. Each picture composed of three channels (NO fluorescence, chlorophyll fluorescence, and bright field). The intensity of NO fluorescence was measured using the software Image J.

**Detection of superoxide**. Superoxide accumulation was detected using the NBT (nitro-blue-tetrazolium) staining method[65]. Samples of *P. patens* were dehydrated for 20 h, rehydrated 1 h, and then recovered 2 days. The plants were then incubated in 10 mM potassium phosphate buffer containing 0.5 mg mL$^{-1}$ NBT, pH 7.6, for 2 h in the dark at 25 °C.

**Lipid analyses**. Total fatty acids were extracted from homogenized freeze-dried tissues using the method of Zhang et al.[66] Tripentadecanoic acid (15:0 TAG) was added to homogenized tissues to act as an internal standard. A proportion of the total fatty acid extract was subjected to transmethylation, and fatty acid methyl esters were quantified by gas chromatography-flame ionization detection with reference to the standard.

Ten-week old wild-type and *yan/altyan* plants on BCD medium were submerged for 30 s in a beaker of chloroform containing 10 μg octadecane as internal standards. Extracts were dried under a gentle stream of nitrogen and re-

suspended in chloroform. Wax compound identification was performed by gas chromatography-mass spectrometer (GC-MS) as described by Wang et al.[67] Individual waxes were quantified based on peak integration of the total ion chromatogram.

**Chromosome ploidy analyses**. Flow cytometric analyses were performed using Partec CyFlow SL (Partec GmbH, Münster, Germany), following the method of Bainard and Newmaster[68], with a blue solid-state laser tuned at 20 mW and operating at 488 nm. Fluorescence intensity at 630 nm was measured on a log scale, and forward scatter and side scatter data were recorded.

**Data availability**. Sequence data generated from this study have been deposited in GenBank, and their accession numbers (MG254881–MG254886) are cited in the text and online Supplementary Information. Other data supporting the findings of this study are available upon request to the corresponding author.

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

## Acknowledgements

We are grateful to Drs. M. Long, G. Sun, Y. Guo, and Q. Zhang for critically reading the manuscript and their suggestions for improvement. Thanks are also given to Drs. Y. He for generous help with protoplast transformation, L. Zhang, W. Ma, A. Liu, and X. Zhang for providing living plant materials, X. Hu and Y. Yang for providing lab facilities for related experiments, Y. Yang for help with NBT staining, J. Wu for providing pM999 vector, M. Hasebe for providing pTN182 and pPOG1 vectors. This study was supported in part by the CAS "Light of West China" Program to J.H., National Natural Science Foundation of China to Y.G. and H.S. (grant numbers 31500224 and 31590823), National Science Foundation to M.P.R. (grant number OIA-1355438), and CAS Pioneer Hundred Talents Program to J.Hu., and L.L.

## Author contributions

J.H. and Y.G. conceived and designed the study. Y.G., J.Z., Q.W., P.L., and J.H. performed experiments and data analyses. J.Hu., Z.Y., H.S., and L.L. contributed to experiments and data analyses. J.H., Y.G., and M.P.R. wrote the manuscript.

## Additional information

**Competing interests:** The authors declare no competing interests.

