## [Peer Review File · Nature Communications]

Reviewers' comments:

Reviewer #1 (Remarks to the Author):

This manuscript is in the field of evolution, especially conquest of land by plants. Based on their analysis that a Hr gene is absent from the Physcomitrella genome but encodes instead a dual gene YAN/AltYAN authors come to far-reaching conclusions, that I am not completely comfortable with: The Hr gene can be found in bacteria, fungi, some chlorophyte algae, Marchantia, Sphagnum and Selaginella but not in Physcomitrella or seed plants. Authors take this as evidence that the plant Hr gene was acquired from fungi, which, at least from my point of view would require more data than presented in this ms. However, this claim is a minor claim and may be toned down. More far reaching is the conclusion that Physcomitrella instead evolved the dual YAN/AltYAN gene, which authors can't find in other organisms. I can't see from this data that their claim to have discovered an important genetic mechanism for the adaptation to land is justified. And I am not sure if toning this statement down still would justify publication in Nature Communications.

Neither is Physcomitrella a direct progenitor of seed plants, nor is data presented that this gene conversion from Hr to YAN/AltYAN occurred recently, as the authors claim. At least information about expression of these genes should be presented from the 1 Kp project: Is this specific to Physcomitrella, to certain mosses, etc.?

The data itself is highly interesting, completely novel for plants, and may provide a textbook example for gene evolution. Therefore, it should be published.

Minor comments:

The introduction is not up to date: The reference for the statement that many adaptation mechanisms remain enigmatic is 10 years old; at least two recent publications in this context (stomata: Chater et al. 2016 Nature Plants; cuticle: Renault et al. 2017 Nature Communications) are not discussed or cited. Especially Renault et al. provide data on the Physcomitrella cuticle which seems to be important for the discussion in this ms.

YAN/AltYAN are annotated as splice variants (see gene IDs provided in the ms). Authors use descriptors like "dual-coding gene", "two overlapping genes" etc. which is not a very consistent nomenclature.

It is not clear from the text how many independent knockout and OE mutants, respectively, were created and subsequently analysed and if these showed consistent results. It should be at least three for each construct under normal circumstances, which may not be the case. However, this is hard to judge based on the data provided in the current version of the ms.

Are all these plants haploid? FCM analysis is mentioned in Materials and Methods but no statement about haploidy or diploidy of the transgenics included in the main text.

Standard deviations are given in figures but not in the main text. Are these SDs based on independent biological replicates, especially different mutants? In the current form it is hard to evaluate the statistic significance of the measurements.

Authors refer to Phytozome gene expression profiles but do not provide that data in the manuscript (or I can't find it). However, there are at least two excellent resources for Physcomitrella available (Hiss et al. 2014 Plant Journal, Ortiz-Ramirez et al. 2016 Molecular Plant) which should be queried. The use of the term "Gametophyte" seems to be wrong: Protonema is juvenile gametophyte, gametophore is adult gametophyte.

Are the mutants affected in sexual reproduction (= can they produce sporophytes)? If so, can spores germinate? Is F1 homogenous (as should be for haploid, self-fertilizing plants)?

Finally, authors thank "Dr. Yikun He for technical assistance with protoplast transformation" - I am pretty sure that he would not appreciate such an acknowledgement which is normally reserved for a technical assistant and not for a full professor.

Ralf Reski.

Reviewer #2 (Remarks to the Author):

Guan et al.

Gene refashioning through innovative shifting of reading frames in mosses

general: The authors demonstrate that the PpYAN locus evolved from an ancestral Hr locus, and that the newly evolved locus encodes two transcripts encoding proteins that function in dehydration tolerance. This represents the only Hr locus in *P. patens*. The two proteins appear to have different subcellular localisations with YAN in the oil bodies and AtYAN in the chloroplasts, but both function in oil body biogenesis, either directly or indirectly via regulation of fatty acid biosynthesis. The oil bodies function in an undefined way in dehydration tolerance, perhaps linked to wax formation on the cell surface.

The major finding in this study is co-option of a gene into a new gene with presumably a different molecular function. That it may be involved in a similar overall biological process could reflect that the regulatory regions were conserved, and thus the potential protein product(s) would be produced at the same time as the ancestral protein. Two important points not addresses in the manuscript are (1) whether the ancestral proteins do indeed function in dehydration tolerance — while their expression is correlated, there is no functional evidence available; and (2) if the novel gene architecture is specific to *Physcomitrella* or more broadly to mosses in general. Without this data, the discussion should be restricted to that pertaining directly to the physiology of *Physcomitrella* rather than land plant colonisation.

specific comments:

lines 45-47: some hypotheses on the origin of land plants suggest that some terrestrialisation occurred during the charophycean algal lineage.

line 57: better to say that they represent the earliest extant lineages

line 81: specify which resources were searched.

line 290: 'oil bodies' in mosses and liverworts are quite distinct entities, but the discussion here seems to equate them.

Figure S1: *Marchantia* is not a moss

Reviewer #3 (Remarks to the Author):

In the present article the authors describe a locus homologous to fungal hemerythrin that is present in some non-seed plants, but not in seed plants. While the locus looks like it might encode the hemerythrin e.g. in the liverwort *Marchantia polymorpha*, in the moss *Physcomitrella patens* it appears to be defunct due to a premature stop codon. Yet, two gene models were predicted for the locus, encoding two overlapping cDNAs. The authors analysed the localization and function of the encoded proteins and find that they are involved in lipid metabolism and appear to convey resistance to drought stress. They argue that these genes are "novel", and hypothesize that such a mode of gene acquisition (horizontal gene transfer - HGT- of the fungal gene) and modification (novel function based on an existing – defunct – locus) might be a hallmark of non-seed plants.

This story is potentially interesting and the analyses appear sound for the most part. However, I have some problems with methods, phrasing and also the hypotheses the authors put forward. I will first introduce my major concern and then list all author points.

Here is my major concern. There is evidence that the locus in question might have been acquired by

HGT, and lost in seed plants, and pseudogenized in *P. patens* (but see below for some concerns on methods and sampling). If indeed this locus evolved into encoding two proteins that are unrelated to the original one this is intriguing. But it is very far-fetched to speculate, based on this single evidence, that this might be a general phenomenon in non-seed plants, resp. early diverging plant lineages. I think the discussion and conclusions need to be toned down drastically.

Also, I think a more thorough analysis of the locus in *P. patens* is in order. I took a quick look in the genome browser and while the downstream region of the locus looks pretty normal, there is much evidence for transposable elements upstream of it. Also, is there detectable synteny between *P. patens*, *M. polymorpha* and *S. moellendorffii* surrounding the locus?

Other concerns:

Abstract: "...by the ancestral land plant" -> ancestral should be omitted, or rephrased to make clear that the reference is to the lineage that gave rise to extant *P. patens*. We are talking about an extant plant, not a living fossil.

Introduction, "de novo gene generation", I think genes should not be described as being "generated".

"Mosses, liverworts, and hornworts are the earliest extant land plants.": Needs to be rephrased, they are the extant representatives of the earliest divergences. The sentence after that requires citations – cuticle and stomata are known from bryophytes, so this should be detailed and maybe "often" is not very accurate.

Results, "identified homologs (E-value cutoff $1e-6$)": a simple e-value cutoff is not sufficient to determine homology. So either talk of hits or use a more appropriate cutoff. The blast matrix should be mentioned. It would be good to show a taxonomic representation of the hits.

"No homologs were found in seed plants and charophyte green algae": the absence in charophytes cannot be taken as proof that the gene is not present in these organisms, since there is very limited genomic information available on this grade. This impacts on the evolutionary scenario, as it cannot be ruled out that the locus was already acquired by charophytes rather than in the earliest land plants.

"animal and other eukaryotic Hrs differ significantly from fungal": fungi are eukaryotes as well.

The gene copy is referred to as "novel" e.g. in abstract and introduction; how old is it?

"No homolog was identified, suggesting that this dual-coding gene evolved recently from the pre-existing Hr locus." No, it can be an orphan specific to the *P. patens* lineage and could potentially be very old. They even acknowledge the high number of orphans in the discussion.

Also, if the YAN cds overlaps to a large extent with the Hr gene, why can't homology to the HHE domain be detected?

It would be good if the age of the genes, or at least the age of the pseudogenic (Hr) part, could be determined. E.g. look at the number of substitutions found in the part after the stop (potentially not under selective pressure) and compare with the homologous loci in other organisms.

"We here refer to the longer transcript Pp3c21_19720V3.2 as YAN": Why? What's the meaning of YAN?

"Both transcripts were found to be strongly induced by dehydration and rehydration.": please provide evidence.

"in Figure 2B, the transcription levels of YAN and AltYAN increased oscillatorily during dehydration and

rehydration treatments; their expression levels, on the other hand, were highest at 2 hours after rehydration and then decreased continuously”: The oscillatory expression seems only be true for YAN, and the highest expression for AltYAN is 36h after dehydration.

Reference 22: There are many more papers that deal with *P. patens* dehydration (resp. induced dessication) tolerance, not all of them in line with regard to their conclusions. It does thus not suffice to cite only one, relatively old, paper.

“four-week old gametophytes” should probably be “four-week old gametophores”? State of plant development in Fig. 4 cannot be assessed.

Discussion, 1st paragraph: not true. The authors are directed e.g. to Neme and Tautz 2016 eLife 5:e09977. Also, there are e.g. papers on how genes transferred to from the plastid to the nuclear genome acquire targeting signals. And the authors themselves cite a few other studies later on.

“Wax crystals on plant surface are the main barrier to water loss, UV-radiation, insect herbivory and pathogen infection.”: Requires citations.

“Using land plant Hrs sequences as queries, our search of the NCBI EST database identified most similar hits to dehydration and rehydration transcripts in two resurrection plants, the twisted moss *Syntrichia ruralis* and the spikemoss *Selaginella lepidophylla* (75, 76).”: please provide evidence.

Conclusions: “Early land plants”: no, *P. patens* is an extant plant. Please see above wrt the use of “novel” and “new”, and “species-specific benefits to *P. patens*” might not be correct, since we do not know about the taxonomic distribution of the genes.

Methods: The dehydration assay lack details on how much water was lost, and on the environmental conditions under which the dehydration took place. Repetition would be impossible for a third party. The ploidy measurements do not seem to be mentioned in results?

Supplementary Fig. 1: Please add species names to the upper part. Lower part: *Marchantia* is not a moss. Why are many nodes lacking support values? Sub-division of “Fungi” would be helpful to evaluate the tree. Please provide the alignment on which the tree is based. Which was the optimal model used?

Supplementary Fig. 2: Did the authors check existing RNA-seq evidence? The primer hops could be misleading and no full length cDNA was cloned and sequenced.

Supplementary Fig. 5: The term colony should be avoided when talking about *P. patens* plants.

Supplementary Fig. 7: Which of the OE lines mentioned in Fig. S6 was used? Same in Figure 4 and Fig. S8.

Major changes in the revision

We are truly grateful to Dr. Ralf Reski and two other reviewers for their insightful comments and suggestions. Revision has been made based on these comments. All new sequences generated from this revision have been submitted to GenBank, and their accession numbers are included in the manuscript. Some of the major changes are outlined in the following:

1. *Distributions of the Hr gene in plants.* We have made some comprehensive searches of nr and 1KP databases using different queries. It is now clear that *Hr* is present in liverworts and mosses and in the lycophytes *Selaginella* and *Isoetes*. Although *Hr* was identified in two hornworts from 1KP and also confirmed by our own RT-PCR reactions and sequencing, we cannot definitely conclude whether the amplified *Hr* gene was indeed from liverworts or symbiotic fungi. Furthermore, although *Hr* sequences were identified in angiosperms and other seedless vascular plants (ferns and other lycophytes) from 1KP, they are closely related to separate fungal homologs and therefore most likely due to contamination. We also discuss the issue of potential false positives for the algal *Hr* sequences in Supplementary Note 1 and Table S3.

2. *Distribution of the dual-coding gene YAN/AltYAN.* In addition to a comprehensive search of 1KP, we performed our own RT-PCR reactions and sequencing on a species of *Physcomitrium*, a genus most closely related to *Physcomitrella*. We show that both *Hr* and *YAN/AltYAN* exist in the species and conclude that *YAN/AltYAN* most likely evolved in the *Physcomitrium-Physcomitrella* species complex.

3. *The function of plant Hr gene.* Currently there is no experimental data for *Hr* in plants. However, our search of 1KP identified a gene fusion event between oleosin and *Hr* in two *Isoetes* species and a liverwort (possible contamination). We also confirmed this fusion event independently through our own RT-PCR reaction and sequencing using another species (*Isoetes yunguiensis*). Given the common belief that gene fusion is indicative of functional linkage, we conclude that *Hr* most likely is functionally related to oil body biogenesis. This suggestion is consistent with the role of oil bodies in dehydration resistance and the fact that *Hr* is induced by both dehydration and rehydration treatments in two resurrection plants.

4. *Frequency of dual-coding genes.* Given the limited time during this revision, it is difficult to perform a comprehensive genome analysis of dual-coding genes in *Physcomitrella*. We, however, performed manual inspection of *P. patens* chromosome 21, where *YAN/AltYAN* is located. The results showed that at least three additional dual-coding genes exist on chromosome 21. We briefly mentioned this observation in Discussion and suggested that further comprehensive analyses are needed.

5. *Dual-coding (overlapping) and origin of new genes.* We speculate that, if nonsense mutations occur in the primary transcript, the alternative transcript will essentially become a new gene. In our opinion, this could be a very interesting mechanism for gene origination. This possibility is briefly discussed in Discussion.

6. *Discussion in general.* Discussion has been modified accordingly based on the new data outlined above as well as the comments and suggestions of the reviewers. Because of the distribution of *Hr* primarily in basal land plants as well as its possible role in oil body biogenesis and dehydration resistance, it is impossible to avoid the issue of early evolution of land plants. The discussion, nevertheless, has been kept as minimal as possible in this revision.

Responses to comments by reviewer 1

1. This manuscript is in the field of evolution, especially conquest of land by plants. Based on their analysis that a *Hr* gene is absent from the *Physcomitrella* genome but encodes instead a dual gene *YAN/AltYAN* authors come to far-reaching conclusions, that I am not completely comfortable with: The *Hr* gene can be found in bacteria, fungi, some chlorophyte algae, *Marchantia*, *Sphagnum* and *Selaginella* but not in *Physcomitrella* or seed plants. Authors take this

as evidence that the plant *Hr* gene was acquired from fungi, which, at least from my point of view would require more data than presented in this ms.

The origin of plant *Hr* gene is determined based on overall evidence (distribution, phylogeny, and sequence similarity). The tree topology is not perfect, but this is understandable especially given the short length of the conserved HHE domain (about 120 aa). However, we do indicate other possible scenarios in the Discussion section (lines 274-275). We think it is important to discuss these alternative scenarios.

As a side note, our analyses also show that many angiosperm and seedless vascular *Hr* sequences (and possibly the algal ones) in 1KP are most likely contaminated by fungi. Although this cannot be used as direct evidence to support the suggestion of fungal origin of *Hr* in basal land plants, it does point to the intimate association between fungi and plants as well as the possibility of HGT from fungi to plants. The fungal origin has also been concluded for the tardigrade *Hr* (Hashimoto et al. 2016. Nature Communications 7:12808), though such a conclusion cannot be drawn based on the tree topology (tardigrade *Hr* protein sequences do share much higher percent identities with fungal homologs). Another interesting note is that the recently published *Marchantia* genome paper also suggested many transferred genes from fungi (“KOGs found in *M. polymorpha*, but not in other land plants, are often homologous with fungal genes or related to mobile elements, suggestive of horizontal gene transfer”) (Bowman et al. 2017. Cell 171:287-304).

2. More far reaching is the conclusion that *Physcomitrella* instead evolved the dual YAN/AltYAN gene, which authors can't find in other organisms. I can't see from this data that their claim to have discovered an important genetic mechanism for the adaptation to land is justified.

In our original submission, we suggested that if gene refashioning through innovative frameshifting and re-use of pre-existing regulatory regions were common, it might rapidly lead to new genes and facilitate the adaptation of early land plants. We agree with Dr. Reski here that this suggestion is speculative and needs more evidence. In this revision, we briefly discussed the existence of other dual-coding genes in the *Physcomitrella* genome. Given the overall high quality of gene annotation for *Physcomitrella*, these genes are most likely real. We suggest that detailed analyses are needed on this issue in the Discussion section (lines 379-383).

2. Neither is *Physcomitrella* a direct progenitor of seed plants, nor is data presented that this gene conversion from *Hr* to YAN/AltYAN occurred recently, as the authors claim. At least information about expression of these genes should be presented from the 1 Kp project: Is this specific to *Physcomitrella*, to certain mosses, etc.?

We agree with Dr. Reski that *Physcomitrella* is not the direct ancestor of seed plants. As we indicate in the Introduction section, *Physcomitrella* is merely used in this study (and in many others) as a model to study the biology of early land plants (lines 52-53).

In this revision, we performed a comprehensive search of the 1KP database, which covers 32 liverworts, 16 hornworts and 43 mosses, including an unspecified species of *Physcomitrium*. The search using YAN protein sequence as query provided many hits, but the vast majority of them (including three *Physcomitrium* sequences) were either very short (<50 aa) or had premature stop codons. The search of 1KP using AltYAN protein sequence as query provided no hits to any taxa outside *Physcomitrium*. Only a single hit to *Physcomitrium* (YEPO-2062682) was found. This single hit covered the 5'UTR and first exon of *AltYAN*, which is upstream of the *HHE* locus. Therefore, although whether *YAN* and *AltYAN* exist in *Physcomitrium sp.* cannot be answered by our search of 1KP, there is no evidence for the two transcripts, at least for *AltYAN*, in other taxa outside *Physcomitrium*. On the other hand, our own RT-PCR reactions and sequencing on another unspecified species of *Physcomitrium* identified both *Hr* and *YAN/AltYAN*, suggesting that *YAN/AltYAN* most likely evolved in the *Physcomitrium-Physcomitrella* species complex. We

provide this information in the revised manuscript (lines 151-162 and Supplementary Figures S5-S6).

3. The data itself is highly interesting, completely novel for plants, and may provide a textbook example for gene evolution. Therefore, it should be published.

Thanks. As we indicated above, other dual-coding genes also exist in *Physcomitrella* according to our manual inspection of the genome data. We suggest in the Discussion section that this issue merits detailed genome analyses and functional investigations (lines 379-383)

4. The introduction is not up to date: The reference for the statement that many adaptation mechanisms remain enigmatic is 10 years old; at least two recent publications in this context (stomata: Chater et al. 2016 Nature Plants; cuticle: Renault et al. 2017 Nature Communications) are not discussed or cited. Especially Renault et al. provide data on the *Physcomitrella* cuticle which seems to be important for the discussion in this ms. *YAN/AltYAN* are annotated as splice variants (see gene IDs provided in the ms). Authors use descriptors like "dual-coding gene", "two overlapping genes" etc. which is not a very consistent nomenclature.

We have revised the Introduction section and cited several recent papers to reflect the progress in the field. Changes were also made in the text to indicate *YAN* and *AltYAN* are two transcripts to avoid confusion (lines 137-138). The term dual-coding refers to the phenomenon that the same coding region encodes distinct proteins. This term has been used by multiple studies on animals, and it is therefore also used in this manuscript to maintain consistency in literature.

5. It is not clear from the text how many independent knockout and OE mutants, respectively, were created and subsequently analysed and if these showed consistent results. It should be at least three for each construct under normal circumstances, which may not be the case. However, this is hard to judge based on the data provided in the current version of the ms.

We created five *yan/altyan* mutants, three *YAN-OE* lines, and four *AltYAN-OE* lines. Subsequent analyses showed consistent results. We provided this information in this revised manuscript (lines 192-193, 215-216)

6. Are all these plants haploid? FCM analysis is mentioned in Materials and Methods but no statement about haploidy or diploidy of the transgenics included in the main text. Standard deviations are given in figures but not in the main text. Are these SDs based on independent biological replicates, especially different mutants? In the current form it is hard to evaluate the statistic significance of the measurements.

We have added a sentence in this revision, indicating these knockout mutants were all haploids based on flow cytometry analyses (lines 192-193).

Standard deviations were calculated based on three independent biological replicates on each mutant. It is stated in all relevant figure legends.

7. Authors refer to Phytozome gene expression profiles but do not provide that data in the manuscript (or I can't find it). However, there are at least two excellent resources for *Physcomitrella* available (Hiss et al. 2014 Plant Journal, Ortiz-Ramirez et al. 2016 Molecular Plant) which should be queried.

The original gene expression data from Phytozome have been added to the Supplementary Materials (Supplementary Figure S7).

We also thank Dr. Reski for referring to other resources. We checked the GENEVESTIGATOR site (https://genevestigator.com/gv/doc/intro_plant.jsp) and the *Physcomitrella* ePB browser (http://bar.utoronto.ca/efp_physcomitrella/cgi-bin/efpWeb.cgi). Although both appear to be wonderful resources, they only accept gene identifiers from cosmos

V1.6 or earlier versions and do not provide features like BLAST search. Because *YAN/AltYAN* were annotated by later releases of cosmos (V3.1 and V3.2), they are not found in the two sites.

8. The use of the term "Gametophyte" seems to be wrong: Protonema is juvenile gametophyte, gametophore is adult gametophyte.

Corrected.

9. Are the mutants affected in sexual reproduction (= can they produce sporophytes)? If so, can spores germinate? Is F1 homogenous (as should be for haploid, self-fertilizing plants)?

We performed sporophyte induction experiments during this revision. Plants were grown for 8 weeks at 25°C before they were transferred to a short-day regime at 15°C. In both wild-type and *yan/altyan* plants, sporophytes became visible after 4 weeks of induction (lines 234-238, Supplementary Fig. S13). The sporophytes development of *AltYAN-OE* and *YAN-OE* was slightly delayed compared to the wild type. At present, the sporophytes are not completely mature, and whether the spores are viable remains to be investigated.

10. Finally, authors thank "Dr. Yikun He for technical assistance with protoplast transformation" - I am pretty sure that he would not appreciate such an acknowledgement which is normally reserved for a technical assistant and not for a full professor.

Sentence has been rephrased into "for his generous help with protoplast transformation" (line 558).

Responses to comments by reviewer 2

1. The major finding in this study is co-option of a gene into a new gene with presumably a different molecular function. That it may be involved in a similar overall biological process could reflect that the regulatory regions were conserved, and thus the potential protein product(s) would be produced at the same time as the ancestral protein. Two important points not addresses in the manuscript are (1) whether the ancestral proteins do indeed function in dehydration tolerance — while their expression is correlated, there is no functional evidence available; and (2) if the novel gene architecture is specific to *Physcomitrella* or more broadly to mosses in general. Without this data, the discussion should be restricted to that pertaining directly to the physiology of *Physcomitrella* rather than land plant colonisation.

We thank this reviewer for pointing out issues related to the function of the ancestral *Hr* gene and the distribution of the dual-coding *YAN/AltYAN*. The functional information of *Hr* in plants is indeed important for our discussion. Unfortunately experimental data related to *Hr* are not available. Nevertheless, indirect evidence suggests that *Hr* is indeed related to dehydration resistance. A critical piece of evidence in this regard is the gene fusion event between *Hr* and oleosin in the lycophytes *Isoetes tegetiformans*, *I. yunguiensis* and *I. sp.* (also in the liverwort *Treubia lacunosa* according to 1KP, but this may be due to contamination) (Supplementary Figure S1). It has long been accepted that genes linked by fusion events are usually functionally associated (Marcotte et al. 1999. Nature 402:83-86; Enright et al. 1999. Nature 402:86-90; Yanai et al. 2001. PNAS 98:7840-7945). Gene fusion data are also the basis for the popular STRING protein-protein interaction database (Mering et al. 2003. Nucleic Acids Res. 31:258-261). Oleosin is an integral component of oil bodies and has been thought to be an important feature of dehydration tolerance in *Physcomitrella* and other nonvascular plants (Huang et al. 2009. Plant Physiology 150:1192-1203). The functional linkage between *Hr* and oleosin therefore suggests a role of the ancestral *Hr* in plant oil body biogenesis and dehydration resistance. This suggestion is consistent with the fact that *Hr* is induced by both dehydration and rehydration in two

resurrection plants, the twisted moss *Syntrichia ruralis* and the spikemoss *Selaginella lepidophylla*, and that the two processes are mediated by the same sets of genes (Hiss et al. 2014. *Plant Journal* 79:530-539).

In terms of the distribution of *YAN/AltYAN*, we performed some comprehensive searches of the 1KP database, which contains transcriptomic data of over 90 nonvascular plants, including an unspecified species of *Physcomitrium* (a paraphyletic group that gave rise to *Physcomitrella*). We also conducted our own RT-PCR reactions and sequencing on another species of *Physcomitrium*. We were able to determine that the dual-coding gene *YAN/AltYAN* most likely evolved in the *Physcomitrium-Physcomitrella* species complex (lines 151-162).

Following the suggestion of this reviewer, we significantly modified the Discussion section. Evidence for the possible role of the ancestral *Hr* in oil body biogenesis and dehydration tolerance has been provided in this revision (lines 342-353). The discussion on the relationship to plant colonization of land has been kept as minimal as possible. Furthermore, we discussed very briefly the existence of other dual-coding genes in *Physcomitrella* in the Discussion section, and suggested that detailed studies on these genes are needed (lines 379-383).

2. lines 45-47: some hypotheses on the origin of land plants suggest that some terrestrialisation occurred during the charophycean algal lineage.

It is true that some charophytes, for instance *Klebsormidium*, *Interfilum* and their close relatives (Klebsormidiophyceae), are commonly discussed as inhabitants of terrestrial habitats. In fact, terrestrial forms are common in certain groups of both chlorophytes and charophytes; they are found in at least four of the six charophyte classes, including Chlorokybophyceae, Klebsormidiophyceae, Zygnematophyceae and Coleochaetophyceae (Holzinger, A. and U. Karsten. 2013. *Frontiers in Plant Science* 4: article 327). However, the term land plants is usually reserved for the group embryophytes. Furthermore, although land plants evolved from within charophytes, they are not particularly related to terrestrial charophytes based on the current understanding of green plant phylogeny (Leliaert et al. 2012. *Critical Reviews in Plant Sciences* 31:1-46; Zhong et al. 2015. *Evolutionary Bioinformatics* 11:137-141).

To avoid confusion, we indicate in this revision specifically that land plants in this manuscript are equivalent to embryophytes (line 38).

3. line 57: better to say that they represent the earliest extant lineages

Thanks. This sentence has been changed to “extant representatives of the earliest land plant lineages” (lines 49-50).

4. line 81: specify which resources were searched.

The entire section has been rewritten in this revision (lines 67-107). We indicate in this revised manuscript that nr and 1KP were searched using different queries. We did search other resources such as NCBI dbEST and JGI databases, however.

5. line 290: 'oil bodies' in mosses and liverworts are quite distinct entities, but the discussion here seems to equate them.

Thanks again for this comment. We have rephrased the sentence into the following: “While the oil bodies in liverworts and mosses might differ fundamentally in structure and development,” (lines 313-314).

6. Figure S1: *Marchantia* is not a moss

Correction has been made.

Responses to comments by reviewer 3

We thank this reviewer tremendously for his/her meticulous comments and suggestions. These suggestions were carefully considered, and changes have been made accordingly. Please see the following for details.

1. There is evidence that the locus in question might have been acquired by HGT, and lost in seed plants, and pseudogenized in *P. patens* (but see below for some concerns on methods and sampling). If indeed this locus evolved into encoding two proteins that are unrelated to the original one, this is intriguing. But it is very far-fetched to speculate, based on this single evidence, that this might be a general phenomenon in non-seed plants, resp. early diverging plant lineages. I think the discussion and conclusions need to be toned down drastically.

We agree with this reviewer that a general conclusion should not be drawn from a single example. In the Conclusions of our original submission, we wrote “Presumably, if innovative shifting of reading frames is common, some old genes may be rapidly refashioned into novel genes”. We recognize that this is very speculative, and whether this phenomenon is indeed important depends on its scope of occurrence. Following the comments of this and the two other reviewers, we performed additional investigations on dual-coding genes in *Physcomitrella* genome. Our manual inspections of chromosome 21 indicated that at least three additional dual-coding genes exist on the chromosome. We have added this information to the Discussion and suggested that more detailed analyses are needed on this issue (lines 379-383).

Following the comments by this reviewer, the discussion has been significantly modified.

2. Also, I think a more thorough analysis of the locus in *P. patens* is in order. I took a quick look in the genome browser and while the downstream region of the locus looks pretty normal, there is much evidence for transposable elements upstream of it. Also, is there detectable synteny between *P. patens*, *M. polymorpha* and *S. moellendorffii* surrounding the locus?

We performed additional analyses following the suggestion of this reviewer. Ten protein-coding genes upstream and downstream of *YAN/AltYAN* and corresponding regions in *M. polymorpha* and *S. fallax* were obtained to assess whether there is clear synteny between the three genomes. Genes surrounding *YAN/AltYAN* in *Physcomitrella* and *Hr* in *M. polymorpha* and *S. fallax* are mostly not homologous (see figure below). This is not entirely unexpected given the distant relationships between these taxa (they do after all belong to two different major lineages, i.e., liverworts and mosses. *Sphagnum* is considered basal to other mosses, and it is not closely related to *Physcomitrella*)

10-kb genomic regions upstream and downstream of the *YAN/AltYAN* gene were also used to search putative transposable elements (TEs). RepeatMasker search identified a total of nine TEs. All the detected TEs belong to long terminal repeat (LTR) retrotransposons. In addition, all these TEs are located in the upstream of the *YAN/AltYAN* locus. Results are shown in the table below.

No.	Score	Repeat-matching begin	Repeat end	Length	Repeat class/family
1	1185	-10149	-9996	154	LTR/Gypsy
2	1185	-9937	-9782	156	LTR/Gypsy
3	13360	-9781	-7532	2250	LTR/Gypsy
4	3270	-7482	-6973	510	LTR/Gypsy
5	5764	-6972	-5599	1374	LTR/Gypsy
6	1073	-5587	-5287	301	LTR/Gypsy
7	1441	-5201	-4865	337	LTR/Gypsy
8	1424	-4461	-4009	453	LTR/Copia
9	12460	-4007	-1519	2489	LTR/Copia

We chose not to include the above information in the text and Supplementary Materials, since we were unsure whether TEs would affect the origin of *YAN/AltYAN*. Although TEs occasionally are linked to new genes and frequently transferred between species, we didn't detect a clear link between TEs and *YAN/AltYAN* in our current data.

3. Abstract: "...by the ancestral land plant" -> ancestral should be omitted, or rephrased to make clear that the reference is to the lineage that gave rise to extant *P. patens*. We are talking about an extant plant, not a living fossil.

We have rephrased the sentence into "was related to fungal and tardigrade homologs" in view of new data (line 27).

4. Introduction, "de novo gene generation", I think genes should not be described as being "generated".

Changed to "de novo gene origination" (line 46).

5. "Mosses, liverworts, and hornworts are the earliest extant land plants.": Needs to be rephrased, they are the extant representatives of the earliest divergences. The sentence after that requires citations – cuticle and stomata are known from bryophytes, so this should be detailed and maybe "often" is not very accurate.

This sentence has been changed to "extant representatives of the earliest land plant lineages" (lines 49-50).

We also added a citation to the sentence. We do try to cite important papers related to topics discussed in the manuscript. However, because of the limited number of citations allowed by the journal (max 70 citations), it is difficult to cite relevant papers in many cases. We also deleted "often" from the paragraph here.

6. Results, "identified homologs (E-value cutoff $1e-6$): a simple e-value cutoff is not sufficient to determine homology. So either talk of hits or use a more appropriate cutoff. The blast matrix should be mentioned. It would be good to show a taxonomic representation of the hits.

We appreciate the comments by this reviewer. In our opinion, the entire idea of BLAST search centers on the assessment of sequence homology (relatedness versus un-relatedness). While we agree that a simple e-value cutoff may not be sufficient to determine sequence homology, it does provide a measurement of random background noise or confidence of homology assessment. If homology of the identified sequences (or hits) cannot be assumed, we will lose the basis for subsequent analyses, including phylogenetic reconstruction. Ideally, one could decrease the e-value cutoff to increase the stringency of the search. However, because the alignment score, thus e-value, is determined by both sequence similarity and length, it is not particularly useful to employ a lower e-value cutoff in our case, given the fact that the conserved HHE domain is only about 120 aa.

Following the suggestion of this reviewer, we indicate in this revision that the default protein substitution scoring matrix, BLOSUM62, was used in the search (line 72). Furthermore, description lines, including species information and identifiers, of two searches of 1KP using *Sphagnum* and *Gonium* sequences as queries are now included in Supplementary Tables S1 and S2. The taxonomic report for BLAST search of NCBI nr database using a charophyte sequence identified in 1KP as query is also included in Supplementary Table S3.

7. “No homologs were found in seed plants and charophyte green algae”: the absence in charophytes cannot be taken as proof that the gene is not present in these organisms, since there is very limited genomic information available on this grade. This impacts on the evolutionary scenario, as it cannot be ruled out that the locus was already acquired by charophytes rather than in the earliest land plants.

We agree with the reviewer here in principle. This sentence merely states the fact that no *Hr* homologs were identified in charophytes during our database search. In practice, we can only present the data objectively, though the data might be interpreted differently. Although we did suggest that *Hr* was likely transferred from fungi to the ancestral land plant, this suggestion was not solely based on the lack of identifiable *Hr* homologs in charophytes; it was based on the overall evidence of taxonomic distribution, sequence similarity and phylogeny. Further, we also cautioned in our original submission that other scenarios, including differential loss, cannot be confidently excluded.

In this revision, we indicate that hits to charophytes were indeed found in 1KP, but these charophyte hits usually shared higher sequence percent identities with fungal and/or bacterial sequences. We also indicate that search of 1KP using *Sphagnum* *Hr* provided no hits to any green algal sequences (though 186 green algal species are covered in the database). We discuss this and related issues (including potential contamination) in the Supplementary Note 1.

8. “animal and other eukaryotic Hrs differ significantly from fungal”: fungi are eukaryotes as well.

This sentence has been removed from the manuscript.

9. The gene copy is referred to as “novel” e.g. in abstract and introduction; how old is it? “No homolog was identified, suggesting that this dual-coding gene evolved recently from the pre-existing *Hr* locus.” No, it can be an orphan specific to the *P. patens* lineage and could potentially be very old. They even acknowledge the high number of orphans in the discussion.

We again thank the reviewer for this comment. In this revision, we provide evidence that *YAN/AltYAN* most likely evolved in the *Physcomitrium-Physcomitrella* species complex (lines 151-162). We also removed the term “novel” from the manuscript.

10. Also, if the *YAN* cds overlaps to a large extent with the *Hr* gene, why can’t homology to the HHE domain be detected?

Sorry for the confusion here. No homology can be detected only at the level of protein sequences (*YAN* and *Hr* encode different proteins). Of course, they are homologous at the nucleotide sequence level since they share a portion of their sequences.

11. It would be good if the age of the genes, or at least the age of the pseudogenic (*Hr*) part, could be determined. E.g. look at the number of substitutions found in the part after the stop (potentially not under selective pressure) and compare with the homologous loci in other organisms.

We concur with this reviewer that it is interesting to date the *Hr* gene loss event. This, however, requires not only a clear understanding of the distribution of loss, but also data availability from the close relative of the taxon where the loss initially occurred. Currently we still do not have a complete picture of the gene loss distribution. We feel that the dating would

not be as meaningful as we would hope if a random *Physcomitrium* species is selected for comparison. We plan to continue working on *Hr*, and the dating will be part of our future work.

12. “We here refer to the longer transcript Pp3c21_19720V3.2 as YAN”: Why? What’s the meaning of YAN?

We indicate in this revision that YAN is a popular name of newborns in Chinese, meaning “stunning and fascinating” (lines 139-140). We chose this word because the dual-coding gene was new compared to the pre-existing *Hr*, and we thought it was intriguing.

13. “Both transcripts were found to be strongly induced by dehydration and rehydration.”: please provide evidence.

Thanks for this suggestion. We have added the original expression data from Phytozome to Supplementary Materials (Supplementary Figure S7).

14. “in Figure 2B, the transcription levels of YAN and AltYAN increased oscillatorily during dehydration and rehydration treatments; their expression levels, on the other hand, were highest at 2 hours after rehydration and then decreased continuously”: The oscillatory expression seems only be true for YAN, and the highest expression for AltYAN is 36h after dehydration.

This sentence has been rephrased to “the transcription levels of *YAN* and *AltYAN* increased in a fluctuating manner during dehydration and decreased gradually after rehydration” (lines 183-185).

15. Reference 22: There are many more papers that deal with *P. patens* dehydration (resp. induced dessication) tolerance, not all of them in line with regard to their conclusions. It does thus not suffice to cite only one, relatively old, paper.

We apologize that many relevant papers are not cited. The journal policy only allows a maximum of 70 references for an article. The Frank et al. 2005 was cited for its specific data about water loss tolerance in *Physcomitrella*.

16. “four-week old gametophytes” should probably be “four-week old gametophores”? State of plant development in Fig. 4 cannot be assessed.

It has been changed to “four-week old gametophores” (line 224). We also made corrections throughout the entire manuscript.

We have added a detailed figure about the plant development in Figure 4c showing WT had developed into gametophores, but *YAN-OE* and *AltYAN-OE* exhibited delayed development.

17. Discussion, 1st paragraph: not true. The authors are directed e.g. to Neme and Tautz 2016 eLife 5:e09977. Also, there are e.g. papers on how genes transferred to from the plastid to the nuclear genome acquire targeting signals. And the authors themselves cite a few other studies later on.

Thanks for pointing out the references. Neme and Tautz 2016 paper is indeed interesting and has been cited in this revision.

The transfer from plastids to the nucleus is somewhat different in our opinion, since these plastid genes were already within the cell before the relocation; they are basically the same genes and proteins in most cases. Furthermore, such intracellular gene transfer will not affect phenotypes in general. For instance, relocation of photosynthetic genes from plastids to the nucleus will not significantly affect photosynthesis. Organisms with temporal cyanobacterial endosymbionts or permanent plastid organelles (e.g., *Elysia chlorotica*, *Paulinella chromatophora*, other primary or secondary photosynthetic eukaryotes), with or without intracellular gene transfer, can perform photosynthesis.

18. “Wax crystals on plant surface are the main barrier to water loss, UV-radiation, insect herbivory and pathogen infection.”: Requires citations.

A citation has been added in this revised manuscript (line 320).

19. “Using land plant Hrs sequences as queries, our search of the NCBI EST database identified most similar hits to dehydration and rehydration transcripts in two resurrection plants, the twisted moss *Syntrichia ruralis* and the spikemoss *Selaginella lepidophylla* (75, 76).”: please provide evidence.

We have added the original search results to the Supplementary Materials, indicating that the transcripts were generated from dehydration and rehydration libraries (Supplementary Figure S15).

20. Conclusions: “Early land plants”: no, *P. patens* is an extant plant.

We have added a sentence to indicate that *P. patens* is used as “a model organism to study the biology of early land plants” (line 367).

21. Please see above wrt the use of “novel” and “new”, and “species-specific benefits to *P. patens*” might not be correct, since we do not know about the taxonomic distribution of the genes.

Thanks for this suggestion. Discussion has been revised to reflect our new findings. We have provided evidence that *YAN/AltYAN* most likely evolved in the *Physcomitrium-Physcomitrella* species complex. Most of above terms have also been removed from this revised manuscript. We also use “lineage-specific” whenever possible.

22. Methods: The dehydration assay lack details on how much water was lost, and on the environmental conditions under which the dehydration took place. Repetition would be impossible for a third party.

More details have been added to the dehydration assay section. This section now reads: “Two-week old plants of *P. patens* were used to perform dehydration assays. The cellophane-layered plants were transferred to a sterile empty petri dish and dehydrated for 20 hours in fume hood. The dehydrated plants were rehydrated for 1 hour with sterilized water and then transferred onto standard medium. Plants were photographed and the chlorophyll content was measured after two weeks of recovery. Water loss percentage was calculated based on the dehydrated plant weight compared with the initial weight of the plants” (lines 418-423).

23. The ploidy measurements do not seem to be mentioned in results?

Thanks for catching this omission. This information has been added to the manuscript. We indicate “they were all haploids based on flow cytometry analyses (lines 192-193)”.

24. Supplementary Fig. 1: Please add species names to the upper part. Lower part: *Marchantia* is not a moss. Why are many nodes lacking support values? Sub-division of “Fungi” would be helpful to evaluate the tree. Please provide the alignment on which the tree is based. Which was the optimal model used?

Thanks. Analyses were re-done by adding new sequences. Following the suggestion of this reviewer, information about subdivision has been added to the tree and errors have been corrected. Nodes without supporting values show values lower than 50% from both analyses. This information and protein substitution model have also been added to the figure legend (Supplementary Figure S2).

25. Supplementary Fig. 2: Did the authors check existing RNA-seq evidence? The primer hops could be misleading and no full length cDNA was cloned and sequenced.

We did check the RNA-Seq coverage as well as PASA assembled and aligned EST/cDNA evidence for the gene in Phytozome. RNA-Seq coverage is good in general; there is little expression before and no expression at all after the locus (see below).

26. Supplementary Fig. 5: The term colony should be avoided when talking about *P. patens* plants.

Thanks again. Correction has been made in this revision.

27. Supplementary Fig. 7: Which of the OE lines mentioned in Fig. S6 was used? Same in Figure 4 and Fig. S8.

AltYAN-OE-11# and *YAN-OE-11#* were used. Figures have been modified accordingly to include this information (Figure 4, and Supplementary Figures S12, S14).

Reviewers' Comments:

Reviewer #1:

Remarks to the Author:

Authors have done everything possible to improve their manuscript.

They might wish to consider checking Genevestigator and/or ePB browser because gene accessions can easily be converted from V1.6 to V3.1 and V3.2.

Congratulations to this exciting work which opens a new field in plant research.

Ralf Reski.

Reviewer #2:

Remarks to the Author:

Guan et al.

Gene refashioning through innovative shifting of reading frames in mosses

A much improved version, just a few minor comments.

While acknowledging that it may have been the original rationale for the study, the focus on early land plant evolution is not as relevant as the origin of new adaptive genes — e.g. the origin of YAN/AltYAN is not relevant to the origin of land plants, but rather some adaptation that arose in the Physcomitrium clade of mosses, relatively recently.

lines 43-55: somewhat antiquated given the prevalence of CRISPR-Cas mediated gene modification.

lines 55-63: these are really conclusions and could be reduced here to state that an origin of a novel gene was investigated.

lines 128-129, line 280: might be better to state that the Hr gene (coding sequence) has evolved rather than been lost?

line 154-155: a paraphyletic group in which Physcomitrella is embedded?

line 314: it is not clear that the oil bodies in liverworts are 'critical' to dehydration resistance — this seems to be based on a single circumstantial report. Also, there is still a confusion with the oil bodies in liverworts — they are not the same as the oil bodies in mosses or lycophytes, e.g. they do not have oleosin as a major component.

Reviewer #3:

Remarks to the Author:

The authors have significantly revised the manuscript according to the reviewers' comments; the present version is much improved.

However, I still have problems with some of the analyses. Below, I will refer to the numbering used in the authors response to my initial comments.

In summary, I cannot recommend to publish this work in its present form because some conclusions are incorrect. The authors are welcome to contact me with regard to the analyses I describe below.

Stefan A. Rensing, stefan.rensing@biologie.uni-marburg.de

6.

Please note that a blast search per se will detect similarity, as opposed to homology. Using the E-value as cutoff is not an appropriate measure to decide whether a database hit is actually homologous to the query sequence. This is particularly true at relatively high values like the one used by the authors (1E-06). The authors are referred to Rost et al. 1999 Protein Eng, where appropriate

combinations of identity and alignment length are described that allow to confidently determine homology based on blast results. This matter, unfortunately, is crucial for the paper, in particular since the conserved regions are relatively short and the evolutionary distances high. Along these lines, the default blast matrix that was used, BLOSUM62, might not be appropriate, since it assumes ca. 62% identity between query and hit - using e.g. BLOSUM45 might yield better results.

7.

Because I was worried that the analyses on which much of this paper relies were flawed I spent some time doing my own analyses. I used the *Marchantia* Hr sequence mentioned by the authors to search against a database of sequenced plant and algal genomes, 1KP bryophyte transcriptomes, and published charophyte and moss transcriptomes. A first glimpse at the resulting alignment and tree shows e.g. that there are homologs in charophyte algae (*Nitella hyalina*), in *Naegleria gruberi* (an amoeba), in *Volvox carteri* and *Chlamydomonas reinhardtii* (Chlorophyta), in *Cyanophora paradoxa* (Glaucophyta), and that the fungal sequences are nested in the plant sequences. Hence, the most parsimonious explanation for the distribution of the gene is that it was acquired (maybe by HGT from fungi) by the common ancestor of Archaeplastida and lost during vascular plant evolution (and maybe secondarily on some other lineages like red algae). Yet, a more detailed phylogenetic analysis would be in order to gain confidence.

9. / 21.

Since I considered it quite unlikely that the Yan/AltYan gene was acquired by the *Physcomitrium-Physcomitrella* species complex I also conducted an analysis in this regard, using the PpYan and AltYan sequences for a similar approach as mentioned under 8. While the authors describe only a single hit outside *Physcomitrella*, namely in a 1KP *Physcomitrium* sequence, my searches also found hits in *Encalypta streptocarpa* from 1KP, in the published transcriptome data of *Ceratodon purpureus* and *Funaria hygrometrica* (Szovenyi et al. 2014 and 2010), and in *Carica papaya*. While the latter might be a contamination, the fact that the gene can be detected not only in other Funariaceae, but also in *Ceratodon*, suggests that it was acquired a lot earlier, maybe even in a common ancestor of all Bryopsida. The lack of 1KP evidence might be due to the fact that the genes are not strongly expressed in the developmental stages from which the samples were generated.

11.

The fact that there is a gene in *Physcomitrium* at least allows to mention that it is older than the *Physcomitrium-Physcomitrella* species complex (see million year data in McDaniel et al.), but see comment above.

15.

Since the authors place emphasis on de- and rehydration (cf. 19.) I think appropriate references (not just one) should be cited.

As a side note, Prof. Reski mentioned that existing expression repositories might be tackled. The authors said this would not be possible due to the fact that those use v1.2 or v1.6 gene IDs. I would like to comment that this does not present a big problem, since gene model lookup is part of e.g. the browsers at Phytozome, cosmoss and CoGe. Looking up three gene models for old versions is not a major effort.

Major changes in this revision

We thank Drs. Ralf Reski, Stefan Rensing and the third reviewer for their comments and suggestions to improve this manuscript. We have carefully considered these comments and made corresponding changes. Some of the major changes are listed in the following.

1. E-value threshold for BLASTP of nr was changed to 1. This significant increase in the E-value threshold is somewhat usual, but it reflects our efforts to include more algal hits in our search. Fortunately, even with this usual threshold, the generated hits are overwhelmingly annotated as hemerythrins. We have provided this information in the manuscript (lines 69-72) and Supplementary Figure 1.
2. Relationships of algal and land plant hemerythrins. We performed some additional comprehensive searches for hemerythrins in glaucophytes and green algae by employing higher E-value thresholds (see above) and using various queries and databases. We realize that no search can be exhaustive, but our searches only identified hemerythrins in the glaucophyte *Cyanophora paradoxa* and the green algal class Chlorophyceae. We didn't find conclusive evidence for hemerythrins in charophytes and other chlorophytes, but this issue remains to be thoroughly investigated. We have provided this information in the manuscript (lines 91-99) and Supplementary Notes 1 and 2. Additionally, we have also added information of sequence comparisons and phylogenetic re-analyses, which suggests that algal and land plant hemerythrins are distantly related (Supplementary Figure S2 and Supplementary Figure 4b).
3. The origin of *Hr* in land plants. Because of the existence of the hemerythrin gene in glaucophytes and green plants, it is somewhat tempting to speculate that the gene was transferred from fungi to the most recent common ancestor of Archaeplastida, followed by different gene losses. We have specifically discussed this scenario in the manuscript (lines 285-292). Overall, we do not favor this scenario because **a**) algal and land plant sequences are not particularly related based on our analyses, and **b**) major lineages of fungi are much younger than the ancestor of Archaeplastida. However, we also discussed alternative scenarios and cautioned that the explanation may change if other evidence from new data, particularly those related to glaucophytes and charophytes, becomes available in future (lines 294-298).
4. Results of spore germination have been added to the manuscript. These results were not included in our last submission because of the time limitation. We show that spores from both wild-type and mutant plants can germinate and grow into chloronemata (lines 250-252; Supplementary Figure 16).
5. Other changes such as these about citations, oil bodies etc. as suggested by the reviewers.

Response to comments by reviewer 1 (Dr. Ralf Reski)

Authors have done everything possible to improve their manuscript. They might wish to consider checking Genevestigator and/or ePB browser because gene accessions can easily be converted from V1.6 to V3.1 and V3.2. Congratulations to this exciting work which opens a new field in plant research.

We are grateful to Dr. Reski for his kind words and encouragement. The improvement was only made possible thanks to the insightful comments and suggestions of Drs. Reski, Rensing and the third reviewer.

Following the suggestions of Drs. Reski and Rensing, we were able to locate a corresponding gene (Phypa_151693 in cosmass V1.2 and Pp1S342_30V6.1 in cosmass V1.6). The gene was originally annotated as *Hr* (HHE domain protein) in V1.6 and the annotation was changed since cosmass V3.1. According to ePB browser, Phypa_151693 is up-regulated in gametophores, rhizoids, spores and archegonia. Additionally, the gene is up-regulated during dehydration and rehydration according to Genevestigator (see figures below), which is consistent with the data of

Phytozome and our own RT-PCR experiments. We didn't include this information in the Supplementary Information because the annotation has already been changed in cosmoss.

Response to comments by reviewer 2

1. A much improved version, just a few minor comments. While acknowledging that it may have been the original rationale for the study, the focus on early land plant evolution is not as relevant as the origin of new adaptive genes — e.g. the origin of YAN/AltYAN is not relevant to the origin of land plants, but rather some adaptation that arose in the *Physcomitrium* clade of mosses, relatively recently.

Thanks for this suggestion. In this revision, we rephrased the first sentence into the following: “*Early-diverging land plants (embryophytes) provide some unique opportunities to understand the mechanisms of plant adaptation to terrestrial environments*” (lines 36-37). Hopefully, this will shift the focus of the study from land plant evolution to the adaptation mechanisms. Because *Hr* is distributed in many early-diverging land plants, it is difficult to discuss *YAN/AltYAN* without touching the evolution of early land plants.

2. Lines 43-55: somewhat antiquated given the prevalence of CRISPR-Cas mediated gene modification.

This sentence is deleted.

3. Lines 55-63: these are really conclusions and could be reduced here to state that an origin of a novel gene was investigated.

We appreciate this suggestion. Many readers do not have sufficient time to read the entire article. A brief summary in the Introduction will allow these readers to understand the major findings of the study without reading the entire article. As such, we chose to keep these sentences in this revision.

4. Lines 128-129, line 280: might be better to state that the *Hr* gene (coding sequence) has evolved rather than been lost?

We changed the first part into “is either lost or has evolved into new functions in *P. patens*” (line 141). The word “lost” is kept here because the presence of a premature stop codon itself does not suggest gene evolution. We realized that the gene has evolved into *YAN/AltYAN* only after two transcripts from the same locus region were identified.

In the second part, we deleted “in *Physcomitrella*” so that the *Hr* gene loss only applies to seed plants (line 303).

5. Line 154-155: a paraphyletic group in which *Physcomitrella* is embedded?

Changed (line 167).

6. Line 314: it is not clear that the oil bodies in liverworts are 'critical' to dehydration resistance — this seems to be based on a single circumstantial report. Also, there is still a confusion with the oil bodies in liverworts — they are not the same as the oil bodies in mosses or lycophytes, e.g. they do not have oleosin as a major component.

Thanks again for this comment on the role of liverwort oil bodies in desiccation resistance. In this revision, we removed sentences on oil bodies in liverworts. Since the oil bodies in liverworts and other land plants are different in structure, development and function, lumping them together in a discussion will only create confusion. We now write “*In mosses and many other land plants (e.g., seed plants), oil bodies play an important role in desiccation tolerance of plant tissues*” (lines 334-335). We realize this is still vague since the phrase “many other land plants” is not clearly defined. Nevertheless, this should keep the basic information correct.

Response to comments by reviewer 3 (Dr. Stefan Rensing)

1. Please note that a blast search per se will detect similarity, as opposed to homology. Using the E-value as cutoff is not an appropriate measure to decide whether a database hit is actually homologous to the query sequence. This is particularly true at relatively high values like the one used by the authors (1E-06). The authors are referred to Rost et al. 1999 Protein Eng, where appropriate combinations of identity and alignment length are described that allow to confidently determine homology based on blast results. This matter, unfortunately, is crucial for the paper, in particular since the conserved regions are relatively short and the evolutionary distances high. Along these lines, the default blast matrix that was used, BLOSUM62, might not be appropriate, since it assumes ca. 62% identity between query and hit - using e.g. BLOSUM45 might yield better results.

We thank Dr. Rensing for his comments on the relationships of sequence similarity and homology. Indeed, both Rost 1999 paper and BLAST incorporated sequence identity (or similarity) and length. The empirical rules proposed by Rost and E-values adopted by BLAST complement each other rather than being mutually exclusive [“*The thresholds for sequence identity and similarity defined here, complemented the levels for ‘significance’ provided by BLAST*” (Rost 1999. Protein Engineering 12:85-94)].

In our analyses, the conserved HHE domain is about 120 aa, which is relatively short for phylogenetic analyses, but still sufficient for the assessment of homology. The choice of E-value $1e-6$ as the cutoff essentially reflects our consideration of sequence length in BLAST search, since it is not possible to yield high bit scores, thus low E-values, for more distant Hr homologs over 120 aligned amino acid pairs. This E-value cutoff, however, should not significantly affect our assessment of sequence homology. This can be evidenced by our BLASTP of the nr database even with a much higher E-value cutoff in this revision [see figure below, which shows part of the BLASP results using a *Marchantia* Hr sequence (JGI ID: Mapoly0042s0093) as query]. In this search, E-value = 1 was used as the cutoff (see our response to next comment) and BLOSUM45 was chosen as the substitution scoring matrix. Only the most dissimilar hits (i.e., those with the lowest bit scores and the highest E-values) from the search are shown. Almost all these hits are clearly annotated as hemerythrins. We have added the above information to the revised manuscript [“*These hits are overwhelmingly annotated as hemerythrins* (Supplementary Figure S1)” (line 72)].

hemerythrin [Micromonospora echinospora]	40.3	40.3	43%	0.79	26%	WP_088984205.1
hemerythrin [Micromonospora inyonensis]	40.3	40.3	43%	0.81	26%	WP_091462030.1
hemerythrin HHE cation-binding protein [Streptomyces endus]	40.3	40.3	59%	0.81	26%	WP_067074069.1
hemerythrin [Streptomyces glaucescens]	40.6	40.6	46%	0.82	23%	WP_043505779.1
MULTISPECIES: hypothetical protein [Actinoplanes]	40.3	40.3	44%	0.83	32%	WP_014689342.1
hemerythrin [Mycobacterium marsellense]	40.3	40.3	58%	0.85	22%	WP_083019024.1
hemerythrin [Sphingomonas indica]	40.6	40.6	33%	0.85	30%	WP_085217127.1
hemerythrin [Sphingopyxis macrogoltabida]	40.3	40.3	42%	0.86	29%	WP_084758623.1
hemerythrin [Mycobacterium sp. IEC1808]	40.3	40.3	60%	0.87	25%	WP_085184661.1
hemerythrin HHE cation binding domain-containing protein [Mycobacterium sp. TS-1]	40.9	40.9	62%	0.88	21%	WP_023954761.1
Hemerythrin HHE cation binding domain-containing protein [Actinoadura echinospora]	40.3	40.3	62%	0.89	25%	SEG89047.1
hemerythrin [alpha proteobacterium LLX12A]	40.0	40.0	43%	0.89	25%	WP_017503021.1
hemerythrin [Sphingomonas sp. Y57]	40.0	40.0	43%	0.90	25%	WP_047166529.1
hemerythrin [Burkholderia multivorans]	39.8	39.8	34%	0.92	37%	WP_088923459.1
hypothetical protein [Streptomyces hygrosopicus]	40.3	40.3	59%	0.93	26%	WP_030830277.1
MULTISPECIES: hemerythrin [Sphingobium]	40.0	40.0	43%	0.93	25%	WP_069335624.1
Hemerythrin HHE cation binding region [Mycobacterium smegmatis str. MC2 155]	39.8	39.8	45%	0.95	28%	AFP39690.1
MULTISPECIES: hemerythrin [Streptomyces]	40.3	40.3	54%	0.96	22%	WP_031142329.1
hemerythrin [Mycobacterium avium]	40.3	40.3	58%	0.96	26%	WP_084022155.1
hemerythrin [Streptomyces sp. NRRL WC-3795]	40.3	40.3	60%	0.97	21%	WP_031019374.1
hemerythrin [Mycobacterium sp. 1245111.1]	40.3	40.3	58%	0.97	22%	WP_067331242.1
hemerythrin [Streptomyces rimosus subsp. pseudoverticillatus]	40.0	40.0	45%	0.97	22%	KOT87501.1
hypothetical protein [Actinoadura madurae]	40.3	40.3	57%	0.98	24%	WP_021593450.1
hemerythrin [Mycobacterium colombiense]	40.3	40.3	59%	1.00	23%	WP_077091455.1

We also concur with Dr. Rensing that BLOSUM45 is a better choice for BLAST search of the hemerythrin gene. With *Marchantia* hemerythrins as query, we compared the BLAST results using the two matrices. The generated hits were basically the same, though their bit scores and E-values were slightly different. In this revision, we indicate that BLOSUM45 was used (line 70).

2. Because I was worried that the analyses on which much of this paper relies were flawed I spent some time doing my own analyses. I used the *Marchantia* Hr sequence mentioned by the authors to search against a database of sequenced plant and algal genomes, 1KP bryophyte transcriptomes, and published charophyte and moss transcriptomes. A first glimpse at the resulting alignment and tree shows e.g. that there are homologs in charophyte algae (*Nitella hyalina*), in *Naegleria gruberi* (an amoeba), in *Volvox carteri* and *Chlamydomonas reinhardtii* (Chlorophyta), in *Cyanophora paradoxa* (Glaucophyta), and that the fungal sequences are nested in the plant sequences. Hence, the most parsimonious explanation for the distribution of the gene is that it was acquired (maybe by HGT from fungi) by the common ancestor of Archaeplastida and lost during vascular plant evolution (and maybe secondarily on some other lineages like red algae). Yet, a more detailed phylogenetic analysis would be in order to gain confidence.

We appreciate the comments of Dr. Rensing about the distribution of *Hr* in algae. In our last submission, we did identify *Hr* hits in green algae, including both chlorophytes and charophytes. The charophytes hits were only from 1KP, and the possibility that these sequences resulted from contamination was discussed in Supplementary Table S3 and Supplementary Note 1 (now Supplementary Note 2). We noted in our last submission that contamination is a serious issue for 1KP data. In this revision, we also searched the *Nitella hyalina* transcriptomic data at NCBI (<https://trace.ncbi.nlm.nih.gov/Traces/sra/?run=SRR064326>), but only two matches of 19-32 nucleotides were found; these two matches were mapped onto different regions (separated by about 470 nucleotides) of the *Hr* CDS (see figure below, upper panel). Search of the transcriptomic data of the charophyte [Closterium peracerosum-strigosum-littorale complex](https://www.ncbi.nlm.nih.gov/bioproject?term=PRJNA296352) at NCBI (<https://www.ncbi.nlm.nih.gov/bioproject?term=PRJNA296352>) provided a similar result (also see below, lower panel). We further searched the over 650 transcriptomes in the Marine Microbial Eukaryote Transcriptome Sequencing Project (MMETSP), and the results were largely consistent with our previous findings.

BLASTN search of transcriptomic data of charophytes *Nitella hyalina* (top panel) and *Closterium peracerosum-strigosum-littorale* complex (lower panel) at NCBI. Query is the CDS of *Marchantia polymorpha* Hr (JGI ID: Mapoly0042s0093).

We realize that the assessment of sequence homology is more difficult at the level of nucleotide sequences, and that lack of sufficient query coverage in the above search does not necessarily suggest the absence of *Hr* in the two charophytes. As such, we further investigated whether the 2-3 charophyte hits were specific to *Hr* (i.e., HHE domain). To this end, we performed the BLASTN search of the NCBI non-redundant nucleotide sequence database using the same *Marchantia polymorpha* Hr (JGI ID: Mapoly0042s0093) as query. Indeed, the results included hits corresponding to those from the two charophytes (*Nitella hyalina* and *Closterium peracerosum-strigosum-littorale* complex) (see figure below, top panel). However, further inspections of these hits indicated that they were not particular to *Hr*. For instance, the hits to the 670-600 bp region of the query were annotated as part of the genes encoding small integral membrane protein 10-like protein 2A, peroxidase 7-like protein, and myomegalin-like protein; they were found in both flowering plants (*Lupinus angustifolius*, *Vitis venifera*, and *Cucurbita maxima*) and animals (*Crocodylus porosus*, *Oncorhynchus mykiss*, and *Columba livia*). Similarly, hits to the 150-200 bp region of the query were annotated as genes encoding erythrocyte membrane protein, RP1 like 1 (rp111) protein, and retrotransposon Gag like 5 (Rtl5) protein; they could be found in animals (e.g., *Oryzias latipes*, *Heterocephalus glaber*, *Labrus bergylta*) and apicomplexan parasites (*Plasmodium falciparum*). On the other hand, the most significant hits, which also had the longest query coverages, were from *Selaginella moellendorffii*, the fungus *Fusarium verticillioides* and *Physcomitrella patens* (see figure below, lower panel). The former two (*Selaginella* and *Fusarium*) were part of the *Hr* gene, whereas the later (i.e. *P. patens*) evolved directly from *Hr*. These data suggest that the hits from the two charophytes (*Nitella hyalina* and *Closterium peracerosum-strigosum-littorale* complex) might be associated with genes other than *Hr*. In addition, it also likely points to the close relationship between fungal and land plant hemerythrins. Nevertheless, this issue of *Hr* in charophytes remains to be thoroughly investigated, and we cautioned in this manuscript that the conclusion may change if new data from glaucophytes and charophytes become available in future (lines 296-298).

Because complete genome sequence data have longer scaffolds, they are more reliable sources for assessing the existence of a gene in a given genome. As such, our search also relies heavily on complete genome sequence data. Our initial BLAST search of the nr database, which contains annotated protein sequences of the vast majority of published algal genomes, adopted an E-value cutoff $1e-6$. This cutoff is sufficient in most searches and identified sequences from chlorophyte *Gonium pectoral* and *Monoraphidium neglectum*. We also performed various other searches of both nr and Phytozome; these searches indeed identified sequences from additional chlorophytes such as *Volvox carteri* and *Chlamydomonas reinhardtii*, but this would require using a much higher E-value cutoff or employing chlorophyte *Gonium* or *Monoraphidium* sequences as query. All these green algae (*Gonium*, *Monoraphidium*, *Volvox*, *Chlamydomonas*, and *Chromochloris*) belong to the class Chlorophyceae. We didn't identify sequences from charophytes and other chlorophytes (admittedly, not many charophyte genomes have been sequenced; the NCBI charophyte *Spirogyra* sp. AU1 BioProject site does not appear to be functional and no data can be accessed). We have added the information on *Hr* distribution in other Chlorophyceae in this revision (lines 70-77). To reflect the change in search results, we have changed the E-value cutoff from $1e-6$ to 1 in this revision. Although this high E-value cutoff is somewhat unusual for most searches, fortunately it does not have a major effect on our identification of *Hr* homologs (Supplementary Fig. S1; also see our reply to comment 1). Furthermore, information of pairwise comparisons has also been included, showing that land plants Hrs share higher sequence percent identities with fungal homologs than with chlorophyte and glaucophyte sequences (Supplementary Fig. S2).

In terms of the scenario that the *Hr* gene was transferred from fungi to the most recent common ancestor of Archaeplastida, we have carefully weighted different lines of evidence. Overall, we do not favor this scenario, particularly for the following reasons:

a. As indicated above, green algal hits were also found in our initial analyses. We performed very comprehensive analyses of these algal genes using various queries and databases. Both sequence similarity comparisons and phylogenetic analyses suggest that these algal sequences are not particularly related to land plant hemerythrins. These analyses were detailed in our last submission (now lines 66-90; now Supplementary Figure S4; Supplementary Tables 1-3; now Supplementary Note 2). Results of our additional phylogenetic analyses in this revision are consistent with our earlier conclusions. As shown in Supplementary Figure S4b, sequences from two chlorophycean (*Gonium* and *Volvox*) are only distantly related to land plant homologs (the glaucophyte *Cyanophora* sequence was removed from the analyses because of its much shorter length; inclusion of *Cyanophora* sequences in the analyses provided a similar topology with lower branch support).

b. Given the limited genome data for glaucophytes, the distribution *Hr* in this group remains to be seriously investigated. However, the seeming restriction of *Hr* to Chlorophyceae in green algae would require multiple loss events for a scenario of HGT to the ancestor of Archaeplastida. Although loss of *Hr* does happen (e.g., in most vascular plants), the more loss events postulated, the less likely the scenario. On the other hand, an independent HGT to Chlorophyceae is at least an equally parsimonious explanation.

c. When assessing the occurrence and direction of gene transfer events, an important consideration is the temporal sequence in which the donor and recipient evolved. The donor must evolve no later than the recipient (Huang and Gogarten 2006. Trends in Genetics 22: 361-366). In the specific scenario of fungi-to-Archaeplastida gene transfer, Archaeplastida evolved about 1400 MYA (Hedges and Kumar eds 2009. The Time Tree of Life; hereafter). Although fungi split from other opisthokonts about the same time, major fungal lineages only evolved 980-1150 MYA. This much younger age of major fungal lineages suggests that horizontal transfer of the *Hr* gene from fungi to the most recent common ancestor of Archaeplastida is an unlikely scenario. We have discussed this issue in this revision (lines 284-292)

We also would like to note here that the determination of HGT is based on overall evidence. For every HGT event proposed, there are multiple other explanations. In particular, differential gene loss is always an alternative explanation to HGT. As indicated above, the more loss events to be postulated, the less likely the gene loss scenario. Specifically for *Hr*, it is unrealistic to expect a well-resolved phylogeny because of the short length of HHE domain, which in turn translates into difficulties in assessing the origin of *Hr* in algae. The suggestion of HGT from fungi to the ancestor of land plants is based on our assessment of gene distribution, phylogeny, and sequence similarity. As new methods or sequence data become available, it is possible that the conclusion (or explanation) will change. This is exactly the reason that other possible explanations (including gene loss) were included in our last and current submissions (lines 294-296).

3. Since I considered it quite unlikely that the Yan/AltYan gene was acquired by the Physcomitrium-Physcomitrella species complex I also conducted an analysis in this regard, using the PpYan and AltYan sequences for a similar approach as mentioned under 8. While the authors describe only a single hit outside Physcomitrella, namely in a 1KP Physcomitrium sequence, my searches also found hits in Encalypta streptocarpa from 1KP, in the published transcriptome data of Ceratodon purpureus and Funaria hygrometrica (Szovenyi et al. 2014 and 2010), and in Carica papaya. While the latter might be a contamination, the fact that the gene can be detected not only in other Funariaceae, but also in Ceratodon, suggests that it was acquired a lot earlier, maybe even in a common ancestor of all Bryopsida. The lack of 1KP evidence might be due to the fact that the genes are not strongly expressed in the developmental stages from which the samples were generated.

We are sorry for the confusion around the distribution of *YAN* and *AltYAN*. In our last submission, we wrote “In particular, the search of 1KP for *AltYAN* provided no hit outside *Physcomitrium*; only a single hit (1KP ID: YEPO-2062682), which corresponded to the 5’UTR and the first exon of *AltYAN*, was identified in *Physcomitrium*” (now lines 168-171). The single hit in our search only applies to *AltYAN* (see figure below).

```
Query= Pp3c21_19720V3.1 AA (AltYAN peptide)
Length=123
Sequences producing significant alignments:
      scaffold-YEPO-2062682-cf._Physcomitrium_sp.          54.3   2e-07
> scaffold-YEPO-2062682-cf._Physcomitrium_sp.
Length=315
Score = 54.3 bits (129), Expect = 2e-07, Method: Compositional matrix adjust.
Identities = 43/47 (91%), Positives = 44/47 (94%), Gaps = 0/47 (0%)
Frame = -1
Query 1   MAAVAYIPLSAVASARLattgasssnaasqpsaGIVAFKRAVTPSCL 47
          MAAVAYIPL+AVASARLATTQA SNAASQPSAGIVAFKRA TPSCL
Sbjct 141 MAAVAYIPLNAVASARLATTQARFSNAASQPSAGIVAFKRAATPSCL 1
```

On the other hand, because *Hr* and *YAN* partially share the conserved HHE region (same nucleotide sequence, but different genes) and because *Hr* is found in many bryophytes, it is not surprising that hits will be found in other bryophytes when *YAN* is used as query. This has been discussed in our last submission (now lines 167-168: “No homolog was identified in nr whereas only fragmented sequence matches, often with premature stop codons, were found in 1KP”). That said, these hits most likely do not represent intact *YAN*; they most likely are matches to *Hr* instead of *YAN*.

During this revision, we carefully performed re-analyses of the distribution of *YAN* and

AltYAN. Results of TBLASTN search of 1KP and nr databases were consistent with the findings in our last submission. Only one hit of *AltYAN* was identified in *Physcomitrium* in 1KP (see figure above); there were hits of *YAN* in other bryophytes (including *Encalypta streptocarpa* and *Ceratodon purpureus*), but these hits contain premature stop codons, indicating they are not *YAN* (see figures below). Even if *YAN* exists in other bryophytes, it does not automatically suggest the existence the dual-coding gene *YAN/AltYAN*. It will likely support our speculation that “*YAN* might have initially evolved as an alternative transcript of *Hr*, and their overlapping in coding regions led to functional linkage” (now lines 398-399). The suggestion of *YAN/AltYAN* existence requires finding both transcripts from the same genome.

```
> scaffold-FFPD-2009361-Ceratodon_purpureus
Length=1258

Score = 105 bits (263), Expect = 2e-23, Method: Compositional matrix adjust.
Identities = 90/189 (48%), Positives = 108/189 (57%), Gaps = 4/189 (2%)
Frame = +2

Query 60 TVHHGRSRIHPLERCFREVGNHPSQLQQCGLPAQRGHCCVQASCHAILPAVGEPVLRPP 119
          HHGR + R P +L QCG AQ + C + A G+ L R
Sbjct 95 IFHHGRFSMRSTHRGSHELELAGPCEL*QCGPAAQLPGVEEELHCRVVCSAFGQSFLWRS 274

Query 120 RHFLaaaaaaaPGREETGRVDQMLRHSEEFGGGRRHHRQSEAGPQAGGGVLQEQVPOA 179
          R L + EE + H + G R HHRQ EAGPQAGGG+LQEQVPOA
Sbjct 275 RDILGICCKG----EERRWIRPGHGHROHGEGRRGHHRQSEAGPQAGGGIQLQEQVPOA 442

Query 180 GQRGRGRIMVQPVVRVGDLSPCRHRGAGVVPVDRVAGROGPEAGGPVARRAPEDEGHAGGD 239
          G+RG G+ +VQP+RVG + P R GA VPVDR+AG +GPEAGGPVAR AP + HAGGD
Sbjct 443 GRRGGGQVVQPIRVGGVPLRDGGARVPVDRIAGGEGPEAGGPVARGAPGGQEHAGGD 622

Query 240 PGHRGRRPV 248
          PG + RPV
Sbjct 623 PGDQRPRPV 649

> scaffold-KEFD-2005771-Encalypta_streptocarpa
Length=926

Score = 52.4 bits (124), Expect = 9e-05, Method: Compositional matrix adjust.
Identities = 57/147 (39%), Positives = 75/147 (51%), Gaps = 9/147 (6%)
Frame = -1

Query 99 CVOASCHAI--LPAVGEPVLRPRHFLaaaaaaPGREETGRVDQMLRHSEEFGGGRRHH 156
          C+Q + L A+ + LRR +FL+ Q G +E M G RRH+
Sbjct 698 CIQEESDGLFFLSAISKAFLLRSSYFLRLHCQ---GIQEWPP--SPMHGQQGHKGRRRHY 534

Query 157 RQSEAGPQAGGGVLQEQVPOAGQRGRGRIMVQPVVRVGDLSPCRHRGAGVVPVDRVAGR 216
          RQ A P VLQL EVPQ GQR R +VQP+RVG+L P RHRG +P+D
Sbjct 533 RQD*ARPXXA--VLQL*EVPOGGORRRR*VVQPIRVGNLPPRRHRGTRPLPLDWFH*H 360

Query 217 QGPEAGGPVARRAPEDEGHAGGDPGHR 243
          G E G + A +++G G DPGH+
Sbjct 359 *GQEFG**IPCGASDNQGLVGRDPGHQ 279
```

Following the comments of Dr. Rensing, we also downloaded the original transcriptomic data generated by Szovenyi et al. for *Ceratodon purpureus* and *Funaria hygrometrica*. Raw reads of *Funaria hygrometrica* were also assembled using Trinity. We then performed BLAST search, both BLASTN and TBLASTN (E-value cutoff = 10), of the downloaded sequences for *YAN* and *AltYAN*. No hits of *AltYAN* were found in *Ceratodon purpureus* and only a match of 12 amino acids was found in *Funaria hygrometrica*. Consistent with the TBLASTN results of 1KP data, hits of *YAN* were found in *Ceratodon purpureus*, but they contain premature stop codons. These BLAST results and alignments are shown in the following, where the prefixes isotig and

TRINITY_DN indicate sequences from *Ceratodon purpureus* and *Funaria hygrometrica*, respectively.

a. BLAST result of *Ceratodon purpureus* transcriptomic data for *AltYAN*. Results of BLASTN indicate that the lowest E-value is 2.4 and the highest bit score is 32.2. TBLASTN provides no hits.

```

# BLASTN 2.2.26 [Sep-21-2011]
# Query: Pp3c21_19720V3.1 CDS PpAltYAN
# Database: R40_454Isotigs.fna
# Fields: Query id, Subject id, % identity, alignment length, mismatches, gap openings, q. start, q. end, s. start, s. end, e-value, bit score
Pp3c21_19720V3.1 isotig15492 100.00 16 0 0 268 283 281 266 2.4 32.2
Pp3c21_19720V3.1 isotig14853 95.00 20 1 0 271 290 61 80 2.4 32.2
Pp3c21_19720V3.1 isotig09623 100.00 16 0 0 267 282 378 393 2.4 32.2
Pp3c21_19720V3.1 isotig03531 100.00 16 0 0 310 325 3464 3449 2.4 32.2
Pp3c21_19720V3.1 isotig03530 100.00 16 0 0 310 325 3487 3472 2.4 32.2
Pp3c21_19720V3.1 isotig01478 100.00 16 0 0 271 286 2158 2143 2.4 32.2
Pp3c21_19720V3.1 isotig01477 100.00 16 0 0 271 286 2108 2093 2.4 32.2
Pp3c21_19720V3.1 isotig01476 100.00 16 0 0 271 286 2169 2154 2.4 32.2
Pp3c21_19720V3.1 isotig01475 100.00 16 0 0 271 286 2119 2104 2.4 32.2
Pp3c21_19720V3.1 isotig19152 100.00 15 0 0 16 30 246 260 9.4 30.2
Pp3c21_19720V3.1 isotig14849 100.00 15 0 0 346 360 518 532 9.4 30.2
Pp3c21_19720V3.1 isotig11812 100.00 15 0 0 272 286 1351 1365 9.4 30.2
Pp3c21_19720V3.1 isotig10874 100.00 15 0 0 30 44 1260 1246 9.4 30.2
Pp3c21_19720V3.1 isotig09042 100.00 15 0 0 97 111 22 8 9.4 30.2
Pp3c21_19720V3.1 isotig09021 100.00 15 0 0 271 285 332 346 9.4 30.2
Pp3c21_19720V3.1 isotig08926 100.00 15 0 0 36 50 1629 1615 9.4 30.2
Pp3c21_19720V3.1 isotig08824 100.00 15 0 0 19 33 432 446 9.4 30.2
Pp3c21_19720V3.1 isotig08685 100.00 15 0 0 305 319 690 704 9.4 30.2
Pp3c21_19720V3.1 isotig08492 100.00 15 0 0 34 48 1532 1546 9.4 30.2
Pp3c21_19720V3.1 isotig07918 100.00 15 0 0 269 283 14 28 9.4 30.2
Pp3c21_19720V3.1 isotig07892 100.00 15 0 0 262 276 1512 1526 9.4 30.2
Pp3c21_19720V3.1 isotig07859 100.00 15 0 0 269 283 3547 3533 9.4 30.2
Pp3c21_19720V3.1 isotig05481 100.00 15 0 0 157 171 673 659 9.4 30.2
Pp3c21_19720V3.1 isotig05480 100.00 15 0 0 157 171 673 659 9.4 30.2
Pp3c21_19720V3.1 isotig03961 100.00 15 0 0 70 84 324 338 9.4 30.2
Pp3c21_19720V3.1 isotig03960 100.00 15 0 0 70 84 324 338 9.4 30.2

```

BLASTN 2.2.26 [Sep-21-2011]

Reference: Altschul, Stephen F., Thomas L. Madden, Alejandro A. Schaffer, Jinghui Zhang, Zheng Zhang, Webb Miller, and David J. Lipman (1997), "Gapped BLAST and PSI-BLAST: a new generation of protein database search programs", *Nucleic Acids Res.* 25:3389-3402.

Reference for compositional score matrix adjustment: Altschul, Stephen F., John C. Wootton, E. Michael Gertz, Richa Agarwala, Aleksandr Morgulis, Alejandro A. Schaffer, and Yi-Kuo Yu (2005) "Protein database searches using compositionally adjusted substitution matrices", *FEBS J.* 272:5101-5109.

Query= Pp3c21_19720V3.1 AltYAN
(122 letters)

Database: R40_454Isotigs.fna
20,431 sequences; 34,112,501 total letters

Searching..... done

***** No hits found *****

Database: R40_454Isotigs.fna
Posted date: Nov 27, 2017 5:46 PM
Number of letters in database: 34,112,501
Number of sequences in database: 20,431

b. BLAST result of *Funaria hygrometrica* transcriptomic data for *AltYAN*. Results of BLASTN indicate that the lowest E-value is 0.96 and the highest bit score is 32.2. TBLASTN provides a single hit of only 12 aa.

```
# BLASTN 2.2.26 [Sep-21-2011]
# Query: Pp3c21_19720V3.1 CDS PpAltYAN
# Database: Trinityall.fasta
# Fields: Query id, Subject id, % identity, alignment length, mismatches, gap openings, q. start, q. end, s. start, s. end, e-value, bit score
Pp3c21_19720V3.1 TRINITY_DN5722_c0_g1_i2 100.00 16 0 0 289 304 219 204 0.96 32.2
Pp3c21_19720V3.1 TRINITY_DN4289_c143_g1_i1 100.00 15 0 0 279 293 21 35 3.8 30.2
Pp3c21_19720V3.1 TRINITY_DN9083_c0_g1_i1 100.00 15 0 0 221 235 1681 1667 3.8 30.2
Pp3c21_19720V3.1 TRINITY_DN3266_c0_g1_i1 100.00 15 0 0 208 222 294 308 3.8 30.2
Pp3c21_19720V3.1 TRINITY_DN5454_c0_g2_i4 100.00 15 0 0 113 127 384 398 3.8 30.2
Pp3c21_19720V3.1 TRINITY_DN5454_c0_g2_i3 100.00 15 0 0 113 127 411 425 3.8 30.2
Pp3c21_19720V3.1 TRINITY_DN5454_c0_g2_i2 100.00 15 0 0 113 127 346 360 3.8 30.2
Pp3c21_19720V3.1 TRINITY_DN5454_c0_g2_i1 100.00 15 0 0 113 127 134 148 3.8 30.2
Pp3c21_19720V3.1 TRINITY_DN5537_c0_g1_i3 100.00 15 0 0 328 342 241 227 3.8 30.2
Pp3c21_19720V3.1 TRINITY_DN5537_c0_g1_i2 100.00 15 0 0 328 342 375 361 3.8 30.2
Pp3c21_19720V3.1 TRINITY_DN5537_c0_g1_i1 100.00 15 0 0 328 342 375 361 3.8 30.2
```

```
TBLASTN 2.2.26 [Sep-21-2011]
Reference: Altschul, Stephen F., Thomas L. Madden, Alejandro A. Schaffer,
Jinghui Zhang, Zheng Zhang, Webb Miller, and David J. Lipman (1997),
"Gapped BLAST and PSI-BLAST: a new generation of protein database search
programs", Nucleic Acids Res. 25:3389-3402.
```

```
Reference for compositional score matrix adjustment: Altschul, Stephen F.,
John C. Wootton, E. Michael Gertz, Richa Agarwala, Aleksandr Morgulis,
Alejandro A. Schaffer, and Yi-Kuo Yu (2005) "Protein database searches
using compositionally adjusted substitution matrices", FEBS J. 272:5101-5109.
```

```
Query= Pp3c21_19720V3.1 AltYAN
(122 letters)
```

```
Database: Trinityall.fasta
25,144 sequences; 13,914,117 total letters
Searching.....done
```

```
Sequences producing significant alignments:
Score E
(bits) Value
TRINITY_DN6098_c0_g2_i1 len=656 path=[379:0-24 381:25-74 431:75-... 25 4.2
>TRINITY_DN6098_c0_g2_i1 len=656 path=[379:0-24 381:25-74 431:75-86
443:87-124 481:125-156 513:157-337 694:338-359
716:360-389 746:390-396 753:397-416 773:417-487
844:488-507 864:508-611 968:612-631 988:632-655] [-1,
379, 381, 431, 443, 481, 513, 694, 716, 746, 753, 773,
844, 864, 968, 988, -2]
Length = 656
Score = 25.4 bits (54), Expect = 4.2, Method: Compositional matrix adjust.
Identities = 10/22 (45%), Positives = 14/22 (63%)
Frame = +2
```

```
Query: 100 RAAIVTWWMSVGGNCAVLPAGH 121
R I TW ++ G C++L AGH
Sbjct: 476 RPRISTWLLADSGKCSMLRAGH 541
```

```
Database: Trinityall.fasta
Posted date: Nov 29, 2017 3:25 PM
Number of letters in database: 13,914,117
Number of sequences in database: 25,144
```

c. BLAST result of *Ceratodon purpureus* transcriptomic data for *YAN*. BLASTN search shows two significant matches to *YAN*. However, both sequences contain premature stop codons, as shown in the TBLASTN results.

```

# BLASTN 2.2.26 [Sep-21-2011]
# Query: Pp3c21_19720V3.2 CDS PpYAN
# Database: R40_454Isotigs.fna
# Fields: Query id, Subject id, % identity, alignment length, mismatches, gap openings, q. start, q. end, s. start, s. end, e-value, bit score
Pp3c21_19720V3.2 isotig02033 87.12 295 38 0 450 744 359 653 8e-076 283
Pp3c21_19720V3.2 isotig15024 84.00 275 42 2 450 723 796 523 9e-045 180
Pp3c21_19720V3.2 isotig15598 92.45 53 4 0 563 615 577 525 1e-012 73.8
Pp3c21_19720V3.2 isotig18683 100.00 18 0 0 507 524 543 560 0.32 36.2
Pp3c21_19720V3.2 isotig11960 100.00 18 0 0 700 717 479 496 0.32 36.2
Pp3c21_19720V3.2 isotig15908 95.24 21 1 0 721 741 880 900 1.3 34.2
Pp3c21_19720V3.2 isotig15908 100.00 16 0 0 490 505 455 440 4.9 32.2
Pp3c21_19720V3.2 isotig13658 95.24 21 1 0 646 666 972 952 1.3 34.2
Pp3c21_19720V3.2 isotig00707 100.00 17 0 0 438 454 438 422 1.3 34.2
Pp3c21_19720V3.2 isotig00704 100.00 17 0 0 438 454 1724 1708 1.3 34.2
Pp3c21_19720V3.2 isotig00703 100.00 17 0 0 438 454 2598 2582 1.3 34.2
Pp3c21_19720V3.2 isotig17508 100.00 16 0 0 490 505 601 616 4.9 32.2
Pp3c21_19720V3.2 isotig16419 100.00 16 0 0 584 599 375 390 4.9 32.2
Pp3c21_19720V3.2 isotig14953 100.00 16 0 0 607 622 199 214 4.9 32.2
Pp3c21_19720V3.2 isotig14856 100.00 16 0 0 519 534 484 499 4.9 32.2
Pp3c21_19720V3.2 isotig14349 100.00 16 0 0 687 702 467 452 4.9 32.2
Pp3c21_19720V3.2 isotig14087 100.00 16 0 0 494 509 1104 1089 4.9 32.2
Pp3c21_19720V3.2 isotig12533 95.00 20 1 0 157 176 182 163 4.9 32.2
Pp3c21_19720V3.2 isotig12247 100.00 16 0 0 653 668 457 442 4.9 32.2
Pp3c21_19720V3.2 isotig11523 100.00 16 0 0 441 456 1717 1732 4.9 32.2
Pp3c21_19720V3.2 isotig11417 100.00 16 0 0 491 506 1760 1745 4.9 32.2
Pp3c21_19720V3.2 isotig10538 100.00 16 0 0 161 176 1061 1046 4.9 32.2
Pp3c21_19720V3.2 isotig06562 100.00 16 0 0 695 710 1317 1332 4.9 32.2
Pp3c21_19720V3.2 isotig02185 100.00 16 0 0 173 188 436 421 4.9 32.2
Pp3c21_19720V3.2 isotig02184 100.00 16 0 0 173 188 436 421 4.9 32.2
Pp3c21_19720V3.2 isotig01623 100.00 16 0 0 707 722 615 600 4.9 32.2
Pp3c21_19720V3.2 isotig01622 100.00 16 0 0 707 722 750 735 4.9 32.2

```

```

#BLASTN 2.2.26 [Sep-21-2011]
Reference: Altschul, Stephen F., Thomas L. Madden, Alejandro A. Schaffer,
Jinghui Zhang, Zheng Zhang, Webb Miller, and David J. Lipman (1997),
"Gapped BLAST and PSI-BLAST: a new generation of protein database search
programs", Nucleic Acids Res. 25:3389-3402.

```

```

Reference for compositional score matrix adjustment: Altschul, Stephen F.,
John C. Wootton, E. Michael Gertz, Richa Agarwala, Aleksandr Morgulis,
Alejandro A. Schaffer, and Yi-Kuo Yu (2005) "Protein database searches
using compositionally adjusted substitution matrices", FEBS J. 272:5101-5109.

```

```

Query= Pp3c21_19720V3.2 YAN
(248 letters)

```

```

Database: R40_454Isotigs.fna
20,431 sequences; 34,112,501 total letters

```

```

Searching..... done

```

```

Sequences producing significant alignments:
Score E
(bits) Value
isotig15024 gene=isogroup10253 length=1137 numContigs=1 104 5e-026
isotig02033 gene=isogroup00267 length=1306 numContigs=3 101 7e-025
isotig15598 gene=isogroup10827 length=1054 numContigs=1 61 6e-011
isotig07822 gene=isogroup03051 length=4146 numContigs=1 28 5.9

```

```

>isotig15024 gene=isogroup10253 length=1137 numContigs=1
Length = 1137

```

```

Score = 104 bits (259), Expect = 5e-026, Method: Compositional matrix adjust.
Identities = 73/165 (44%), Positives = 90/165 (54%), Gaps = 1/165 (0%)
Frame = -1

```

```

Query: 78 EVGNHPSQLQQCGPLPAQRGHC-CVQASCHAILPAVGEPVLRPRHFLXXXXXXXXPGREET 136
E + LQ+C +PA R C Q C + +P+ R HFL ++
Sbjct: 996 EYNSELHSLQRC-IPATRIQLHCTQVQCEL---EIRQPIRWRSSHFLKLLSCQGIKQWP 829

```

```

Query: 137 GRVDQMLRHSEEFGGRRHHRQSEAGPQGAGGGVLQLQEVPQAGQRGRGRIMVQPVVVD 196
+ M H E G R HHRQ EAGPQGAGG +LQLQE+PQA QR R +VQP+RVG
Sbjct: 828 ---NAMHGHE*ECTGRR*HHRQGEAGPQGAGGSLQLQEIPQAEQRRRSG*VVQPIRVGS 658

```

```

Query: 197 LSPCRHRGAGVVPVDRVAGRQGPVARRAPEDEGHAGGDPG 241

```

```

>isotig02033  gene=isogroup00267  length=1306  numContigs=3
      Length = 1306

Score = 101 bits (252), Expect = 7e-025, Method: Compositional matrix adjust.
Identities = 89/189 (47%), Positives = 106/189 (56%)
Frame = +3

Query: 60  TVHHGRSRIHPLERCRFRFVGNHPSQLQQCGLPAQRGHCCVQASCHAILPAVGEPVLRFP 119
           HHGR + R P +L QCG AQ + + A G+ L R
Sbjct: 99  IFHHGRFMSMRSTHRGSHELELAGPCEL*QCGPAAQLPGVEEELHRRVVCSAFGQSFLWRS 278

Query: 120 RHFLXXXXXXXXPGREETGRVDQMLRHSEEFGGGRRHHRQSEAGPQGAGGGVLQLQEVPPA 179
           R L EE + H + G R HHRQ EAGPQGAGGG+LQLQEVPPA
Sbjct: 279 RDILGICCKG----EERRWIRPQGHRQHGEGRGHRHHRQSEAGPQGAGGGILQLQEVPPA 446

Query: 180 GQRGRGRIMVQPVVGVGDLSPCRHRGAGVVPVDRVAGROGPEAGGPVARRAPEDEGHAGGD 239
           G+RG G+ +VQP+RVG + P R GA VPVDR+AG +GPEAGGPVAR AP + HAGGD
Sbjct: 447 GRRGGGQQVQPIRVGGVPLRDGGARPVVPVDRIAGGEGPEAGGPVARGAPGGQEHAGGD 626

Query: 240 PGHRGRRPV 248
           PG + RPV
Sbjct: 627 PGDQRPRPV 653

```

d. TBLASTN result of *Funaria hygrometrica* transcriptomic data for *YAN*, indicating that no hits were found.

```

|BLASTN 2.2.26 [Sep-21-2011]

Reference: Altschul, Stephen F., Thomas L. Madden, Alejandro A. Schaffer,
Jinghui Zhang, Zheng Zhang, Webb Miller, and David J. Lipman (1997),
"Gapped BLAST and PSI-BLAST: a new generation of protein database search
programs", Nucleic Acids Res. 25:3389-3402.

Reference for compositional score matrix adjustment: Altschul, Stephen F.,
John C. Wootton, E. Michael Gertz, Richa Agarwala, Aleksandr Morgulis,
Alejandro A. Schaffer, and Yi-Kuo Yu (2005) "Protein database searches
using compositionally adjusted substitution matrices", FEBS J. 272:5101-5109.

Query= Pp3c21_19720V3.2  YAN
      (248 letters)

Database: Trinityall.fasta
      25,144 sequences; 13,914,117 total letters

Searching..... done

**** No hits found ****

Database: Trinityall.fasta

```

4. The fact that there is a gene in *Physcomitrium* at least allows to mention that it is older than the *Physcomitrium-Physcomitrella* species complex (see million year data in McDaniel et al.), but see comment above.

We might have missed something here. The gene is present in an unspecified species of *Physcomitrium*. Whether this species is closely related to individual subspecies of *Physcomitrella patens* or other *Physcomitrium* species remains to be investigated. We think it is useful to simply state the fact rather than drawing a conclusion in this manuscript.

5. Since the authors place emphasis on de- and rehydration (cf. 19.) I think appropriate references (not just one) should be cited.

We have cited two recent publications on *Physcomitrella* dehydration or desiccation tolerance (Koster et al. 2010. Plant Growth Regulation 62: 293-302; Xiao et al. 2017. Plant Cell Environ Epub ahead of print) (line 207). We removed the original citation because of the limit for the citations allowed.

6. As a side note, Prof. Reski mentioned that existing expression repositories might be tackled. The authors said this would not be possible due to the fact that those use v1.2 or v1.6 gene IDs. I would like to comment that this does not present a big problem, since gene model lookup is part of e.g. the browsers at Phytozome, cosmos and CoGe. Looking up three gene models for old versions is not a major effort.

We thank Dr. Rensing again for this suggestion. We have figured this out and located a corresponding gene [Phypa_151693 in cosmos V1.2 and Pp1S342_30V6.1 in cosmos V1.6; annotated as *Hr* (HHE domain protein)]. According to ePB browser, Phypa_151693 is up-regulated in gametophores, rhizoids, spores and archegonia. This gene is also highly expressed during dehydration and rehydration according to Genevestigator (see figures below), which is consistent with the data of Phytozome and our own RT-PCR experiments. We didn't add these data to the Supplementary Information because the annotation has already been changed in cosmos.

Reviewers' comments:

Reviewer #2 (Remarks to the Author):

All of my previously raised concerns have been addressed.

Reviewer #3 (Remarks to the Author):

I have entered my comments as RESPONSE in the authors letter. I also upload my Hemerythrin and Yan/Altyan alignments for the authors use.

Reviewer #3's comments

Major changes in this revision

We thank Drs. Ralf Reski, Stefan Rensing and the third reviewer for their comments and suggestions to improve this manuscript. We have carefully considered these comments and made corresponding changes. Some of the major changes are listed in the following.

1. E-value threshold for BLASTP of nr was changed to 1. This significant increase in the E-value threshold is somewhat usual, but it reflects our efforts to include more algal hits in our search. Fortunately, even with this usual threshold, the generated hits are overwhelmingly annotated as hemerythrins. We have provided this information in the manuscript (lines 69-72) and Supplementary Figure 1.

RESPONSE: Please re-read my comment 6. on the last version. There I pointed out that 1E-06 is already a comparatively HIGH value to decide on homology. I also pointed out that the E-value might not be appropriate to decide on homology. The authors now RAISED the E-value and argue that most of the detected hits are annotated as hemerythrin. I do not get this argument. I would have expected that either they LOWER the E-value to make sure that the detected hits are homologous, or to use more appropriate parameters like % identity and alignment length to determine homology. As it stands, nowhere in the manuscript it is mentioned how exactly homology is assigned.

2. Relationships of algal and land plant hemerythrins. We performed some additional comprehensive searches for hemerythrins in glaucophytes and green algae by employing higher E-value thresholds (see above) and using various queries and databases. We realize that no search can be exhaustive, but our searches only identified hemerythrins in the glaucophyte *Cyanophora paradoxa* and the green algal class Chlorophyceae. We didn't find conclusive evidence for hemerythrins in charophytes and other chlorophytes, but this issue remains to be thoroughly investigated. We have provided this information in the manuscript (lines 91-99) and Supplementary Notes 1 and 2. Additionally, we have also added information of sequence comparisons and phylogenetic re-analyses, which suggests that algal and land plant hemerythrins are distantly related (Supplementary Figure S2 and Supplementary Figure 4b).

RESPONSE: As I pointed out in 7. in my last review, I do have detected homologs from charophytes. I can make a file with the sequences available to the authors. It would be interesting to learn why they do not consider them homologs (compare with the comment above, it is not clear how they define homology). Moreover, neither their tree nor the tree that I did suggests that "algal and land plant hemerythrins are distantly related". They are not separated by a long branch.

3. The origin of *Hr* in land plants. Because of the existence of the hemerythrin gene in glaucophytes and green plants, it is somewhat tempting to speculate that the gene was transferred from fungi to the most recent common ancestor of Archaeplastida, followed by different gene losses. We have specifically discussed this scenario in the manuscript (lines 285-292). Overall, we do not favor this scenario because **a)** algal and land plant sequences are not particularly related based on our analyses, and **b)** major lineages of fungi are much younger than the ancestor of Archaeplastida. However, we also discussed alternative scenarios and cautioned that the explanation may change if other evidence from new data, particularly those related to glaucophytes and charophytes, becomes available in future (lines 294-298).

RESPONSE: In the light of the presented evidence, I do not see how one of the two scenarios is more likely than the other. Currently, it just cannot be resolved.

4. Results of spore germination have been added to the manuscript. These results were not included in our last submission because of the time limitation. We show that spores from both wild-type and mutant plants can germinate and grow into chloronemata (lines 250-252; Supplementary Figure 16).
5. Other changes such as these about citations, oil bodies etc. as suggested by the reviewers.

Response to comments by reviewer 1 (Dr. Ralf Reski)

Authors have done everything possible to improve their manuscript. They might wish to consider checking Genevestigator and/or ePB browser because gene accessions can easily be converted from V1.6 to V3.1 and V3.2. Congratulations to this exciting work which opens a new field in plant research.

We are grateful to Dr. Reski for his kind words and encouragement. The improvement was only made possible thanks to the insightful comments and suggestions of Drs. Reski, Rensing and the third reviewer.

Following the suggestions of Drs. Reski and Rensing, we were able to locate a corresponding gene (Phypa_151693 in cosmass V1.2 and Pp1S342_30V6.1 in cosmass V1.6). The gene was originally annotated as *Hr* (HHE domain protein) in V1.6 and the annotation was changed since cosmass V3.1. According to ePB browser, Phypa_151693 is up-regulated in gametophores, rhizoids, spores and archegonia. Additionally, the gene is up-regulated during dehydration and rehydration according to Genevestigator (see figures below), which is consistent with the data of Phytozome and our own RT-PCR experiments. We didn't include this information in the Supplementary Information because the annotation has already been changed in cosmass.

Response to comments by reviewer 2

1. A much improved version, just a few minor comments. While acknowledging that it may have been the original rationale for the study, the focus on early land plant evolution is not as relevant as the origin of new adaptive genes — e.g. the origin of YAN/AltYAN is not relevant to the origin of land plants, but rather some adaptation that arose in the Physcomitrium clade of mosses, relatively recently.

Thanks for this suggestion. In this revision, we rephrased the first sentence into the following: “*Early-diverging land plants (embryophytes) provide some unique opportunities to understand the mechanisms of plant adaptation to terrestrial environments*” (lines 36-37). Hopefully, this will shift the focus of the study from land plant evolution to the adaptation mechanisms. Because *Hr* is distributed in many early-diverging land plants, it is difficult to discuss YAN/AltYAN without touching the evolution of early land plants.

RESPONSE: Embryophyta equal land plants, therefore “Early-diverging land plants (embryophytes)” is non-sensical.

2. Lines 43-55: somewhat antiquated given the prevalence of CRISPR-Cas mediated gene modification.

This sentence is deleted.

3. Lines 55-63: these are really conclusions and could be reduced here to state that an origin of a novel gene was investigated.

We appreciate this suggestion. Many readers do not have sufficient time to read the entire article. A brief summary in the Introduction will allow these readers to understand the major findings of the study without reading the entire article. As such, we chose to keep these sentences in this revision.

4. Lines 128-129, line 280: might be better to state that the *Hr* gene (coding sequence) has evolved rather than been lost?

We changed the first part into “is either lost or has evolved into new functions in *P. patens*” (line 141). The word “lost” is kept here because the presence of a premature stop codon itself does not suggest gene evolution. We realized that the gene has evolved into YAN/AltYAN only after two transcripts from the same locus region were identified.

In the second part, we deleted “in *Physcomitrella*” so that the *Hr* gene loss only applies to seed plants (line 303).

5. Line 154-155: a paraphyletic group in which *Physcomitrella* is embedded?

Changed (line 167).

6. Line 314: it is not clear that the oil bodies in liverworts are 'critical' to dehydration resistance

— this seems to be based on a single circumstantial report. Also, there is still a confusion with the oil bodies in liverworts — they are not the same as the oil bodies in mosses or lycophytes, e.g. they do not have oleosin as a major component.

Thanks again for this comment on the role of liverwort oil bodies in desiccation resistance. In this revision, we removed sentences on oil bodies in liverworts. Since the oil bodies in liverworts and other land plants are different in structure, development and function, lumping them together in a discussion will only create confusion. We now write “*In mosses and many other land plants (e.g., seed plants), oil bodies play an important role in desiccation tolerance of plant tissues*” (lines 334-335). We realize this is still vague since the phrase “many other land plants” is not clearly defined. Nevertheless, this should keep the basic information correct.

Response to comments by reviewer 3 (Dr. Stefan Rensing)

1. Please note that a blast search per se will detect similarity, as opposed to homology. Using the E-value as cutoff is not an appropriate measure to decide whether a database hit is actually homologous to the query sequence. This is particularly true at relatively high values like the one used by the authors (1E-06). The authors are referred to Rost et al. 1999 Protein Eng, where appropriate combinations of identity and alignment length are described that allow to confidently determine homology based on blast results. This matter, unfortunately, is crucial for the paper, in particular since the conserved regions are relatively short and the evolutionary distances high. Along these lines, the default blast matrix that was used, BLOSUM62, might not be appropriate, since it assumes ca. 62% identity between query and hit - using e.g. BLOSUM45 might yield better results.

We thank Dr. Rensing for his comments on the relationships of sequence similarity and homology. Indeed, both Rost 1999 paper and BLAST incorporated sequence identity (or similarity) and length. The empirical rules proposed by Rost and E-values adopted by BLAST complement each other rather than being mutually exclusive [“*The thresholds for sequence identity and similarity defined here, complemented the levels for ‘significance’ provided by BLAST*” (Rost 1999. Protein Engineering 12:85-94)].

In our analyses, the conserved HHE domain is about 120 aa, which is relatively short for phylogenetic analyses, but still sufficient for the assessment of homology. The choice of E-value 1e-6 as the cutoff essentially reflects our consideration of sequence length in BLAST search, since it is not possible to yield high bit scores, thus low E-values, for more distant Hr homologs over 120 aligned amino acid pairs. This E-value cutoff, however, should not significantly affect our assessment of sequence homology. This can be evidenced by our BLASTP of the nr database even with a much higher E-value cutoff in this revision [see figure below, which shows part of the BLASP results using a *Marchantia* Hr sequence (JGI ID: Mapoly0042s0093) as query]. In this search, E-value = 1 was used as the cutoff (see our response to next comment) and BLOSUM45 was chosen as the substitution scoring matrix. Only the most dissimilar hits (i.e., those with the lowest bit scores and the highest E-values) from the search are shown. Almost all these hits are clearly annotated as hemerythrins. We have added the above information to the revised manuscript [“*These hits are overwhelmingly annotated as hemerythrins* (Supplementary Figure S1)” (line 72)].

RESPONSE: “The choice of E-value 1e-6 as the cutoff essentially reflects our consideration of sequence length in BLAST search, since it is not possible to yield high bit scores, thus low E-values, for more distant Hr homologs over 120 aligned amino acid pairs.” – can you please provide evidence/reasoning?

hemerythrin [Micromonospora echinospora]	40.3	40.3	43%	0.79	26%	WP_088984205.1
hemerythrin [Micromonospora inyonensis]	40.3	40.3	43%	0.81	26%	WP_091462030.1
hemerythrin HHE cation-binding protein [Streptomyces endus]	40.3	40.3	59%	0.81	26%	WP_067074069.1
hemerythrin [Streptomyces glaucescens]	40.6	40.6	46%	0.82	23%	WP_043505779.1
MULTISPECIES: hypothetical protein [Actinoplanes]	40.3	40.3	44%	0.83	32%	WP_014689342.1
hemerythrin [Mycobacterium marseillense]	40.3	40.3	58%	0.85	22%	WP_083019024.1
hemerythrin [Sphingomonas indica]	40.6	40.6	33%	0.85	30%	WP_085217127.1
hemerythrin [Sphingopyxis macroglabida]	40.3	40.3	42%	0.86	29%	WP_084758623.1
hemerythrin [Mycobacterium sp. JEC1808]	40.3	40.3	60%	0.87	25%	WP_085184681.1
hemerythrin HHE cation binding domain-containing protein [Mycobacterium sp. TS-1]	40.9	40.9	62%	0.88	21%	WP_023954761.1
Hemerythrin HHE cation binding domain-containing protein [Actinomadura echinospora]	40.3	40.3	62%	0.89	25%	SEG89047.1
hemerythrin [alpha proteobacterium LLX12A]	40.0	40.0	43%	0.89	25%	WP_017503021.1
hemerythrin [Sphingomonas sp. Y57]	40.0	40.0	43%	0.90	25%	WP_047166529.1
hemerythrin [Burkholderia multivorans]	39.8	39.8	34%	0.92	37%	WP_088923459.1
hypothetical protein [Streptomyces hygrosopicus]	40.3	40.3	59%	0.93	26%	WP_030830277.1
MULTISPECIES: hemerythrin [Sphingobium]	40.0	40.0	43%	0.93	25%	WP_069335624.1
Hemerythrin HHE cation binding region [Mycobacterium smegmatis str. MC2 155]	39.8	39.8	45%	0.95	28%	AFP39690.1
MULTISPECIES: hemerythrin [Streptomyces]	40.3	40.3	54%	0.96	22%	WP_031142329.1
hemerythrin [Mycobacterium avium]	40.3	40.3	58%	0.96	26%	WP_084022155.1
hemerythrin [Streptomyces sp. NRRL WC-3795]	40.3	40.3	60%	0.97	21%	WP_031019374.1
hemerythrin [Mycobacterium sp. 1245111.1]	40.3	40.3	58%	0.97	22%	WP_067331242.1
hemerythrin [Streptomyces rimosus subsp. pseudovercillatus]	40.0	40.0	45%	0.97	22%	KOT87501.1
hypothetical protein [Actinomadura madaurae]	40.3	40.3	57%	0.98	24%	WP_021593450.1
hemerythrin [Mycobacterium colombiense]	40.3	40.3	59%	1.00	23%	WP_077091455.1

We also concur with Dr. Rensing that BLOSUM45 is a better choice for BLAST search of the hemerythrin gene. With *Marchantia* hemerythrins as query, we compared the BLAST results using the two matrices. The generated hits were basically the same, though their bit scores and E-values were slightly different. In this revision, we indicate that BLOSUM45 was used (line 70).

2. Because I was worried that the analyses on which much of this paper relies were flawed I spent some time doing my own analyses. I used the *Marchantia Hr* sequence mentioned by the authors to search against a database of sequenced plant and algal genomes, 1KP bryophyte transcriptomes, and published charophyte and moss transcriptomes. A first glimpse at the resulting alignment and tree shows e.g. that there are homologs in charophyte algae (*Nitella hyalina*), in *Naegleria gruberi* (an amoeba), in *Volvox carteri* and *Chlamydomonas reinhardtii* (Chlorophyta), in *Cyanophora paradoxa* (Glaucophyta), and that the fungal sequences are nested in the plant sequences. Hence, the most parsimonious explanation for the distribution of the gene is that it was acquired (maybe by HGT from fungi) by the common ancestor of Archaeplastida and lost during vascular plant evolution (and maybe secondarily on some other lineages like red algae). Yet, a more detailed phylogenetic analysis would be in order to gain confidence.

We appreciate the comments of Dr. Rensing about the distribution of *Hr* in algae. In our last submission, we did identify *Hr* hits in green algae, including both chlorophytes and charophytes. The charophytes hits were only from 1KP, and the possibility that these sequences resulted from contamination was discussed in Supplementary Table S3 and Supplementary Note 1 (now Supplementary Note 2). We noted in our last submission that contamination is a serious issue for 1KP data. In this revision, we also searched the *Nitella hyalina* transcriptomic data at NCBI (<https://trace.ncbi.nlm.nih.gov/Traces/sra/?run=SRR064326>), but only two matches of 19-32 nucleotides were found; these two matches were mapped onto different regions (separated by about 470 nucleotides) of the *Hr* CDS (see figure below, upper panel). Search of the transcriptomic data of the charophyte Closterium peracerosum-strigosum-littorale complex at NCBI (<https://www.ncbi.nlm.nih.gov/bioproject?term=PRJNA296352>) provided a similar result (also see below, lower panel). We further searched the over 650 transcriptomes in the Marine Microbial Eukaryote Transcriptome Sequencing Project (MMETSP), and the results were largely consistent with our previous findings.

BLASTN search of transcriptomic data of charophytes *Nitella hyalina* (top panel) and *Closterium peracerosum-strigosum-littorale* complex (lower panel) at NCBI. Query is the CDS of *Marchantia polymorpha Hr* (JGI ID: Mapoly0042s0093).

We realize that the assessment of sequence homology is more difficult at the level of nucleotide sequences, and that lack of sufficient query coverage in the above search does not necessarily suggest the absence of *Hr* in the two charophytes. As such, we further investigated whether the 2-3 charophyte hits were specific to *Hr* (i.e., HHE domain). To this end, we performed the BLASTN search of the NCBI non-redundant nucleotide sequence database using the same *Marchantia polymorpha Hr* (JGI ID: Mapoly0042s0093) as query. Indeed, the results included hits corresponding to those from the two charophytes (*Nitella hyalina* and *Closterium peracerosum-strigosum-littorale* complex) (see figure below, top panel). However, further inspections of these hits indicated that they were not particular to *Hr*. For instance, the hits to the 670-600 bp region of the query were annotated as part of the genes encoding small integral

membrane protein 10-like protein 2A, peroxidase 7-like protein, and myomegalin-like protein; they were found in both flowering plants (*Lupinus angustifolius*, *Vitis venifera*, and *Cucurbita maxima*) and animals (*Crocodylus porosus*, *Oncorhynchus mykiss*, and *Columba livia*). Similarly, hits to the 150-200 bp region of the query were annotated as genes encoding erythrocyte membrane protein, RP1 like 1 (rp111) protein, and retrotransposon Gag like 5 (Rtl5) protein; they could be found in animals (e.g., *Oryzias latipes*, *Heterocephalus glaber*, *Labrus bergylta*) and apicomplexan parasites (*Plasmodium falciparum*). On the other hand, the most significant hits, which also had the longest query coverages, were from *Selaginella moellendorffii*, the fungus *Fusarium verticillioides* and *Physcomitrella patens* (see figure below, lower panel). The former two (*Selaginella* and *Fusarium*) were part of the *Hr* gene, whereas the later (i.e. *P. patens*) evolved directly from *Hr*. These data suggest that the hits from the two charophytes (*Nitella hyalina* and *Closterium peracerosum-strigosum-littorale* complex) might be associated with genes other than *Hr*. In addition, it also likely points to the close relationship between fungal and land plant hemerythrins. Nevertheless, this issue of *Hr* in charophytes remains to be thoroughly investigated, and we cautioned in this manuscript that the conclusion may change if new data from glaucophytes and charophytes become available in future (lines 296-298).

Because complete genome sequence data have longer scaffolds, they are more reliable sources for assessing the existence of a gene in a given genome. As such, our search also relies heavily on complete genome sequence data. Our initial BLAST search of the nr database, which contains annotated protein sequences of the vast majority of published algal genomes, adopted an E-value cutoff $1e-6$. This cutoff is sufficient in most searches and identified sequences from chlorophyte *Gonium pectoral* and *Monoraphidium neglectum*. We also performed various other searches of both nr and Phytozome; these searches indeed identified sequences from additional chlorophytes such as *Volvox carteri* and *Chlamydomonas reinhardtii*, but this would require using a much higher E-value cutoff or employing chlorophyte *Gonium* or *Monoraphidium* sequences as query. All these green algae (*Gonium*, *Monoraphidium*, *Volvox*, *Chlamydomonas*, and *Chromochloris*) belong to the class Chlorophyceae. We didn't identify sequences from charophytes and other chlorophytes (admittedly, not many charophyte genomes have been sequenced; the NCBI charophyte *Spirogyra sp.* AU1 BioProject site does not appear to be functional and no data can be

accessed). We have added the information on *Hr* distribution in other Chlorophyceae in this revision (lines 70-77). To reflect the change in search results, we have changed the E-value cutoff from 1e-6 to 1 in this revision. Although this high E-value cutoff is somewhat unusual for most searches, fortunately it does not have a major effect on our identification of *Hr* homologs (Supplementary Fig. S1; also see our reply to comment 1). Furthermore, information of pairwise comparisons has also been included, showing that land plants Hrs share higher sequence percent identities with fungal homologs than with chlorophyte and glaucophyte sequences (Supplementary Fig. S2).

In terms of the scenario that the *Hr* gene was transferred from fungi to the most recent common ancestor of Archaeplastida, we have carefully weighted different lines of evidence. Overall, we do not favor this scenario, particularly for the following reasons:

a. As indicated above, green algal hits were also found in our initial analyses. We performed very comprehensive analyses of these algal genes using various queries and databases. Both sequence similarity comparisons and phylogenetic analyses suggest that these algal sequences are not particularly related to land plant hemerythrins. These analyses were detailed in our last submission (now lines 66-90; now Supplementary Figure S4; Supplementary Tables 1-3; now Supplementary Note 2). Results of our additional phylogenetic analyses in this revision are consistent with our earlier conclusions. As shown in Supplementary Figure S4b, sequences from two chlorophycean (*Gonium* and *Volvox*) are only distantly related to land plant homologs (the glaucophyte *Cyanophora* sequence was removed from the analyses because of its much shorter length; inclusion of *Cyanophora* sequences in the analyses provided a similar topology with lower branch support).

b. Given the limited genome data for glaucophytes, the distribution *Hr* in this group remains to be seriously investigated. However, the seeming restriction of *Hr* to Chlorophyceae in green algae would require multiple loss events for a scenario of HGT to the ancestor of Archaeplastida. Although loss of *Hr* does happen (e.g., in most vascular plants), the more loss events postulated, the less likely the scenario. On the other hand, an independent HGT to Chlorophyceae is at least an equally parsimonious explanation.

c. When assessing the occurrence and direction of gene transfer events, an important consideration is the temporal sequence in which the donor and recipient evolved. The donor must evolve no later than the recipient (Huang and Gogarten 2006. Trends in Genetics 22: 361-366). In the specific scenario of fungi-to-Archaeplastida gene transfer, Archaeplastida evolved about 1400 MYA (Hedges and Kumar eds 2009. The Time Tree of Life; hereafter). Although fungi split from other opisthokonts about the same time, major fungal lineages only evolved 980-1150 MYA. This much younger age of major fungal lineages suggests that horizontal transfer of the *Hr* gene from fungi to the most recent common ancestor of Archaeplastida is an unlikely scenario. We have discussed this issue in this revision (lines 284-292)

We also would like to note here that the determination of HGT is based on overall evidence. For every HGT event proposed, there are multiple other explanations. In particular, differential gene loss is always an alternative explanation to HGT. As indicated above, the more loss events to be postulated, the less likely the gene loss scenario. Specifically for *Hr*, it is unrealistic to expect a well-resolved phylogeny because of the short length of HHE domain, which in turn translates into difficulties in assessing the origin of *Hr* in algae. The suggestion of HGT from fungi to the ancestor of land plants is based on our assessment of gene distribution, phylogeny, and sequence similarity. As new methods or sequence data become available, it is possible that the conclusion (or explanation) will change. This is exactly the reason that other possible explanations (including gene loss) were included in our last and current submissions (lines 294-296).

3. Since I considered it quite unlikely that the Yan/AltYan gene was acquired by the Physcomitrium-Physcomitrella species complex I also conducted an analysis in this regard, using

the PpYan and AltYan sequences for a similar approach as mentioned under 8. While the authors describe only a single hit outside Physcomitrella, namely in a 1KP Physcomitrium sequence, my searches also found hits in Encalypta streptocarpa from 1KP, in the published transcriptome data of Ceratodon purpureus and Funaria hygrometrica (Szovenyi et al. 2014 and 2010), and in Carica papaya. While the latter might be a contamination, the fact that the gene can be detected not only in other Funariaceae, but also in Ceratodon, suggests that it was acquired a lot earlier, maybe even in a common ancestor of all Bryopsida. The lack of 1KP evidence might be due to the fact that the genes are not strongly expressed in the developmental stages from which the samples were generated.

We are sorry for the confusion around the distribution of YAN and AltYAN. In our last submission, we wrote “In particular, the search of 1KP for AltYAN provided no hit outside Physcomitrium; only a single hit (1KP ID: YEPO-2062682), which corresponded to the 5’UTR and the first exon of AltYAN, was identified in Physcomitrium” (now lines 168-171). The single hit in our search only applies to AltYAN (see figure below).

```

Query= Pp3c21_19720V3.1 AA (AltYAN peptide)
Length=123
Sequences producing significant alignments:
      scaffold-YEPO-2062682-cf._Physcomitrium_sp.          54.3   2e-07
> scaffold-YEPO-2062682-cf._Physcomitrium_sp.
Length=315
Score = 54.3 bits (129), Expect = 2e-07, Method: Compositional matrix adjust.
Identities = 43/47 (91%), Positives = 44/47 (94%), Gaps = 0/47 (0%)
Frame = -1
Query 1   MAAVAYIPLSAVASARLattqasssnaasqpsaGIVAFKRAVTPSCL 47
          MAAVAYIPL+AVASARLATTQA SNAASQPSAGIVAFKRA TPSCL
Sbjct 141 MAAVAYIPLNAVASARLATTQARFSNAASQPSAGIVAFKRAATPSCL 1

```

On the other hand, because *Hr* and YAN partially share the conserved HHE region (same nucleotide sequence, but different genes) and because *Hr* is found in many bryophytes, it is not surprising that hits will be found in other bryophytes when YAN is used as query. This has been discussed in our last submission (now lines 167-168: “No homolog was identified in nr whereas only fragmented sequence matches, often with premature stop codons, were found in 1KP”). That said, these hits most likely do not represent intact YAN; they most likely are matches to *Hr* instead of YAN.

During this revision, we carefully performed re-analyses of the distribution of YAN and AltYAN. Results of TBLASTN search of 1KP and nr databases were consistent with the findings in our last submission. Only one hit of AltYAN was identified in Physcomitrium in 1KP (see figure above); there were hits of YAN in other bryophytes (including Encalypta streptocarpa and Ceratodon purpureus), but these hits contain premature stop codons, indicating they are not YAN (see figures below). Even if YAN exists in other bryophytes, it does not automatically suggest the existence the dual-coding gene YAN/AltYAN. It will likely support our speculation that “YAN might have initially evolved as an alternative transcript of *Hr*, and their overlapping in coding regions led to functional linkage” (now lines 398-399). The suggestion of YAN/AltYAN existence requires finding both transcripts from the same genome.

RESPONSE: I will happily provide the sequences I found to the authors for their evaluation. They are the experts to check which gene they represent.

> scaffold-FFPD-2009361-Ceratodon_purpureus
 Length=1258

Score = 105 bits (263), Expect = 2e-23, Method: Compositional matrix adjust.
 Identities = 90/189 (48%), Positives = 108/189 (57%), Gaps = 4/189 (2%)
 Frame = +2

```

Query 60 TVHHGRSRIHPLERCRFREVGNHPSQLQQCGLPAQRGHCCVQASCHAILPAVGEPVLRPP 119
          HHGR + R P +L QCG AQ + C + A G+ L R
Sbjct 95 IFHHGRFSMRSTHRGSHELELAGPCEL*QCGPAAQLPGVEEELHCRVVCSAFGQSFWRSS 274

Query 120 RHFLaaaaaaaaPGREETGRVDOMLRHSEEFGGGRRHHRQSEAGPQAGGGVLQEQVPOA 179
          R L + EE + H + G R HHRQ EAGPQAGGG+LQEQVPOA
Sbjct 275 RDILGICCKG----EERRWIRPGHGHROHGEGRRGHHRQSEAGPQAGGGIQLQEQVPOA 442

Query 180 GQRGRGRIMVQPVVVDLSPCRHRGAGVVPVDRVAGROGPEAGGPVARRAPEDEGHAGGD 239
          G+RG G+ +VQP+RVG + P R GA VPVDR+AG +GPEAGGPVAR AP + HAGGD
Sbjct 443 GRRGGGQVQVQPIRVGGVPLRDGGARVPVDRIAGGEGPEAGGPVARGAPGGQEHAGGD 622

Query 240 PGHRGRRPV 248
          PG + RPV
Sbjct 623 PGDQRPRPV 649
  
```

> scaffold-KEFD-2005771-Encalypta_streptocarpa
 Length=926

Score = 52.4 bits (124), Expect = 9e-05, Method: Compositional matrix adjust.
 Identities = 57/147 (39%), Positives = 75/147 (51%), Gaps = 9/147 (6%)
 Frame = -1

```

Query 99 CVOASCHAT--LPAVGEPVLRPRRHFLaaaaaaaaPGREETGRVDOMLRHSEEFGGGRRHH 156
          C+Q + L A+ + LRR +FL+ Q G +E M G RRH+
Sbjct 698 CIQEESDGLFFLSAISKAFLLRSSYFLRLHCQ---GIQEWL--SPMHGQQGHKGRRRHY 534

Query 157 RQSEAGPQAGGGVLQEQVPOAGQRGRGRIMVQPVVVDLSPCRHRGAGVVPVDRVAGR 216
          RQ A P VLQL EVPQ GQR R +VQP+RVG+L P RHRG +P+D
Sbjct 533 RQD*ARPXXA--VLQL*EVPOGGQRRRR*VVQPIRVGNLPPRRHRGTRPLPLDWFH*H 360

Query 217 QGPEAGGPVARRAPEDEGHAGGDPGHR 243
          G E G + A +++G G DPGH+
Sbjct 359 *GQEFG**IPCGASDNOGLVGRDPGHQ 279
  
```

Following the comments of Dr. Rensing, we also downloaded the original transcriptomic data generated by Szovenyi et al. for *Ceratodon purpureus* and *Funaria hygrometrica*. Raw reads of *Funaria hygrometrica* were also assembled using Trinity. We then performed BLAST search, both BLASTN and TBLASTN (E-value cutoff = 10), of the downloaded sequences for *YAN* and *AltYAN*. No hits of *AltYAN* were found in *Ceratodon purpureus* and only a match of 12 amino acids was found in *Funaria hygrometrica*. Consistent with the TBLASTN results of 1KP data, hits of *YAN* were found in *Ceratodon purpureus*, but they contain premature stop codons. These BLAST results and alignments are shown in the following, where the prefixes isotig and TRINITY_DN indicate sequences from *Ceratodon purpureus* and *Funaria hygrometrica*, respectively.

a. BLAST result of *Ceratodon purpureus* transcriptomic data for *AltYAN*. Results of BLASTN indicate that the lowest E-value is 2.4 and the highest bit score is 32.2. TBLASTN provides no hits.

```

# BLASTN 2.2.26 [Sep-21-2011]
# Query: Pp3c21_19720V3.1 CDS PpAltYAN
# Database: R40_454Isotigs.fna
# Fields: Query id, Subject id, % identity, alignment length, mismatches, gap openings, q. start, q. end, s. start, s. end, e-value, bit score
Pp3c21_19720V3.1 isotig15492 100.00 16 0 0 268 283 281 266 2.4 32.2
Pp3c21_19720V3.1 isotig14853 95.00 20 1 0 271 290 61 80 2.4 32.2
Pp3c21_19720V3.1 isotig09623 100.00 16 0 0 267 282 378 393 2.4 32.2
Pp3c21_19720V3.1 isotig03531 100.00 16 0 0 310 325 3464 3449 2.4 32.2
Pp3c21_19720V3.1 isotig03530 100.00 16 0 0 310 325 3487 3472 2.4 32.2
Pp3c21_19720V3.1 isotig01478 100.00 16 0 0 271 286 2158 2143 2.4 32.2
Pp3c21_19720V3.1 isotig01477 100.00 16 0 0 271 286 2108 2093 2.4 32.2
Pp3c21_19720V3.1 isotig01476 100.00 16 0 0 271 286 2169 2154 2.4 32.2
Pp3c21_19720V3.1 isotig01475 100.00 16 0 0 271 286 2119 2104 2.4 32.2
Pp3c21_19720V3.1 isotig19152 100.00 15 0 0 16 30 246 260 9.4 30.2
Pp3c21_19720V3.1 isotig14849 100.00 15 0 0 346 360 518 532 9.4 30.2
Pp3c21_19720V3.1 isotig11812 100.00 15 0 0 272 286 1351 1365 9.4 30.2
Pp3c21_19720V3.1 isotig10874 100.00 15 0 0 30 44 1260 1246 9.4 30.2
Pp3c21_19720V3.1 isotig09042 100.00 15 0 0 97 111 22 8 9.4 30.2
Pp3c21_19720V3.1 isotig09021 100.00 15 0 0 271 285 332 346 9.4 30.2
Pp3c21_19720V3.1 isotig08926 100.00 15 0 0 36 50 1629 1615 9.4 30.2
Pp3c21_19720V3.1 isotig08824 100.00 15 0 0 19 33 432 446 9.4 30.2
Pp3c21_19720V3.1 isotig08685 100.00 15 0 0 305 319 690 704 9.4 30.2
Pp3c21_19720V3.1 isotig08492 100.00 15 0 0 34 48 1532 1546 9.4 30.2
Pp3c21_19720V3.1 isotig07918 100.00 15 0 0 269 283 14 28 9.4 30.2
Pp3c21_19720V3.1 isotig07892 100.00 15 0 0 262 276 1512 1526 9.4 30.2
Pp3c21_19720V3.1 isotig07859 100.00 15 0 0 269 283 3547 3533 9.4 30.2
Pp3c21_19720V3.1 isotig05481 100.00 15 0 0 157 171 673 659 9.4 30.2
Pp3c21_19720V3.1 isotig05480 100.00 15 0 0 157 171 673 659 9.4 30.2
Pp3c21_19720V3.1 isotig03961 100.00 15 0 0 70 84 324 338 9.4 30.2
Pp3c21_19720V3.1 isotig03960 100.00 15 0 0 70 84 324 338 9.4 30.2

```

BLASTN 2.2.26 [Sep-21-2011]

Reference: Altschul, Stephen F., Thomas L. Madden, Alejandro A. Schaffer, Jinghui Zhang, Zheng Zhang, Webb Miller, and David J. Lipman (1997), "Gapped BLAST and PSI-BLAST: a new generation of protein database search programs", *Nucleic Acids Res.* 25:3389-3402.

Reference for compositional score matrix adjustment: Altschul, Stephen F., John C. Wootton, E. Michael Gertz, Richa Agarwala, Aleksandr Morgulis, Alejandro A. Schaffer, and Yi-Kuo Yu (2005) "Protein database searches using compositionally adjusted substitution matrices", *FEBS J.* 272:5101-5109.

Query= Pp3c21_19720V3.1 AltYAN
(122 letters)

Database: R40_454Isotigs.fna
20,431 sequences; 34,112,501 total letters

Searching..... done

***** No hits found *****

Database: R40_454Isotigs.fna
Posted date: Nov 27, 2017 5:46 PM
Number of letters in database: 34,112,501
Number of sequences in database: 20,431

b. BLAST result of *Funaria hygrometrica* transcriptomic data for *AltYAN*. Results of BLASTN indicate that the lowest E-value is 0.96 and the highest bit score is 32.2. TBLASTN provides a single hit of only 12 aa.

```
# BLASTN 2.2.26 [Sep-21-2011]
# Query: Pp3c21_19720V3.1 CDS PpAltYAN
# Database: Trinityall.fasta
# Fields: Query id, Subject id, % identity, alignment length, mismatches, gap openings, q. start, q. end, s. start, s. end, e-value, bit score
Pp3c21_19720V3.1 TRINITY_DN5722_c0_g1_i2 100.00 16 0 0 289 304 219 204 0.96 32.2
Pp3c21_19720V3.1 TRINITY_DN4289_c143_g1_i1 100.00 15 0 0 0 279 293 21 35 3.8 30.2
Pp3c21_19720V3.1 TRINITY_DN9083_c0_g1_i1 100.00 15 0 0 221 235 1681 1667 3.8 30.2
Pp3c21_19720V3.1 TRINITY_DN3266_c0_g1_i1 100.00 15 0 0 208 222 294 308 3.8 30.2
Pp3c21_19720V3.1 TRINITY_DN5454_c0_g2_i4 100.00 15 0 0 113 127 384 398 3.8 30.2
Pp3c21_19720V3.1 TRINITY_DN5454_c0_g2_i3 100.00 15 0 0 113 127 411 425 3.8 30.2
Pp3c21_19720V3.1 TRINITY_DN5454_c0_g2_i2 100.00 15 0 0 113 127 346 360 3.8 30.2
Pp3c21_19720V3.1 TRINITY_DN5454_c0_g2_i1 100.00 15 0 0 113 127 134 148 3.8 30.2
Pp3c21_19720V3.1 TRINITY_DN5537_c0_g1_i3 100.00 15 0 0 328 342 241 227 3.8 30.2
Pp3c21_19720V3.1 TRINITY_DN5537_c0_g1_i2 100.00 15 0 0 328 342 375 361 3.8 30.2
Pp3c21_19720V3.1 TRINITY_DN5537_c0_g1_i1 100.00 15 0 0 328 342 375 361 3.8 30.2
```

```
TBLASTN 2.2.26 [Sep-21-2011]
Reference: Altschul, Stephen F., Thomas L. Madden, Alejandro A. Schaffer,
Jinghui Zhang, Zheng Zhang, Webb Miller, and David J. Lipman (1997),
"Gapped BLAST and PSI-BLAST: a new generation of protein database search
programs", Nucleic Acids Res. 25:3389-3402.
```

```
Reference for compositional score matrix adjustment: Altschul, Stephen F.,
John C. Wootton, E. Michael Gertz, Richa Agarwala, Aleksandr Morgulis,
Alejandro A. Schaffer, and Yi-Kuo Yu (2005) "Protein database searches
using compositionally adjusted substitution matrices", FEBS J. 272:5101-5109.
```

```
Query= Pp3c21_19720V3.1 AltYAN
(122 letters)
```

```
Database: Trinityall.fasta
25,144 sequences; 13,914,117 total letters
```

```
Searching.....done
```

```
Sequences producing significant alignments:
Score E
(bits) Value
```

```
TRINITY_DN6098_c0_g2_i1 len=656 path=[379:0-24 381:25-74 431:75-... 25 4.2
```

```
>TRINITY_DN6098_c0_g2_i1 len=656 path=[379:0-24 381:25-74 431:75-86
443:87-124 481:125-156 513:157-337 694:338-359
716:360-389 746:390-396 753:397-416 773:417-487
844:488-507 864:508-611 968:612-631 988:632-655] [-1,
379, 381, 431, 443, 481, 513, 694, 716, 746, 753, 773,
844, 864, 968, 988, -2]
```

```
Length = 656
Score = 25.4 bits (54), Expect = 4.2, Method: Compositional matrix adjust.
Identities = 10/22 (45%), Positives = 14/22 (63%)
Frame = +2
```

```
Query: 100 RAAIVTWWMSVGGNCAVLPAGH 121
R I TW ++ G C++L AGH
Sbjct: 476 RPRISTWLLADSGKCSMLRAGH 541
```

```
Database: Trinityall.fasta
Posted date: Nov 29, 2017 3:25 PM
Number of letters in database: 13,914,117
Number of sequences in database: 25,144
```

c. BLAST result of *Ceratodon purpureus* transcriptomic data for YAN. BLASTN search shows two significant matches to YAN. However, both sequences contain premature stop codons, as shown in the TBLASTN results.

```

# BLASTN 2.2.26 [Sep-21-2011]
# Query: Pp3c21_19720V3.2 CDS PpYAN
# Database: R40_454Isotigs.fna
# Fields: Query id, Subject id, % identity, alignment length, mismatches, gap openings, q. start, q. end, s. start, s. end, e-value, bit score
Pp3c21_19720V3.2 isotig02033 87.12 295 38 0 450 744 359 653 8e-076 283
Pp3c21_19720V3.2 isotig15024 84.00 275 42 2 450 723 796 523 9e-045 180
Pp3c21_19720V3.2 isotig15598 92.45 53 4 0 563 615 577 525 1e-012 73.8
Pp3c21_19720V3.2 isotig18683 100.00 18 0 0 507 524 543 560 0.32 36.2
Pp3c21_19720V3.2 isotig11960 100.00 18 0 0 700 717 479 496 0.32 36.2
Pp3c21_19720V3.2 isotig15908 95.24 21 1 0 721 741 880 900 1.3 34.2
Pp3c21_19720V3.2 isotig15908 100.00 16 0 0 490 505 455 440 4.9 32.2
Pp3c21_19720V3.2 isotig13658 95.24 21 1 0 646 666 972 952 1.3 34.2
Pp3c21_19720V3.2 isotig00707 100.00 17 0 0 438 454 438 422 1.3 34.2
Pp3c21_19720V3.2 isotig00704 100.00 17 0 0 438 454 1724 1708 1.3 34.2
Pp3c21_19720V3.2 isotig00703 100.00 17 0 0 438 454 2598 2582 1.3 34.2
Pp3c21_19720V3.2 isotig17508 100.00 16 0 0 490 505 601 616 4.9 32.2
Pp3c21_19720V3.2 isotig16419 100.00 16 0 0 584 599 375 390 4.9 32.2
Pp3c21_19720V3.2 isotig14953 100.00 16 0 0 607 622 199 214 4.9 32.2
Pp3c21_19720V3.2 isotig14856 100.00 16 0 0 519 534 484 499 4.9 32.2
Pp3c21_19720V3.2 isotig14349 100.00 16 0 0 687 702 467 452 4.9 32.2
Pp3c21_19720V3.2 isotig14087 100.00 16 0 0 494 509 1104 1089 4.9 32.2
Pp3c21_19720V3.2 isotig12533 95.00 20 1 0 157 176 182 163 4.9 32.2
Pp3c21_19720V3.2 isotig12247 100.00 16 0 0 653 668 457 442 4.9 32.2
Pp3c21_19720V3.2 isotig11523 100.00 16 0 0 441 456 1717 1732 4.9 32.2
Pp3c21_19720V3.2 isotig11417 100.00 16 0 0 491 506 1760 1745 4.9 32.2
Pp3c21_19720V3.2 isotig10538 100.00 16 0 0 161 176 1061 1046 4.9 32.2
Pp3c21_19720V3.2 isotig06562 100.00 16 0 0 695 710 1317 1332 4.9 32.2
Pp3c21_19720V3.2 isotig02185 100.00 16 0 0 173 188 436 421 4.9 32.2
Pp3c21_19720V3.2 isotig02184 100.00 16 0 0 173 188 436 421 4.9 32.2
Pp3c21_19720V3.2 isotig01623 100.00 16 0 0 707 722 615 600 4.9 32.2
Pp3c21_19720V3.2 isotig01622 100.00 16 0 0 707 722 750 735 4.9 32.2

```

BLASTN 2.2.26 [Sep-21-2011]

Reference: Altschul, Stephen F., Thomas L. Madden, Alejandro A. Schaffer, Jinghui Zhang, Zheng Zhang, Webb Miller, and David J. Lipman (1997), "Gapped BLAST and PSI-BLAST: a new generation of protein database search programs", *Nucleic Acids Res.* 25:3389-3402.

Reference for compositional score matrix adjustment: Altschul, Stephen F., John C. Wootton, E. Michael Gertz, Richa Agarwala, Aleksandr Morgulis, Alejandro A. Schaffer, and Yi-Kuo Yu (2005) "Protein database searches using compositionally adjusted substitution matrices", *FEBS J.* 272:5101-5109.

Query= Pp3c21_19720V3.2 YAN
(248 letters)

Database: R40_454Isotigs.fna
20,431 sequences; 34,112,501 total letters

Searching..... done

Sequences producing significant alignments:	Score (bits)	E Value
isotig15024 gene=isogroup10253 length=1137 numContigs=1	104	5e-026
isotig02033 gene=isogroup00267 length=1306 numContigs=3	101	7e-025
isotig15598 gene=isogroup10827 length=1054 numContigs=1	61	6e-011
isotig07822 gene=isogroup03051 length=4146 numContigs=1	28	5.9

>isotig15024 gene=isogroup10253 length=1137 numContigs=1
Length = 1137

Score = 104 bits (259), Expect = 5e-026, Method: Compositional matrix adjust.
Identities = 73/165 (44%), Positives = 90/165 (54%), Gaps = 1/165 (0%)
Frame = -1

Query: 78 EVGNHPSQLQQCLPAQRGHC-CVQASCHAILPAVGEPVLRPRHFLXXXXXXXXPGREET 136
E + LQ+C +PA R C Q C + +P+ R HFL ++
Sbjct: 996 EYNSELHSLQRC-IPATRIQLHCTQVQCEL---EIRQPIRWRSSHFLKLLSCQGIQKQWP 829

Query: 137 GRVDQLRHSEEFGGRRHRQSEAGPQGAGGVLQLQEVPQAGRGRGRIMVQPVVVD 196
+ M H E G R HHRQ EAGPQGAGG +LQLQE+PQA QR R +VQP+RVG
Sbjct: 828 ---NAMHGHE*ECTGRR+HHRQGEAGPQGAGGSLQLQEIPQAEQRRRSG+VVQPIRVGS 658

Query: 197 LSPCRHRGAGVVPVDRVAGRQGPVARRAPEDEGHAGGDPG 241

```

>isotig02033  gene=isogroup00267  length=1306  numContigs=3
      Length = 1306

Score = 101 bits (252), Expect = 7e-025, Method: Compositional matrix adjust.
Identities = 89/189 (47%), Positives = 106/189 (56%)
Frame = +3

Query: 60  TVHHGRSRIHPLERCRFRFVGNHPSQLQQCGLPAQRGHCCVQASCHAILPAVGEPVLRFP 119
           HHGR + R P +L QCG AQ + + A G+ L R
Sbjct: 99  IFHHGRFMSMRSTHRGSHELELAGPCEL*QCGPAAQLPGVEELHRRVVCSAFGQSFLWRS 278

Query: 120  RHFLXXXXXXXXPGREETGRVDQMLRHSEEFGGGRRHHRQSEAGPQGAGGGVLQLQEVPPA 179
           R L EE + H + G R HHRQ EAGPQGAGGG+LQLQEVPPA
Sbjct: 279  RDILGICCKG----EERRWIRPQGHRQHGEGRRGHHRQGEAGPQGAGGGILQLQEVPPA 446

Query: 180  GQRGRGRIMVQPVVVDLSPCRHRGAGVVPVDRVAGROGPEAGGPVARRAPEDEGHAGGD 239
           G+RG G+ +VQP+RVG + P R GA VPVDR+AG +GPEAGGPVAR AP + HAGGD
Sbjct: 447  GRRGGGQQVQPIRVGGVPLRDGGARPVVPVDRIAGGEGPEAGGPVARGAPGGQEHAGGD 626

Query: 240  PGHRGRRPV 248
           PG + RPV
Sbjct: 627  PGDQRPRPV 653

```

d. TBLASTN result of *Funaria hygrometrica* transcriptomic data for YAN, indicating that no hits were found.

```

[BLASTN 2.2.26 [Sep-21-2011]

Reference: Altschul, Stephen F., Thomas L. Madden, Alejandro A. Schaffer,
Jinghui Zhang, Zheng Zhang, Webb Miller, and David J. Lipman (1997),
"Gapped BLAST and PSI-BLAST: a new generation of protein database search
programs", Nucleic Acids Res. 25:3389-3402.

Reference for compositional score matrix adjustment: Altschul, Stephen F.,
John C. Wootton, E. Michael Gertz, Richa Agarwala, Aleksandr Morgulis,
Alejandro A. Schaffer, and Yi-Kuo Yu (2005) "Protein database searches
using compositionally adjusted substitution matrices", FEBS J. 272:5101-5109.

Query= Pp3c21_19720V3.2  YAN
      (248 letters)

Database: Trinityall.fasta
      25,144 sequences; 13,914,117 total letters

Searching..... done

***** No hits found *****

Database: Trinityall.fasta

```

4. The fact that there is a gene in *Physcomitrium* at least allows to mention that it is older than the *Physcomitrium-Physcomitrella* species complex (see million year data in McDaniel et al.), but see comment above.

We might have missed something here. The gene is present in an unspecified species of *Physcomitrium*. Whether this species is closely related to individual subspecies of *Physcomitrella patens* or other *Physcomitrium* species remains to be investigated. We think it is useful to simply state the fact rather than drawing a conclusion in this manuscript.

5. Since the authors place emphasis on de- and rehydration (cf. 19.) I think appropriate references (not just one) should be cited.

We have cited two recent publications on *Physcomitrella* dehydration or desiccation tolerance (Koster et al. 2010. *Plant Growth Regulation* 62: 293-302; Xiao et al. 2017. *Plant Cell Environ* Epub ahead of print) (line 207). We removed the original citation because of the limit for the citations allowed.

6. As a side note, Prof. Reski mentioned that existing expression repositories might be tackled. The authors said this would not be possible due to the fact that those use v1.2 or v1.6 gene IDs. I would like to comment that this does not present a big problem, since gene model lookup is part of e.g. the browsers at Phytozome, cosmos and CoGe. Looking up three gene models for old versions is not a major effort.

We thank Dr. Rensing again for this suggestion. We have figured this out and located a corresponding gene [Phypa_151693 in cosmos V1.2 and Pp1S342_30V6.1 in cosmos V1.6; annotated as *Hr* (HHE domain protein)]. According to ePB browser, Phypa_151693 is up-regulated in gametophores, rhizoids, spores and archegonia. This gene is also highly expressed during dehydration and rehydration according to Genevestigator (see figures below), which is consistent with the data of Phytozome and our own RT-PCR experiments. We didn't add these data to the Supplementary Information because the annotation has already been changed in cosmos.

RESPONSE: What is the meaning of “the annotation has already been changed in cosmos”?

Responses to comments of Dr. Stefan Rensing

We thank Dr. Rensing for his meticulous comments and his patience with us. During this revision, we noticed that 1KP had recently gone through major changes, and we therefore performed re-analyses of the 1KP data. Our general findings and conclusions remain unchanged.

1. Please re-read my comment 6. on the last version. There I pointed out that 1E-06 is already a comparatively HIGH value to decide on homology. I also pointed out that the E-value might not be appropriate to decide on homology. The authors now RAISED the E-value and argue that most of the detected hits are annotated as hemerythrin. I do not get this argument. I would have expected that either they LOWER the E-value to make sure that the detected hits are homologous, or to use more appropriate parameters like % identity and alignment length to determine homology. As it stands, nowhere in the manuscript it is mentioned how exactly homology is assigned.

We provided a detailed explanation on this issue in our response letter of last submission. Please also see our additional explanation below on comment 5. If remote algal homologs are to be included in our analyses, it is necessary to adopt a less stringent E-value threshold.

2. As I pointed out in 7. in my last review, I do have detected homologs from charophytes. I can make a file with the sequences available to the authors. It would be interesting to learn why they do not consider them homologs (compare with the comment above, it is not clear how they define homology). Moreover, neither their tree nor the tree that I did suggests that “algal and land plant hemerythrins are distantly related”. They are not separated by a long branch.

We thank Dr. Rensing for sharing his data with us. As we detailed in our previous response letters and Supplementary Notes 1 and 2, we indeed detected homologous sequences from charophytes and other groups (e.g., seed plants). Although we suggested that algal sequences are distantly related to land plant hemerythrins (we did, however, indicate that contamination is a concern for charophyte sequences from 1KP), this by no means should lead to a conclusion that charophyte homologs do not exist or are ignored in our study. On the contrary, we specifically wrote the following sentence in our last submission: “It should also be cautioned that, as evidence from new data, particularly those about glaucophytes and **charophytes**, becomes available, the explanation for the origin of Hr in land plants might also change” (now lines 298-300).

In this revision, we changed “distantly related” to “not closely related” or “not particularly related”.

3. In the light of the presented evidence, I do not see how one of the two scenarios is more likely than the other. Currently, it just cannot be resolved.

We agree with Dr. Rensing here in principle. As we indicated in our last response letter, “For every HGT event proposed, there are multiple other explanations”. Alternative scenarios have been discussed in both previous and current submissions.

4. Embryophyta equal land plants, therefore “Early-diverging land plants (embryophytes)” is non-sensical.

Changed to “Early-diverging lineages of land plants (embryophytes)” (line 36). Embryophytes here refers to land plants, instead of early-diverging land plants.

5. “The choice of E-value 1e-6 as the cutoff essentially reflects our consideration of sequence length in BLAST search, since it is not possible to yield high bit scores, thus low E-values, for more distant Hr homologs over 120 aligned amino acid pairs.” – can you please provide evidence/reasoning?

Raw scores are calculated as the sum of substitution scores for individual paired amino acids in the alignment. E-values are negatively correlated to raw scores (and bit scores). For pairwise

sequence alignment, the shorter the length of a high-scoring segment pair (HSP), the lower the raw score, and the higher the E-value.

6. I will happily provide the sequences I found to the authors for their evaluation. They are the experts to check which gene they represent.

We thank Dr. Rensing again for sharing with us his own data. As we indicated in our response letter of last submission, the “suggestion of *YAN/AltYAN* existence requires finding both transcripts from the same genome”.

In this revision, we write “this refashioning event occurred at least in the *Physcomitrium-Physcomitrella* species complex” (lines 311-312). We also specifically add the following sentence: “Future thorough investigations on the distribution of *YAN/AltYAN* are needed to understand when this dual-coding gene evolved” (lines 312-314).

7. What is the meaning of “the annotation has already been changed in cosmoSS”?

The sequence was annotated as an “HHE-domain containing protein” in *Physcomitrella* annotation V1.6. Currently it is annotated in V3.3 as two different transcripts (i.e., *YAN* and *AltYAN* in this manuscript) without functional information.

Reviewers' comments:

Although Reviewer 3 doesn't have comments to the author, in comments to the editor, Reviewer 3 feels the concerns are not addressed.

Responses to comments

1 Performing the analyses proposed by Reviewer 3 in his last comments for Hr homolog identification (points 1 and 5)

We performed BLAST search of the NCBI non-redundant (*nr*) protein sequence database using three lower E-value thresholds (1e-6, 1e-8, and 1e-10, respectively). The only algal hit was from *Monoraphidium negelectum* (see figure below, 1e-8 as E-value threshold).

Organism	Blast Name	Score	Number of Hits	Description
cellular organisms			1688	
• Eukaryota	eukaryotes		997	
• • Viridiplantae	green plants		10	
• • • Embryophyta	land plants		8	
• • • • Marchantia polymorpha subsp. ruderalis	liverworts	246	1	Marchantia polymorpha subsp. ruderalis hits
• • • • Selaginella moellendorffii	club-mosses	230	4	Selaginella moellendorffii hits
• • • • Physcomitrella patens	mosses	196	1	Physcomitrella patens hits
• • • • Quercus suber	eudicots	107	2	Quercus suber hits
• • • Monoraphidium negelectum	green algae	90.1	2	Monoraphidium negelectum hits
• Sordaria macrospora k-hell	ascmycetes	157	2	Sordaria macrospora k-hell hits
• Phellinus noxius	basidiomycetes	152	1	Phellinus noxius hits
• Thielavia terrestris NRRL 8126	ascmycetes	150	2	Thielavia terrestris NRRL 8126 hits
• Spizellomyces punctatus DAOM BR117	chytrids	148	6	Spizellomyces punctatus DAOM BR117 hits
• Madurella mycetomatis	ascmycetes	147	2	Madurella mycetomatis hits
• Hydnomerulius pinastris MD-312	basidiomycetes	146	2	Hydnomerulius pinastris MD-312 hits
• Neurospora tetrasperma FGSC 2508	ascmycetes	146	2	Neurospora tetrasperma FGSC 2508 hits
• Neurospora tetrasperma FGSC 2509	ascmycetes	146	1	Neurospora tetrasperma FGSC 2509 hits
• Exophiala spinifera	ascmycetes	146	2	Exophiala spinifera hits
• Neurospora crassa OR74A	ascmycetes	145	2	Neurospora crassa OR74A hits
• Neurospora crassa	ascmycetes	145	1	Neurospora crassa hits
• Rhizopogon vinicolor AM-OR11-026	basidiomycetes	144	1	Rhizopogon vinicolor AM-OR11-026 hits
• Saitoella complicata NRRL Y-17804	ascmycetes	145	4	Saitoella complicata NRRL Y-17804 hits
• Rhizopogon vesiculosus	basidiomycetes	144	1	Rhizopogon vesiculosus hits
• Tulasnella calospora MUT 4182	basidiomycetes	144	2	Tulasnella calospora MUT 4182 hits
• Ramazzottius varieornatus	tardigrades	144	3	Ramazzottius varieornatus hits
• Chaetomium globosum CBS 148.51	ascmycetes	142	2	Chaetomium globosum CBS 148.51 hits
• Pisolithus microcarpus 441	basidiomycetes	142	1	Pisolithus microcarpus 441 hits
• Fomitiporia mediterranea MF3/22	basidiomycetes	142	2	Fomitiporia mediterranea MF3/22 hits
• Kwoniella heveanensis BCC8398	basidiomycetes	141	1	Kwoniella heveanensis BCC8398 hits
• Kwoniella heveanensis CBS 569	basidiomycetes	141	1	Kwoniella heveanensis CBS 569 hits
• Paxillus involutus ATCC 200175	basidiomycetes	141	1	Paxillus involutus ATCC 200175 hits
• • • • •	• • • • •	• • • • •	• • • • •	• • • • •

There were two hits from an angiosperm *Quercus suber*. These two hits were from the unpublished draft genome of this species that was recently submitted to NCBI on Dec 20, 2017. Both sequences were significantly more similar to fungal homologs than to other green plant sequences (up to 68% protein sequence identities with fungi versus 36-37% with the seedless vascular plant *Selaginella moellendorffii*). It is unclear whether these two sequences are derived from sequencing contamination or a recent HGT event. We noted the above observation in **Supplementary Note 1**.

Using the *Monoraphidium* sequences identified above as query, we performed further BLAST search. Several additional algal hits were detected, all of which were from the group Chlorophyceae (see figure below, 1e-8 as E-value threshold). We have added this information in **Supplementary Figure 1**.

Organism	Blast Name	Score	Number of Hits	Description
cellular organisms			1727	
• Eukaryota	eukaryotes		718	
• • Viridiplantae	green plants		18	
• • • Chlorophyceae	green algae		11	
• • • • Monoraphidium neglectum	green algae	544	6	Monoraphidium neglectum hits
• • • • Gonium pectorale	green algae	118	1	Gonium pectorale hits
• • • • Tetraabaena socialis	green algae	114	1	Tetraabaena socialis hits
• • • • Chlamydomonas reinhardtii	green algae	110	1	Chlamydomonas reinhardtii hits
• • • • Volvox carteri f. nagariensis	green algae	107	2	Volvox carteri f. nagariensis hits
• • • Selaginella moellendorffii	club-mosses	93.1	4	Selaginella moellendorffii hits
• • • Quercus suber	eudicots	91.9	2	Quercus suber hits
• • • Physcomitrella patens	mosses	83.4	1	Physcomitrella patens hits
• • Colletotrichum chlorophyti	ascomycetes	120	1	Colletotrichum chlorophyti hits
• • Coniochaeta lignaria NRRL 30616	ascomycetes	120	1	Coniochaeta lignaria NRRL 30616 hits
• • Colletotrichum orbiculare MAFF 240422	ascomycetes	117	1	Colletotrichum orbiculare MAFF 240422 hits
• • Allomyces macrogynus ATCC 38327	blastocladiomycetes	116	2	Allomyces macrogynus ATCC 38327 hits
• • Trichoderma reesei QM6a	ascomycetes	115	4	Trichoderma reesei QM6a hits
• • Trichoderma reesei RUT C-30	ascomycetes	115	2	Trichoderma reesei RUT C-30 hits
• • Trichoderma parareesei	ascomycetes	114	2	Trichoderma parareesei hits
• • Coniosporium apollinis CBS 100218	ascomycetes	111	2	Coniosporium apollinis CBS 100218 hits
• • Podospora anserina S mat+	ascomycetes	110	9	Podospora anserina S mat+ hits
• • Hypoxylon sp. EC38	ascomycetes	110	3	Hypoxylon sp. EC38 hits
• • Pyronema omphalodes CBS 100304	ascomycetes	110	1	Pyronema omphalodes CBS 100304 hits
• • Fusarium pseudograminearum CS3096	ascomycetes	109	2	Fusarium pseudograminearum CS3096 hits
• • Fusarium pseudograminearum CS3220	ascomycetes	109	1	Fusarium pseudograminearum CS3220 hits
• • Fusarium pseudograminearum CS3427	ascomycetes	109	1	Fusarium pseudograminearum CS3427 hits
• • Fusarium pseudograminearum CS3487	ascomycetes	109	1	Fusarium pseudograminearum CS3487 hits
• • Elaphomyces granulatus	ascomycetes	113	1	Elaphomyces granulatus hits
• • Spizeliomyces punctatus DAOM BR117	chytrids	108	6	Spizeliomyces punctatus DAOM BR117 hits
• • Fusarium oxysporum f. sp. melonis 26406	ascomycetes	107	7	Fusarium oxysporum f. sp. melonis 26406 hits
• • Schizosaccharomyces pombe	ascomycetes	107	2	Schizosaccharomyces pombe hits
• • Schizosaccharomyces pombe 972h-	ascomycetes	107	1	Schizosaccharomyces pombe 972h- hits

From the above results, it can be seen that both land plant and algal sequences share higher similarities (by bit scores) with fungal homologs than with each other. Importantly, all algal hits identified from the above searches were from the group Chlorophyceae. No hits from red algae and other green algae were found, even though over 20 red algae and other non-Chlorophyceae green algae have been sequenced and their complete genome sequence data are available in NCBI.

Based on the above searches, we sampled sequences from different groups, including green algae, and again performed phylogenetic analyses. The results were consistent with our previous analyses (see figures below). In all these analyses, land plant sequences are more closely related to fungal and tardigrade sequences than to algal sequences. This finding is also consistent with the results of pairwise sequence comparisons from BLAST search.

Comparisons of phylogenetic analyses. The tree on the top was generated by sampling sequences from the BLAST searches described above (e.g., using land plant and *Monoraphidium* sequences as queries with an E-value threshold $1e-8$). The tree on the bottom includes sequences sampled from the BLAST search result using 1.0 as E-value threshold. Although the topologies of these two trees are slightly different, both suggest that land plant sequences are most closely related to tardigrade and fungal homologs.

2. Further toning down the claims on Hr evolution (providing evidence supporting contamination, and stating Hr sequences are found in algae and plant so that no hypothesis is better supported than the other/or without making firm conclusions on the relationship between algal and land plant Hr, points 2).

We provided a comprehensive discussion on algal sequences from the 1KP database in the **manuscript, Supplementary Note 1**, and **Figure 4a** in our last and current submissions. Algal sequences from non-Chlorophyceae groups have also been included in our previous and current submissions. We noted that these algal sequences usually shared the highest similarity with fungal and bacterial homologs in the main text (lines 92-102). We have been trying to stay unbiased in our writing and indicated our concerns of contamination for some algal sequences from 1KP ONLY in **Supplementary Note 1** (we, however, did note in the main text that the hemerythrin sequences in many vascular plants were most likely due to contamination). Although confirmation of possible contaminations requires comprehensive wet-lab experiments, which is beyond the scope of our current study, most readers should be able to make their own judgments based on the presented data. Specifically, our concerns about contamination are based on the following evidence.

- a. Because complete genome sequence data often have longer scaffolds, they provide a reliable source for assessing the existence of a gene in a given genome. **Numerous hemerythrin hits were found from angiosperms (and other vascular plants) in 1KP data, but no hemerythrin has been annotated in any published angiosperm genome thus far.** Only two hits from an unpublished angiosperm draft genome (i.e., *Quercus suber*) were found in our BLAST search of the NCBI *nr* database, which contains complete genome sequence data of many angiosperms. Most importantly, all identified “angiosperm” hemerythrins, including those from *Quercus suber*, are most similar or most closely related to various fungal sequences (see figure below for species underlined in red, which include two angiosperms and two other vascular plants and a hornwort).

- b. Complete genome sequence data for over 20 red algae and non-Chlorophyceae green algae are available in NCBI, but no hemerythrin homologs were found in any of these species (see our response to comment 1 above). On the hand, although hemerythrin homologs were found from several of these algal groups in 1KP, when they were used to search the *nr* database, the most similar hits were all from either fungi or bacteria (see **Supplementary Note 1** for details). No hit has been found from any other non-Chlorophyceae green algal species whose complete genome and scaffold information is available.
- c. The contamination issue of 1KP data sometimes can be obvious. This can be evidenced by a sequence from the liverwort *Treubia lacunosa* (1KP ID: FITN-2089742). This liverwort sequence of 1295 base pairs shares 100% nucleotide identity with a lycophyte sequence (1KP ID: PYHZ-2006808). Liverworts and lycophytes are two different major groups of land plants, the former being nonvascular plants where the latter being seedless vascular plants. We noted this observation in the **Supplementary Figure 3b**.

We deleted all discussion on the specific HGT scenarios. The evolution of hemerythrin in plants is kept as minimal as possible. We now have the following sentence in this revision:

“It is likely that land plant Hr was ultimately derived from other organisms, possibly fungi, but other scenarios, such as vertical inheritance combined with lineage-specific gene loss, cannot be ruled out. Additional investigations are needed to understand the origin of Hr in land plants”. (lines 293-296).

REVIEWERS' COMMENTS:

Although Reviewer 3 doesn't have remarks to the author, in the remarks to the editor Reviewer 3 feels OK with the present version of the manuscript.